# Dreaming in Code for Curriculum Learning in Open-Ended Worlds

Konstantinos Mitsides [1]   Maxence Faldor [1]   Antoine Cully [1]

## Abstract

Open-ended learning frames intelligence as emerging from continual interaction with an ever-expanding space of environments. While recent advances have utilized foundation models to programmatically generate diverse environments, these approaches often focus on discovering isolated behaviors rather than orchestrating sustained progression. In complex open-ended worlds, the large combinatorial space of possible challenges makes it difficult for agents to discover sequences of experiences that remain consistently learnable. To address this, we propose Dreaming in Code (DiCode), a framework in which foundation models synthesize executable environment code to scaffold learning toward increasing competence. In DiCode, "dreaming" takes the form of materializing code-level variations of the world. We instantiate DiCode in Craftax, a challenging open-ended benchmark characterized by rich mechanics and long-horizon progression. Empirically, DiCode enables agents to acquire long-horizon skills, achieving a 17% improvement in mean return over the strongest baseline and non-zero success on late-game combat tasks where prior methods fail. Our results suggest that code-level environment design provides a practical mechanism for curriculum control, enabling the construction of intermediate environments that bridge competence gaps in open-ended worlds.

## 1. Introduction

While the central promise of open-ended learning lies in the emergence of unbounded intelligence, agents operating in such vast domains often exhibit a familiar trajectory: rapid early gains followed by a pronounced performance plateau (Wang et al., 2019; Küttler et al., 2020; Matthews et al., 2024; 2025). Despite substantial advances in learning algorithms and agent architectures (Bauer et al., 2023; Hafner et al., 2025), progress in open-ended worlds does not automatically follow from knowing "how" to learn when suitable experience is lacking (Clune, 2020; Jiang et al., 2023). Sustaining improvement therefore requires a continual stream of experiences that remain both novel and learnable, which in turn demands mechanisms for actively shaping and generating an agent's experience over time (Bengio et al., 2009; Wang et al., 2019; Dennis et al., 2020; Hughes et al., 2024).

This challenge has been studied under the framework of Unsupervised Environment Design (UED), which seeks to automatically adapt or generate environments to maintain a "Goldilocks" level of difficulty for learning agents (Dennis et al., 2020). By controlling the environments from which experience is drawn, UED addresses the stagnation that arises when fixed environments cease to offer meaningful learning signal (Jiang et al., 2021a; Parker-Holder et al., 2022). However, most UED methods are restricted to low-dimensional parameters and rely on search procedures that assume a smooth, well-structured design space (Parker-Holder et al., 2022). These assumptions are restrictive in open-ended domains, where sustaining learning requires a curriculum of structurally evolving environments that introduce long-horizon dependencies (Matthews et al., 2024). As a result, despite its conceptual appeal, the application of UED to truly open-ended problems remains limited.

Recent progress in environment design has begun to relax these limitations by representing environments as executable programs (Liang et al., 2024; Faldor et al., 2025). Instead of tuning a fixed set of parameters, environment logic can now be programmatically specified and composed, enabling richly structured worlds with diverse dynamics. Leveraging foundation models (FMs), Faldor et al. (2025) have shown that such expressive programmatic environment spaces can be effectively explored to synthesize environments that are novel and learnable in isolation. However, because these methods treat environments as disjoint challenges, they do not focus on generating the curricula required to sustain

[1]Department of Computing, Imperial College London, London, United Kingdom. Correspondence to: Konstantinos Mitsides <konstantinos.mitsides23@imperial.ac.uk>.

*Proceedings of the 43rd International Conference on Machine Learning*, Seoul, South Korea. PMLR 306, 2026. Copyright 2026 by the author(s).

Project page and code: konstantinosmitsides.github.io/dreaming-in-code

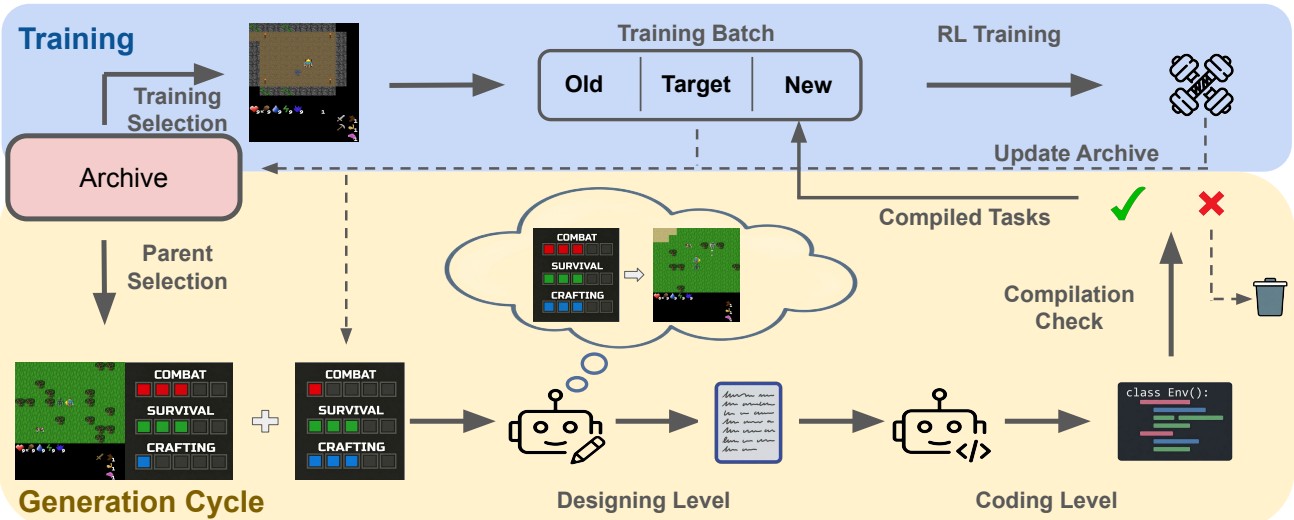

*Figure 1.* **Overview of the Dreaming in Code framework.** The pipeline consists of two processes: Training (top) and the Generation Cycle (bottom). In the generation cycle, a parent level is selected from the Archive based on learnability. Conditioning the foundation model on the parent level and the agent's current competence, it synthesizes a new level description and subsequent executable Python code. Levels that pass a compilation check are added to the Training Batch, which mixes the target environment, newly generated levels, and archived levels sampled via PLR. Agent performance and new levels update the archive, closing the curriculum loop.

progress in open-ended domains. In such settings, sustained learning requires coordinating sequences of environments that progressively build on prior capabilities. These considerations highlight the need for UED methods that can operate directly over programmatic environment representations to orchestrate this structural evolution.

We bridge this gap with Dreaming in Code (DiCode), a UED framework designed to drive progress in complex, open-ended target environments – those in which the agent must make sustained progress. In DiCode, an FM 'dreams' new environment instances by synthesizing executable generation logic, conditioned on the agent's current capabilities. Crucially, this logic is executed by a fixed world engine. This engine can take various forms depending on the application, such as a game engine for video games (e.g., Craftax (Matthews et al., 2024)) or a physics engine for robotics (e.g., MuJoCo (Todorov et al., 2012)). By utilizing the engine directly rather than learning a world model, Di-Code ensures that all generated experiences adhere to valid physics and consistent mechanics. Consequently, the act of dreaming here serves not to improve sample efficiency, but to construct a curriculum that enables agents to acquire increasingly complex behaviors in open-ended worlds.

We instantiate DiCode in Craftax, a challenging open-ended reinforcement learning (RL) environment built around a procedurally generated world with rich mechanics and long-horizon progression. Crucially, we utilize an open-weights FM for generation, demonstrating that our framework is effective without relying on proprietary or non-reproducible APIs. Empirically, DiCode enables agents to acquire long-

horizon skills, such as late-game combat, that remain intractable ($0\%$ success) for standard RL and prior UED methods. These results indicate that generating environment code enables practical curriculum control, allowing agents to sustain learning progress in complex, open-ended domains.

Specifically, we make the following contributions: **(1)** We introduce Dreaming in Code (DiCode), a UED framework that generates executable environment code to shape agent learning trajectories in open-ended worlds. **(2)** We demonstrate that DiCode scaffolds the acquisition of complex behaviors that are otherwise unattainable for state-of-the-art baselines, achieving non-zero success rates on tasks where prior methods fail completely, while improving mean return by $17\%$. **(3)** We provide qualitative analysis revealing that the FM spontaneously develops "teacher-like" strategies – such as removing resource scaffolding to increase difficulty – maintaining the agent in a zone of proximal development. **(4)** We provide ablation studies along four design axes – parent selection, environment reshaping, FM capability, and closed-loop grounding – identifying the contribution of each component and confirming that curriculum quality scales with FM reasoning capability.

## 2. Background

### 2.1. Problem Setting

We model the Reinforcement Learning (RL) (Sutton & Barto, 2018) problem as an Underspecified Partially Observable Markov Decision Process (UPOMDP) (Dennis et al., 2020) denoted by $(\mathcal{L}, S, O, A, r, \mathcal{T}, \rho, \mathcal{I}, \gamma)$. Here, $\mathcal{L}$ is the

space of all possible levels, $A$ and $S$ represent the state and action spaces, respectively, $\mathcal{I} : S \to O$ is the inspection function that maps states to observations, and $\gamma$ is the discount factor. The reward function, $r : \mathcal{L} \times S \times A \to \mathbb{R}$, the transition function, $\mathcal{T} : \mathcal{L} \times S \times A \to \Delta(S)$, and the initial state distribution, $\rho : \mathcal{L} \to \Delta(S)$, all depend on $\lambda$. Training is conducted over a subset $\Lambda \subseteq \mathcal{L}$, where every level $\lambda \in \Lambda$ specifies a distinct POMDP. We define the agent's parameter space as $\mathcal{X}$. The agent operates under a policy $\pi : \mathcal{X} \times O \to \Delta(A)$, which may depend on a hidden state $h$ to handle partial observability and a goal $g$ to handle different reward functions across levels. With $J : \mathcal{L} \times \mathcal{X} \to \mathbb{R}$ representing the expected discounted return for a specific level, the agent's objective is to maximize performance over a distribution of levels $\Lambda(y)$ parametrized by $y$:

$$\max_{x \in \mathcal{X}} \mathbb{E}_{\lambda \sim \Lambda(y)} \left[ J(\pi_x, \lambda) \right] \qquad (1)$$

### 2.2. Unsupervised Environment Design

To drive the emergence of an increasingly capable agent, Unsupervised Environment Design (UED) (Dennis et al., 2020) structures the learning process as a curriculum-generating game between a "student" agent and a "teacher" level generator. In this framework, the teacher generates levels $\lambda$ by maximizing a utility function $U_t(\pi, \lambda)$, while the student maximizes expected return in the standard RL manner. This formulation subsumes Domain Randomization (Tobin et al., 2017) as a specific instance where the teacher's utility is fixed to a constant value, thereby reducing the curriculum generation to random sampling. Another common teacher objective is to maximize agent regret, defined as the gap between a policy's expected return on a level and the optimal return. In complex environments, exact regret is intractable because it requires the optimal policy, so practical methods rely on heuristic proxies such as Positive Value Loss (PVL) and Maximum Monte Carlo (MaxMC) (Jiang et al., 2021a; Parker-Holder et al., 2022). More recently, an alternative objective has been proposed for binary-outcome domains that prioritizes levels with high learnability – levels that the agent can solve intermittently but has not yet mastered (Tzannetos et al., 2023). Given a success rate $p$ on a level, learnability is defined as $p(1 - p)$. Critically, Rutherford et al. (2024) showed that MaxMC and PVL correlate poorly with learnability; their proposed method addresses this by prioritizing learnability directly, often showing benefits over the other two heuristics. Motivated by these findings, we adopt the learnability score to curate environment levels.

### 2.3. Prioritized Level Replay

Prioritized Level Replay (PLR) (Jiang et al., 2021b;a) is a general and empirically effective UED method that has been widely adopted and extended in subsequent work (Parker-Holder et al., 2022). It alternates between two mechanisms

at each training iteration: with a fixed probability, it generates new levels by sampling environment parameter configurations, and otherwise replays levels from a fixed sized replay buffer. Newly generated levels are evaluated by the agent and assigned a score $f(\lambda_i) = S_i$, typically based on heuristics such as PVL or MaxMC. Levels with high score are added to the buffer, and the worse ones are discarded. During replay, levels are sampled from the buffer according to a score-weighted and usage-weighted distribution. Specifically, the probability of sampling a level $\lambda_i$ is given by

$$P(\lambda_i) = (1 - \tau) \frac{h(S_i)^{1/\beta}}{\sum_j h(S_j)^{1/\beta}} + \tau \frac{c - C_i}{\sum_{C_j \in C} c - C_j} \quad (2)$$

where $h(S_i) = 1/\text{rank}(S_i)$, and $\text{rank}(S_i)$ denotes the rank of the score $S_i$ among all stored levels when sorted in descending order. The temperature $\beta$ controls the sharpness of prioritization. The second term assigns probability mass in proportion to a level's staleness $c - C_i$. Here, $c$ denotes the total number of times a level was sampled so far for training, while $C_i$ is the time at which $\lambda_i$ was last sampled, and $\tau \in [0, 1]$ trades off between score-based sampling and staleness-based sampling. The idea here is to sample levels that have high score, or they have not been sampled for a long time. We leverage this prioritization scheme to govern the sampling of levels from the replay buffer.

## 3. Dreaming in Code

Dreaming in Code (DiCode) is a UED framework that shapes learning by synthesizing executable environment code. Its primary objective is to enable the agent to master a specific, fixed target environment (e.g., the Craftax game), which is often complex to solve directly. To facilitate progress toward this goal, DiCode employs a process we term "dreaming": utilizing a foundation model (FM) to conceptualize and imagine the next optimal training scenario, tailored to the agent's current skill frontier, and materializing it into executable level. These generated levels act as stepping stones, bridging the gap between the agent's initial capabilities and the demands of the target environment. They are integrated into training alongside the target environment itself, establishing a closed-loop curriculum where the agent's evolving skill set continuously guides the generative process.

### 3.1. Environment Search Space

In DiCode, an FM generates environments as executable Python programs compatible with the target simulator engine via a custom interface (see Appendix A.1). Given a context $c$, we sample programs, from the conditional distribution induced by a pre-trained FM:

$$(\rho_\lambda, \mathcal{T}_\lambda, g_\lambda) \sim P_{\text{FM}}(\cdot | c) \qquad (3)$$

where $\rho_\lambda, \mathcal{T}_\lambda$, and $g_\lambda$ are the level-specific initial state distribution, transition function, and goal respectively.

It is crucial to distinguish this formulation from the existing Procedurally Content Generation (PCG) UED. In prior work, a "level" typically refers to a fixed random seed that instantiates a single static layout under invariant game rules. In DiCode, a "level" is the executable code that programmatically defines both the world generation and the interaction rules. Consequently, each generated $\lambda$ specifies a distinct POMDP with unique transition dynamics ($\mathcal{T}_\lambda$) – the logic governing game mechanics and entity interactions – and a stochastic initial state distribution ($\rho_\lambda$) that yields a new procedural layout every episode.

To ground this in our experimental domain, the generated code modulates Craftax through a highly expressive programmatic interface. For the initial state $\rho_\lambda$, the generator can algorithmically specify the world topology, placing any combination of blocks, mobs, or resources, and equipping the agent with arbitrary starting inventories and conditions. For the transition dynamics $\mathcal{T}_\lambda$, the code redefines interaction rules, such as combat formulas (e.g., damage scaling, health thresholds) and progression logic (e.g., unlocking conditions for new dungeon floors). Finally, the goal $g_\lambda$ is synthesized as a logical composition of specific in-game achievements, defining the success criteria for the generated world.

The reward and termination structure of generated levels mirrors the target environment, except for goal completion. When a level-defined goal is satisfied, the episode terminates and the agent receives a fixed, objective-agnostic bonus reward $B_t$. This bonus addresses a fundamental alignment problem: the target environment provides dense early-game rewards (e.g., for collecting resources or eating food), and without an explicit goal-completion incentive, the agent exploits these readily available rewards instead of pursuing the level's designed objective. This undermines the core benefit of targeted environment shaping – generated levels become no more useful than the target environment itself, since the agent never engages with the intended skill. Critically, it also breaks the FM's feedback loop: the FM cannot distinguish a level that is genuinely too hard from one the agent is simply ignoring, producing inaccurate learnability signals that corrupt curriculum decisions. The adaptive bonus resolves this by making goal completion the dominant reward event, restoring accurate learnability estimates and enabling the FM to meaningfully shape the agent's progression.

Formally, let $r_{\text{target}}$ be the native reward function of the target environment. The reward function for a generated level, $r_\lambda$, is defined as:

$$r_\lambda(s, a) = r_{\text{target}}(s, a) \cdot \mathbb{I}_{\text{init}} + \mathbb{I}_{\text{success}} \cdot B_t \qquad (4)$$

where $\mathbb{I}_{\text{init}}(s)$ is a binary mask that returns 0 if the achieve-

ment associated with the reward is already satisfied by the initial state of $\lambda$, and 1 otherwise. To ensure the bonus remains attractive as the agent improves, we adaptively scale $B_t$ at training cycle $t$:

$$B_t = \max(d, 2 \times R_{t-1}), \qquad (5)$$

where $R_{t-1}$ is the agent's expected return on the target environment in the previous cycle, and $d$ is a minimum floor. Additionally, to disambiguate level-dependent goals, the agent's policy is conditioned on a multi-hot encoding indicating the active achievements of the current level.

### 3.2. Generation Cycle

Except for the initialization phase – where the agent begins training on pre-designed seed levels (see Appendix A.2) – each generation cycle consists of four sequential steps (see Figure 1): 1) DiCode selects a parent level from an archive that stores level-related information, 2) given the selected parent and associated performance metrics, it generates a natural language description for the new level, 3) given this description, it generates an executable Python program, and 4) it validates the program though a compilation check.

**Archive**   Levels are stored in an archive structured as a directed graph where nodes represent levels (containing executable code, metadata, and performance statistics) and edges represent parent–offspring relationships. For each level, we maintain a performance profile based on the agent's most recent success rate (SR), and any other information related to the agent's capabilities. Here, we include the list of achievement SRs of the agent. We define the status mapping $S(\lambda)$ for a level $\lambda$ based on agent's recent success rate $SR_\lambda$ as: $S(\lambda) = A$ if $SR_\lambda \geq 0.75$; $S(\lambda) = B$ if $SR_\lambda \in [0.50, 0.75)$; $S(\lambda) = C$ if $SR_\lambda \in [0.25, 0.50)$; and $S(\lambda) = D$ otherwise.

**Selection**   We employ a selection strategy designed to promote the diversity of valid evolutionary lineages rather than over-sampling a single successful branch. Let $\mathcal{A}$ be the set of all existing levels in the archive, and let $\mathcal{C}(\lambda)$ denote the set of offspring for level $\lambda$. We define the set of eligible candidates, $\mathcal{A}_{\text{cand}} \subseteq \mathcal{A}$, to be:

$$\mathcal{A}_{\text{cand}} = \{\lambda \in \mathcal{A} \mid S(\lambda) \in \{A, B\} \wedge \forall c \in \mathcal{C}(\lambda), S(c) = D\}. \qquad (6)$$

Then, we sample a parent level $\lambda_p$, with probability

$$P(\lambda) = \begin{cases} \frac{f(\lambda)}{\sum_{k \in \mathcal{A}_{\text{cand}}} f(k)} & \text{if } \lambda \in \mathcal{A}_{\text{cand}} \\ 0 & \text{otherwise} \end{cases} \qquad (7)$$

where $f(\lambda)$ denotes the learnability score (see Section 2.2).

**Description & Code**  To generate a new level, the FM receives the following context: 1) pre-defined domain-specific context $c_1^{\text{target}}$, 2) the description of the parent level $\lambda_p$ along with its performance profile $\text{perf}_p$, 3) the performance profile of the target environment $\text{perf}_{\text{target}}$, and 4) pre-defined mutation instructions $m_1$. Once generated, this description serves as the input for a second inference step, where the model utilizes a pre-defined domain-specific context $c_2^{\text{target}}$, few-shot examples $\{e\}_{i=1}^n$ retrieved based on similarity to $\lambda_p$, and instructions $m_2$ to synthesize the executable level (see Appendix B).

Therefore, mapping the notation from Section 2, $\Lambda(y)$ denotes the distribution parameterized by the total context $y = (c_1^{\text{target}}, c_2^{\text{target}}, \text{perf}_p, \text{perf}_{\text{target}}, m_1, m_2, \{e\}_{i=1}^n, \lambda_p)$. The offspring level $\lambda_o$ is determined by a hierarchical sampling process. First, we sample a latent description $h$:

$$h \sim P_{\text{FM}}(\cdot | c_1^{\text{target}}, \text{perf}_p, \text{perf}_{\text{target}}, m_1, \lambda_p). \quad (8)$$

Then, we sample the program conditioned on $h$:

$$(\rho_{\lambda_o}, \mathcal{T}_{\lambda_o}, g_{\lambda_o}) \sim P_{\text{FM}}(\cdot | c_2^{\text{target}}, \{e\}_{i=1}^n, m_2, h). \quad (9)$$

**Compilation Check**  To guarantee a full batch of valid levels, we generate and validate a surplus of candidates in parallel. Validation consists of the agent executing a short trajectory in the environment level to filter out code that fails to compile or throws runtime errors. We do not perform self-correction on failed code, as empirical results indicated that the computational cost of iterative refinement outweighed the marginal gain in yield. We quantify compilation yield empirically in Section A.5.

### 3.3. Training

We train the RL agent by constructing stratified training batches composed of trajectories from three distinct sources: the target environment, newly generated levels, and archived levels. To ensure grounded progress and prevent distributional shift, we allocate a fixed 20% of the simulation budget in every update to the target environment. This allocation also provides the target performance profile ($\text{perf}_{\text{target}}$) fed back to the FM as generation context. The remaining budget is distributed between newly generated levels and replaying archived levels.

To balance policy stability with curriculum progression, newly generated levels are introduced every $v = 2$ iterations, similarly to how PLR controls the frequency of new level generation. Moreover, when replaying levels from the archive, we utilize the PLR mechanism (Equation 2), using the learnability score derived from the agent's performance over the most recent $N$ training episodes.

**Asynchronous Generation**  To mitigate the inference latency of FMs, DiCode generates levels asynchronously. RL training proceeds concurrently with generation and only blocks if $v - 1$ training cycles elapse without a new batch of valid levels being ready, ensuring high GPU utilization. We quantify this empirically in Section A.4.

## 4. Experiments

### 4.1. Setup

**Benchmark**  We evaluate DiCode on Craftax (Matthews et al., 2024), a challenging open-ended benchmark accelerated in JAX (Bradbury et al., 2018). Craftax places agents in an infinite, procedurally generated world with distinct biomes, requiring mastery of diverse mechanics, including combat, survival, resource gathering, and building. We select Craftax because its open-ended structure, with deep hierarchies and compositional dependencies, presents a challenge distinct from standard UED domains. Unlike flat grid-worlds, progress in Craftax requires navigating a complex technology tree where basic survival and crafting skills are strict prerequisites for advanced capabilities, making it a rigorous test of sustained long-horizon learning.

**Baselines**  We compare DiCode against UED baselines commonly used on Craftax: Prioritized Level Replay (PLR) (Jiang et al., 2021a), Sampling for Learnability (SFL) (Rutherford et al., 2024), and Domain Randomization (DR). To isolate the contribution of the curriculum mechanism from the optimization process, we standardize the underlying RL agent across all methods to PPO with Gated Transformer-XL (PPO-GTrXL) (Parisotto et al., 2020), which is currently the state-of-the-art solver for Craftax (Hamon, 2024). We use a shared PPO-GTrXL implementation across DiCode and all UED baselines, adapting prior UED code from (Monette et al., 2025) where necessary. UED baselines curate over Craftax's random seeds, which determine the terrain layout of each floor – the spatial arrangement of resources, water, stone, lava, and other environmental features. The seed space does not control game mechanics or skill dependencies, which are governed by the technology tree's internal logic. We further include PPO-GTrXL trained on the default Craftax distribution as a non-curriculum reference baseline. Finally, we distinguish between DR and PPO-GTrXL in terms of their training protocols: DR periodically resamples its seed buffer and resets within the same seeds before resampling, following standard Craftax protocol (Matthews et al., 2024; Monette et al., 2025); PPO-GTrXL samples a fresh seed each episode without reuse.

**Training and Evaluation**  We train all methods for $2 \times 10^9$ environment steps across 8 random seeds (unless otherwise stated), instantiating the DiCode generator with the open-weights `Qwen3-235B` model (Yang et al., 2025). For Di-

Code, this budget includes steps from both the target environment and generated levels. Note that SFL performs environment rollouts without RL updates during its level evaluation phase, resulting in fewer gradient updates for the same number of environment steps. During training, we archive policy checkpoints at 50 uniformly spaced intervals. We evaluate each checkpoint on a fixed held-out test set of 1024 procedurally generated Craftax instances, reporting mean return and standard error across seeds. For implementation details and runtime analysis see Appendices A.3 and A.4.

### 4.2. Results

Our experiments aim to answer three key questions: **(1)** Does DiCode enable agents to acquire capabilities and reach competence levels that are unattainable with standard RL or existing UED methods? **(2)** Does the generative process induce a meaningful curriculum that supports learning over time? **(3)** What are the key design components driving DiCode's performance, and how does each contribute?

**Performance on Craftax**  To answer question **(1)**, we compare DiCode against PPO-GTrXL and the leading UED baselines. Figure 2 illustrates the aggregate performance on the held-out test set over the course of training. DiCode establishes a statistically significant lead over the best-performing baseline early in the training process and maintains this dominance throughout the entire training budget. Ultimately, DiCode achieves a final mean return of 48.55, substantially outperforming the strongest baseline which reaches 41.59 – a relative improvement of $\sim 17\%$.

To dissect the source of the performance gap, Figure 3 presents the final success rates for specific in-game achievements ordered by their hierarchical depth (for all achievement results see Appendix D.1). The results show that

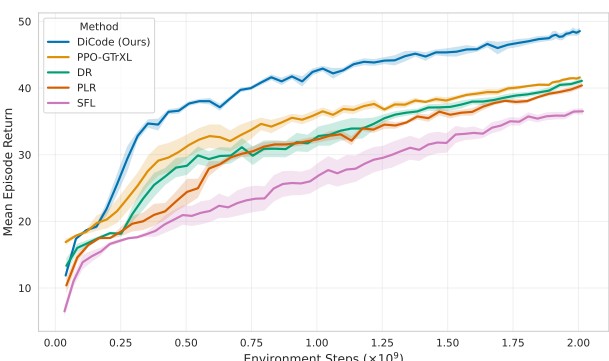

*Figure 2.* **Performance on Craftax.** Mean episode return on the held-out test set (1024 unseen procedurally generated worlds) throughout training. Shaded regions indicate the standard error across 8 seeds.

DiCode's advantage is not merely a uniform improvement, but a structural breakthrough in overcoming specific exploration bottlenecks. First, DiCode dominates on instrumental milestones – subgoals that are not terminal objectives but are critical for surviving long enough to progress. For instance, on *Make Iron Armour*, a crucial prerequisite for survivability, DiCode achieves a success rate of $47\%$, a dramatic improvement over the best baseline's $15\%$. This suggests that while baselines struggle to prioritize defensive preparations, DiCode's curriculum successfully teaches the agent to "gear up" before venturing further. This mastery of instrumental skills directly enables deeper exploration. Because DiCode agents are better equipped, they survive the transition to harder floors significantly more often, entering the *Gnomish Mines* (Floor 2) in $32\%$ of episodes compared to just $9\%$ for the strongest baseline.

Most critically, this extended horizon allows DiCode to master late-stage objectives that remain effectively intractable for standard methods. As shown in Figure 3, baseline performance collapses to $0\%$ on advanced combat tasks like *Defeat Gnome Warrior* and *Defeat Gnome Archer*. In contrast, DiCode achieves success rates of $15\%$ and $11\%$ respectively, demonstrating that "dreaming" of these specific combat scenarios creates the necessary gradient for the agent to learn them. Even on resource-intensive tasks like *Make Diamond Sword*, DiCode doubles baseline success ($6\%$ vs. $3\%$), confirming that its curriculum enables learning of deep hierarchical dependencies.

**Qualitative Analysis**  To answer question **(2)**, Figure 4 visualizes the curriculum dynamics from iteration 15 to 100. At a global scale, we observe a clear semantic progression in the generated levels. Early levels (e.g., Level 112) rely on generous initializations – such as pre-built workstations, enhanced inventories, and more resources nearby – to bypass prerequisites for initial skills like crafting Iron Armour. As training progresses, the model synthesizes levels like Level 287, which expands the core objective and introduces higher mob pressure, effectively layering combat complexity onto the crafting task. Ultimately, this trajectory enables the discovery of deep exploration tasks, such as Level 532 (descending to floor 2), a bottleneck state required to reach the Gnome adversaries rarely visited by agents trained on standard baselines (for more details see Appendix C.1).

At the local level, the FM drives this progression through targeted programmatic mutations. The inset in Figure 4 $(112 \rightarrow 143)$ illustrates this "teacher-like" behavior: the model detects high agent competence and generates an offspring level specifically by removing the resource and workstation scaffolding (red text) and thus consequently expanding the objective (green text). Crucially, the bottom panel in Figure 4 confirms the efficacy of this adaptive mechanism. The average success rate across training levels remains sta-

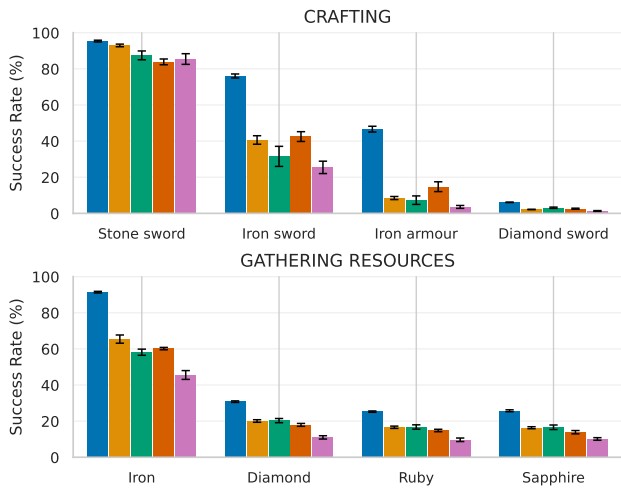
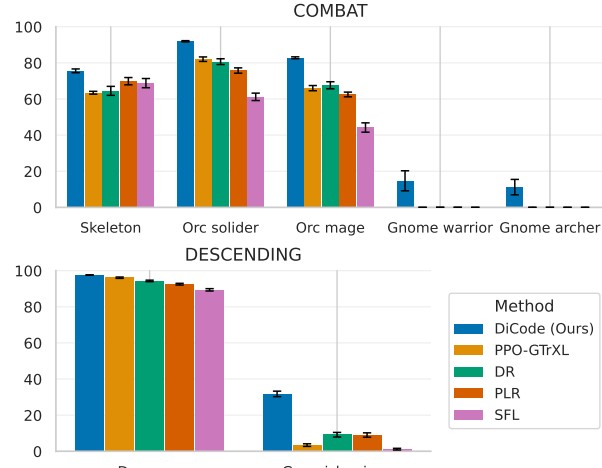

*Figure 3.* **Achievement Breakdown.** Final success rates on selected achievements, ordered by hierarchical depth (left to right). DiCode consistently outperforms all baselines across all evaluated achievements. The performance gap is particularly significant in two key areas: 1) on instrumental milestones (e.g. Iron sword, Iron armour) which are prerequisites for sustaining long-term progress, and 2) on late-stage objectives (e.g. Gnomish archer, Gnome warrior) where baseline performance effectively collapses to zero, rendering them intractable for prior methods. Error bars denote standard error across 8 seeds.

ble at approximately $0.5$ throughout the run, indicating that DiCode successfully maintains the agent in its zone of proximal development, continuously matching level difficulty to the agent's growing capabilities. See our project page for more details.

**Ablation Analysis** To answer question (**3**), we ablate DiCode along four design axes: parent selection strategy, environment reshaping, FM capability, and closed-loop grounding. Table 1 reports the final mean return for each variant; learning curves and per-achievement breakdowns are provided in Appendix D.2 (Figures 8, 9).

*Table 1.* **Ablation Analysis.** Final mean return on the held-out test set. DiCode uses 8 seeds; all ablation variants use 5 seeds. $\pm$ denotes standard error.

| Method | Mean Return |
| --- | --- |
| DiCode (Ours) | **48.55 $\pm$ 0.49** |
| Random Parent Sampling | 46.44 $\pm$ 0.41 |
| Qwen80B | 45.88 $\pm$ 0.40 |
| Qwen30B | 44.38 $\pm$ 0.66 |
| Goal Only | 42.82 $\pm$ 0.39 |
| Open-Loop (DiCode-OL) | 41.12 $\pm$ 0.77 |

**Random Parent Sampling** replaces DiCode's learnability-prioritized parent selection with uniform random selection while retaining the full generation context. Mean return drops modestly, but the per-achievement breakdown (Appendix D.2, Figure 9) reveals a qualitative failure: success on late-game combat tasks collapses to near-zero (Gnome Warrior: 0.6%, Gnome Archer: 0.1% vs. DiCode's 14.7% and 11.2%). This indicates that principled parent selection

is not critical for early/mid-game improvement but is essential for sustaining progression to deep objectives – the FM must build on the right stepping stone to bridge late-game bottlenecks.

**Goal Only** restricts the FM to selecting a goal for each level without modifying initial states or transition dynamics. Performance drops to near-baseline (42.82), confirming that environment reshaping – the ability to provide scaffolding such as enhanced inventories, simplified mechanics, or adjusted spawn rates – is the primary driver of DiCode's gains. Goal selection alone, even with the adaptive bonus, is insufficient.

**FM capability scaling** holds all prompts, API, and hyperparameters fixed while varying only the FM: Qwen3-235B $\rightarrow$ 80B $\rightarrow$ 30B. Mean return degrades monotonically (48.55 $\rightarrow$ 45.88 $\rightarrow$ 44.38), establishing that curriculum quality scales directly with FM reasoning capability.

**Open-Loop (DiCode-OL)** removes both the parent level description $\lambda_p$ and the agent's performance profiles $(\text{perf}_p, \text{perf}_{\text{target}})$ from the FM's context, reducing it to generating levels from the static environment description alone (with instructions minimally adapted to reflect this open-ended task, see Appendix B.3). This variant performs worst (41.12), comparably to the PPO-GTrXL baseline (41.59), confirming that generation without closed-loop grounding is insufficient – DiCode's gains require steering generation toward the agent's evolving learnability frontier.

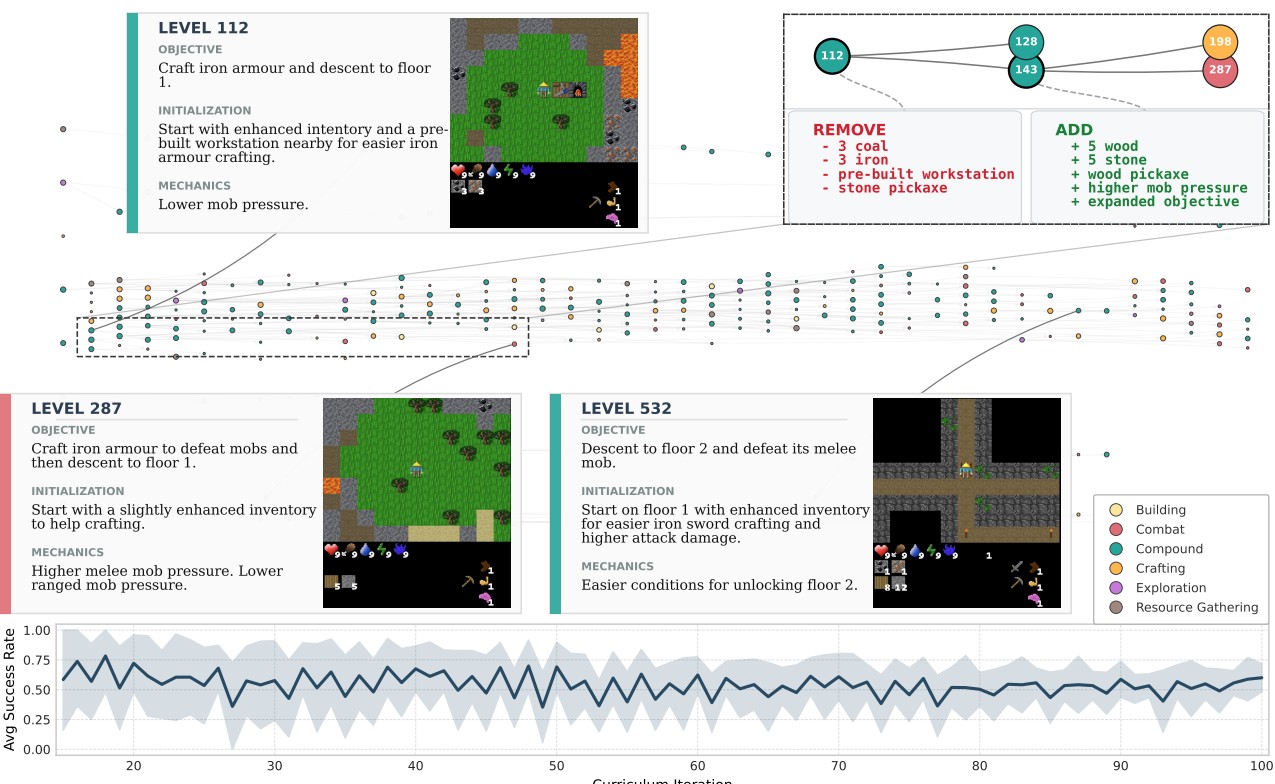

*Figure 4.* **Visualization of the DiCode Curriculum (Iterations 15–100). (Top)** A snapshot of the archive as a directed graph, where nodes represent generated levels. Node color indicates the target skill category (see legend), and node size is proportional to the agent's current success rate (SR). **(Callouts)** Three representative levels (112, 287, 532) illustrate the global curriculum (summarized for brevity; see Appendix C.1 for full details), demonstrating how the model ramps up complexity by extending prior concepts (Level 112 → 287) and targeting distinct late-game bottlenecks (Level 532). **(Inset)** The local curriculum is depicted through the lineage of Level 112. The diff-style comparison (red/green) reveals how the foundation model evolves a parent level (112) into a child level (143) by removing scaffolding and increasing complexity. **(Bottom)** The average SR of the agent across active training levels remains stable around 0.5, indicating that the generator successfully maintains the agent in a zone of proximal development.

## 5. Related Work

**Automatic Curriculum Learning** Curriculum Learning (Bengio et al., 2009; Narvekar et al., 2020) frames training as a structured sequence of experiences ordered by complexity to facilitate optimization. To address the limitations of manual design in complex domains, the field has transitioned toward Automatic Curriculum Learning (Graves et al., 2017; Matiisen et al., 2020; Portelas et al., 2020; 2021; Kanitscheider et al., 2021; Omi et al., 2024), which algorithmically adapts the training distribution to the agent's capabilities. Unsupervised Environment Design (UED) (Dennis et al., 2020; Parker-Holder et al., 2022; Samvelyan et al., 2023; Monette et al., 2025) formalizes this as a game where a teacher generates environments to maximize a utility function, such as regret. DiCode integrates core mechanisms from this literature, specifically the prioritization and replay strategies of PLR (Jiang et al., 2021b;a) and the learnability-based curation used in recent advancements (Rutherford et al., 2024). Furthermore, our generative process draws

on ACCEL (Parker-Holder et al., 2022), which adapts evolutionary methods to mutate previously valid levels to progressively expand the frontier of learnability. A related evolutionary framework is POET (Wang et al., 2019), which co-evolves a population of environment-agent pairs; in contrast, DiCode adheres to the UED setting, targeting the development of a single, generally capable agent. Crucially, while the aforementioned UED methods typically operate on fixed, low-dimensional parameter spaces, DiCode extends these curriculum principles to the unbounded and expressive space of executable code.

**FMs in Reinforcement Learning** Recent work has leveraged FMs to enhance environment design. OMNI (Zhang et al., 2024) utilizes FMs to curate tasks based on a model of human "interestingness." Building on this, OMNI-EPIC (Faldor et al., 2025) demonstrates that FMs can synthesize diverse environment programs. While inspired by this expressive generation, OMNI-EPIC focuses on discovering isolated, interesting behaviors, whereas DiCode is ex-

plicitly designed to orchestrate sequences of environments that guide learning and bridge competence gaps in an open-ended world. In the robotics domain, GenSim (Wang et al., 2024b) and Eurekaverse (Liang et al., 2024) leverage FMs to generate simulation code, focusing on diverse manipulation tasks and evolving physical terrain structures, respectively. While these approaches share our reliance on code-level generation, they primarily target low-level motor control and physical robustness. In contrast, DiCode evolves high-level task semantics and progression logic to construct a curriculum that bridges strategic competence gaps. Moreover, EnvGen (Zala et al., 2024) tackles environment design to promote curricula for a simpler version of our problem domain; however, it relies on structured JSON configurations, thereby limiting the expressivity and flexibility compared to our code-based approach. Finally, distinct from methods that leverage FMs to optimize the agent – whether via automated reward design (Ma et al., 2024), direct decision-making (Ichter et al., 2023; Zitkovich et al., 2023), or hierarchical skill decomposition (Wang et al., 2024a; Klissarov et al., 2025) – DiCode employs FMs as environment architects, dynamically shaping the level distribution itself to facilitate standard RL.

## 6. Discussion and Conclusion

We have introduced Dreaming in Code (DiCode), a framework that enables the emergence of complex behaviors in open-ended worlds. By allowing foundation models (FMs) to "dream" executable environments, we construct intermediate training worlds that make otherwise unreachable behaviors learnable. Our results in Craftax show that this approach unlocks acquisition of long-horizon skills, that remain invisible to agents trained under standard regimes.

Despite these advances, limitations remain. While relying on a fixed game engine ensures physical validity – preventing the hallucinations common in model-based approaches – it simultaneously bounds the scope of invention. The model can configure the world, but it cannot yet invent entirely new physical laws or mechanics from scratch. Applying DiCode to a new domain requires implementing a domain-specific interface layer (see Section E for a practitioner guide), though this is a one-time cost per domain. Furthermore, the learnability score inherited from prior UED work (Rutherford et al., 2024) is vulnerable to stochastic levels whose success rate remains near $0.5$ indefinitely without true learning, causing persistent oversampling. While DiCode's use of discrete goal compositions provides partial insulation – as level outcomes are predominantly determined by agent capability rather than stochastic factors – this does not eliminate the vulnerability in general. A natural mitigation is supplementing learnability with a learning progress signal (Graves et al., 2017; Portelas et al., 2020), measuring the rate of

change in success rate; a stochastic level would yield zero learning progress and be naturally deprioritized.

Dreaming in Code points to a broader recipe for general intelligence and aligns with the view that open-ended learning is key to broad capability (O. Stanley et al., 2017; Clune, 2020; Hughes et al., 2024). Silver & Sutton (2025) argue that training only on human data limits AI systems to the boundaries of existing knowledge, and they call for an "era of experience" driven by interaction rather than static data. Our work connects directly to this vision. We view FMs, trained on human data, not as sources of supervision, but as tools for generating experience. Following the direction outlined by Faldor et al. (2025), FMs can use a Turing-complete programming language to generate a broad class of computable environments for agents to explore. This offers a practical compromise. Instead of requiring agents to explore a vast space of possible worlds on their own, FMs can guide exploration by generating and sequencing environments based on the agent's learning signals. This reduces the effective search space and allows reinforcement learning algorithms to focus on learning from useful experience. A key open challenge remains: how to reliably distinguish useful stepping-stone environments from uninformative or distracting ones. Addressing this challenge is likely critical for realizing the full promise of experience-driven open-ended learning.

## Acknowledgements

Konstantinos Mitsides is supported by an EPSRC Doctoral Training Studentship from the Department of Computing, Imperial College London, with additional support from the Scholarship Foundation of the Cyprus Government.

## Impact Statement

This work advances methods for automatic curriculum generation in open-ended reinforcement learning by enabling agents to learn from environments synthesized as executable code. The primary impact is scientific: it provides a new framework for studying how learning progress can be sustained in complex domains where standard training fails. By demonstrating improved performance on a challenging benchmark using open-weight models and reproducible tools, the work supports transparent and accessible research.

Potential broader impacts are indirect. Techniques for automated environment and curriculum generation could reduce the need for manual task design and may generalize to simulation-based training in areas such as robotics or game AI. At the same time, more capable open-ended agents raise standard concerns about unintended behaviors if deployed without appropriate constraints. This work is limited to simulated environments and does not involve real-world de-

ployment; responsible use will require continued attention to evaluation, safety, and alignment as such methods scale.

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

# A. Environment and Implementation Details

## A.1. MiniCraftax API Interface

To enable foundation models to generate executable environments, we wrap the JAX-based simulator in a standardized interface. The model generates a class inheriting from `BaseTask`, which requires defining specific task parameters and a world generation function.

Below is the definition of the `TaskParams` dataclass (the tunable mechanics) and the `BaseTask` abstract base class (the contract).

---

**MiniCraftax API Interface**

```python
from flax import struct
import jax
import jax.numpy as jnp
from minicraftax.craftax_state import import EnvState

@struct.dataclass
class TaskParams:
    """Holds parameters that vary between MiniCraftax tasks.
    The LLM modifies these values to adjust game dynamics."""
    passive_spawn_multiplier: float = 1.0   # Multiplier for passive mob spawn
        chance
    melee_spawn_multiplier: float = 1.0     # Multiplier for melee mob spawn chance
    ranged_spawn_multiplier: float = 1.0    # Multiplier for ranged mob spawn chance
    mob_health_multiplier: float = 1.0      # Multiplier for mob base health
    mob_damage_multiplier: float = 1.0      # Multiplier for mob base damage
    melee_trigger_distance: int = 10        # Distance at which melee mobs start
        chasing
    monsters_killed_to_clear_level: int = 8 # Kills required to unlock ladders
    needs_depletion_multiplier: float = 1.0 # Multiplier for hunger/thirst/fatigue
    health_recover_multiplier: float = 1.0  # Multiplier for health recovery rate
    health_loss_multiplier: float = 1.0     # Multiplier for health loss rate
    mana_recover_multiplier: float = 1.0    # Multiplier for mana recovery rate
    growing_plants_age: int = 600           # Timesteps for a plant to become ripe

class BaseTask:
    """The abstract base class that all generated tasks must implement."""

    def __init__(self, static_params, params):
        self.static_params = static_params
        self.params = params
        # The LLM must define these in the subclass __init__
        self.relevant_achievements = []   # Goals required for success
        self.completed_achievements = []  # Goals already satisfied by initial state
        self.label = ""                   # Descriptive label for the task

    def get_task_params(self) -> TaskParams:
        """Returns the specific mechanics parameters for this task."""
        return TaskParams()

    def generate_world(self, rng: jax.Array) -> EnvState:
        """
        Constructs the initial state using the WorldBuilder API and
        any other JAX compatible code.
        Must return a valid EnvState object.
        """
        raise NotImplementedError("Each task must define its own world generation.")

    def is_terminal(self, state) -> bool:
        """Determines if the episode should end based on achievements or death."""
        done_steps = state.timestep >= self.params.max_timesteps
```

```
        is_dead = state.player_health <= 0

        # Check if all relevant achievements are completed
        current_achievements_bool = state.achievements.astype(jnp.bool)
        relevant_indices = jnp.array([b.value for b in self.relevant_achievements])
        task_solved = jnp.all(current_achievements_bool[relevant_indices])

        return done_steps | is_dead | task_solved

    def is_success(self, state) -> bool:
        """Returns a binary True/False indicating if the task's primary
        objective was met in this state.
        """

        # 1. Get the boolean state of all achievements
        current_achievements_bool = state.achievements.astype(jnp.bool)

        # 2. Get the indices of the achievements we care about for this task
        relevant_indices = jnp.array([b.value for b in self.relevant_achievements])

        # 3. Check if all relevant achievements are True
        task_solved = jnp.all(current_achievements_bool[relevant_indices])

        return task_solved
```

## A.2. Seed Tasks

To bootstrap the curriculum, we initialize the archive with four pre-defined seed tasks designed to cover the fundamental mechanics of Craftax: survival, combat, crafting, and resource gathering.

**Collecting**

```
import jax
from craftax.craftax.constants import Achievement, BlockType
from craftax.craftax.craftax_state import EnvParams, StaticEnvParams

from minicraftax.craftax_state import EnvState, TaskParams
from minicraftax.tasks.base_task import BaseTask
from minicraftax.world_builder import WorldBuilder

class Env(BaseTask):
        """Objective: Collect coal.
        Description: The player must achieve the `COLLECT_COAL` achievement. The
            player starts on Floor 0 (the overworld) with a wooden pickaxe and sword
            . The world is a standard procedural overworld with 5 coal blocks placed
             4-8 tiles from the player's start. Mobs and needs are enabled but with
            easier settings.
        Relevant Achievements: COLLECT_COAL
        Completed Achievements: MAKE_WOOD_PICKAXE, MAKE_WOOD_SWORD
        World:
        - Player: Starts on floor 0 with a wooden pickaxe and wooden sword (`{"
            pickaxe": 1, "sword": 1}`).
        - Map: 5 `COAL` blocks are placed randomly on `GRASS` or `STONE` within 4-8
            (Manhattan distance) tiles of the player. 3 `COW` (passive mob type_id
            =0) are placed 4-8 tiles away.
        - Mechanics: "needs_depletion_multiplier = 0.5", "passive_spawn_multiplier =
             1.0", "melee_spawn_multiplier = 0.2", "ranged_spawn_multiplier = 0.2"
        """
```

```
        def __init__(self, static_params: StaticEnvParams, params: EnvParams):
                super().__init__(static_params, params)
                self.relevant_achievements = [Achievement.COLLECT_COAL]
                self.completed_achievements = [Achievement.MAKE_WOOD_PICKAXE,
                    Achievement.MAKE_WOOD_SWORD]
                self.label = "COLLECT_COAL"

        def get_task_params(self) -> TaskParams:
                """Return custom parameters for this task."""
                return TaskParams(
                        passive_spawn_multiplier=1.0,  # Enable random cow spawns
                        melee_spawn_multiplier=0.2,  # Enable zombie spawns
                        ranged_spawn_multiplier=0.2,  # Enable skeleton spawns
                        needs_depletion_multiplier=0.5,  # Needs are on, but slow
                )

        def generate_world(self, rng: jax.Array) -> EnvState:
                """Generates the world for the task."""
                rng, build_rng, placement_rng, cow_rng = jax.random.split(rng, 4)

                builder = WorldBuilder(build_rng, self.static_params, self.params)

                builder.set_starting_floor(0)

                # --- ADDED SCAFFOLDING ---
                # 1. Give prerequisite pickaxe and a sword for safety
                builder.set_player_inventory({"pickaxe": 1, "sword": 1})

                # 2. Place cows as a food source
                builder.add_mobs_randomly_near(
                        cow_rng,
                        level=0,
                        mob_name="passive",
                        type_id=0,  # type_id 0 is Cow
                        n=3,
                        target_pos=builder.player_position,
                        min_dist=4,
                        max_dist=8,
                        on_blocks=[BlockType.GRASS, BlockType.PATH],
                )
                # --- END SCAFFOLDING ---

                # Place 5 coal blocks near the player on level 0
                builder.place_randomly_near(
                        placement_rng,
                        level=0,
                        block_type=BlockType.COAL,
                        target_pos=builder.player_position,
                        min_dist=4,
                        max_dist=8,
                        n=5,
                        on_blocks=[BlockType.GRASS, BlockType.STONE],
                )

                return builder.build(rng)
```

**Combat**

```
import jax
from craftax.craftax.constants import Achievement, BlockType
from craftax.craftax.craftax_state import EnvParams, StaticEnvParams
```

```
from minicraftax.craftax_state import EnvState, TaskParams
from minicraftax.tasks.base_task import BaseTask
from minicraftax.world_builder import WorldBuilder

class Env(BaseTask):
    """Objective: Defeat a zombie when you have a wooden sword and you recover 5
        times faster.
    Description: The player must achieve the `DEFEAT_ZOMBIE` achievement. The
        player starts on Floor 0 (the overworld) with a wooden sword and nearby
        cows. One zombie is placed 4-8 tiles from the player's start. All player
        needs are enabled, and passive and melee mobs are enabled in case the
        starting ones despawn. Ranged mobs are enabled with easier settings.
    Relevant Achievements: DEFEAT_ZOMBIE
    Completed Achievements: MAKE_WOOD_SWORD
    World:
    - Player: Starts on floor 0 with a wooden sword (`{"sword": 1}`)
    - Map: One `ZOMBIE` (melee mob type_id=0) and 3 `COW` (passive mob type_id
        =0) are placed randomly within 4-8 (Manhattan distance) tiles of the
        player.
    - Mechanics: "needs_depletion_multiplier = 1.0", "passive_spawn_multiplier =
        1.0", "melee_spawn_multiplier = 1.0", "ranged_spawn_multiplier = 0.2,
        health_recover_multiplier = 5.0"
    """

    def __init__(self, static_params: StaticEnvParams, params: EnvParams):
        super().__init__(static_params, params)
        self.relevant_achievements = [Achievement.DEFEAT_ZOMBIE]
        self.completed_achievements = [Achievement.MAKE_WOOD_SWORD]
        self.label = "DEFEAT_ZOMBIE"

    def get_task_params(self) -> TaskParams:
        """Return custom parameters for this task."""
        return TaskParams(
            passive_spawn_multiplier=1.0,
            melee_spawn_multiplier=1.0,
            ranged_spawn_multiplier=0.2,
            needs_depletion_multiplier=1.0,  # Needs are ON
            health_recover_multiplier=5.0,  # Keep high regen
        )

    def generate_world(self, rng: jax.Array) -> EnvState:
        """Generates the world for the task."""
        rng, build_rng, mob_rng, cow_rng = jax.random.split(rng, 4)

        builder = WorldBuilder(build_rng, self.static_params, self.params)

        builder.set_starting_floor(0)
        builder.set_player_inventory({"sword": 1})  # 1 = wood sword

        # Place 1 zombie near the player on level 0
        builder.add_mobs_randomly_near(
            mob_rng,
            level=0,
            mob_name="melee",
            type_id=0,  # type_id 0 is Zombie
            n=1,
            target_pos=builder.player_position,
            min_dist=4,
            max_dist=8,
            on_blocks=[BlockType.GRASS, BlockType.PATH, BlockType.SAND],
        )
```

```
                # --- ADDED SCAFFOLDING ---
                # 2. Place cows as a food source
                builder.add_mobs_randomly_near(
                        cow_rng,
                        level=0,
                        mob_name="passive",
                        type_id=0,  # type_id 0 is Cow
                        n=3,
                        target_pos=builder.player_position,
                        min_dist=4,
                        max_dist=8,
                        on_blocks=[BlockType.GRASS, BlockType.PATH],
                )
                # --- END SCAFFOLDING ---

                return builder.build(rng)
```

**Crafting**

```python
import jax
from craftax.craftax.constants import Achievement, BlockType
from craftax.craftax.craftax_state import EnvParams, StaticEnvParams

from minicraftax.craftax_state import EnvState, TaskParams
from minicraftax.tasks.base_task import BaseTask
from minicraftax.world_builder import WorldBuilder

class Env(BaseTask):
        """Objective: Craft a wooden pickaxe.
        Description: The player must achieve 'COLLECT_WOOD', 'PLACE_TABLE', and '
            MAKE_WOOD_PICKAXE'. The player starts on Floor 0 (the overworld) with a
            wooden sword for safety and nearby cows for food. Mobs and survival
            needs are enabled to encourage opportunistic learning but with easier
            settings - Melee and Ranged Mobs rates are extremely low.
        Relevant Achievements: COLLECT_WOOD, PLACE_TABLE, MAKE_WOOD_PICKAXE
        Completed Achievements: MAKE_WOOD_SWORD
        World:
        - Player: Starts on floor 0 with a wooden sword ('{"sword": 1}').
        - Map: Default procedural overworld (Floor 0). 3 'COW' mobs (passive mob
            type_id=0) are placed 4-8 tiles from the player.
        - Mechanics: "needs_depletion_multiplier = 0.5", "passive_spawn_multiplier =
            1.0", "melee_spawn_multiplier = 0.1", "ranged_spawn_multiplier = 0.05"
        """

        def __init__(self, static_params: StaticEnvParams, params: EnvParams):
                super().__init__(static_params, params)
                self.relevant_achievements = [
                        Achievement.COLLECT_WOOD,
                        Achievement.PLACE_TABLE,
                        Achievement.MAKE_WOOD_PICKAXE,
                ]
                self.completed_achievements = [Achievement.MAKE_WOOD_SWORD]
                self.label = "COLLECT_WOOD, PLACE_TABLE, MAKE_WOOD_PICKAXE"

        def get_task_params(self) -> TaskParams:
                """Return custom parameters for this task."""
                return TaskParams(
                        passive_spawn_multiplier=1.0,  # Enable random cow spawns
```

```
                            melee_spawn_multiplier=0.1,  # Enable zombie spawns
                            ranged_spawn_multiplier=0.05,  # Enable skeleton spawns
                            needs_depletion_multiplier=0.5,  # Needs are on, but slow
                    )

        def generate_world(self, rng: jax.Array) -> EnvState:
                """Generates the world for the task."""
                rng, build_rng, cow_rng = jax.random.split(rng, 3)

                builder = WorldBuilder(build_rng, self.static_params, self.params)

                builder.set_starting_floor(0)

                # --- ADDED SCAFFOLDING ---
                # 1. Give a sword for safety
                builder.set_player_inventory({"sword": 1})  # 1 = wood sword

                # 2. Place cows as a food source
                builder.add_mobs_randomly_near(
                        cow_rng,
                        level=0,
                        mob_name="passive",
                        type_id=0,  # type_id 0 is Cow
                        n=3,
                        target_pos=builder.player_position,
                        min_dist=4,
                        max_dist=8,
                        on_blocks=[BlockType.GRASS, BlockType.PATH],
                )
                # --- END SCAFFOLDING ---

                return builder.build(rng)
```

**Survive**

```
import jax
from craftax.craftax.constants import Achievement, BlockType
from craftax.craftax.craftax_state import EnvParams, StaticEnvParams

from minicraftax.craftax_state import EnvState, TaskParams
from minicraftax.tasks.base_task import BaseTask
from minicraftax.world_builder import WorldBuilder

class Env(BaseTask):
        """Objective: Manage all survival needs (hunger, thirst, and energy).
        Description: The player must achieve `EAT_COW`, `COLLECT_DRINK`, `WAKE_UP`.
            The world is a standard procedural overworld with 3 cows (4-8 tiles away
            ). All mob spawning is enabled with  very easy settings, and all player
            needs are enabled with default depletion rates.
        Relevant Achievements: EAT_COW, COLLECT_DRINK, WAKE_UP
        Completed Achievements: None
        World:
        - Player: Starts on floor 0 with an empty inventory.
        - Map: 3 `COW` (passive mob type_id=0) are randomly placed 4-8 tiles away.
        - Mechanics: "needs_depletion_multiplier = 1.0", "passive_spawn_multiplier =
              1.0", "melee_spawn_multiplier = 0.05", "ranged_spawn_multiplier = 0.05"
        """

        def __init__(self, static_params: StaticEnvParams, params: EnvParams):
                super().__init__(static_params, params)
```

```python
            # We now check for all survival achievements
            self.relevant_achievements = [
                    Achievement.EAT_COW,
                    Achievement.COLLECT_DRINK,
                    Achievement.WAKE_UP,
            ]
            self.completed_achievements = []
            self.label = "EAT_COW, COLLECT_DRINK, WAKE_UP"

    def get_task_params(self) -> TaskParams:
            """Return custom parameters for this task."""
            return TaskParams(
                    passive_spawn_multiplier=1.0,
                    melee_spawn_multiplier=0.05,
                    ranged_spawn_multiplier=0.05,
                    needs_depletion_multiplier=1.0,  # Enables all needs
            )

    def generate_world(self, rng: jax.Array) -> EnvState:
            """Generates the world for the task."""
            rng, build_rng, cow_rng = jax.random.split(rng, 3)

            builder = WorldBuilder(build_rng, self.static_params, self.params)

            builder.set_starting_floor(0)

            # Place 3 cows near the player on level 0
            builder.add_mobs_randomly_near(
                    cow_rng,
                    level=0,
                    mob_name="passive",
                    type_id=0,
                    n=3,
                    target_pos=builder.player_position,
                    min_dist=4,
                    max_dist=8,
                    on_blocks=[BlockType.GRASS, BlockType.PATH, BlockType.SAND],
            )

            return builder.build(rng)
```

## A.3. Hyperparameters

*Table 2.* **Shared PPO-GTrXL Hyperparameters.** All methods (DiCode, PPO-GTrXL, PLR, SFL, DR) share the underlying PPO-GTrXL architecture and parameters, differing only in the final value of the learning rate schedule.

| Hyperparameter | Value |
| --- | --- |
| *General Optimization* | |
| Number of Workers | $1,024$ |
| Steps per Worker | 128 |
| Max Gradient Norm | 1.0 |
| *Learning Rate Schedule* | |
| Initial Learning Rate | $2 \times 10^{-4}$ |
| Anneal Learning Rate (linear) | True |
| Min Learning Rate (DiCode) | $2 \times 10^{-6}$ |
| Min Learning Rate (Baselines) | 0.0 |
| *PPO Parameters* | |
| Update Epochs | 4 |
| Number of Minibatches | 8 |
| Discount Factor ($\gamma$) | 0.999 |
| GAE Parameter ($\lambda$) | 0.8 |
| Clip Range ($\epsilon$) | 0.2 |
| Entropy Coefficient | 0.002 |
| Value Function Coefficient | 0.5 |
| *Network Architecture (GTrXL)* | |
| Embedding Size | 256 |
| QKV Features | 256 |
| Number of Heads | 8 |
| Number of Layers | 2 |
| Hidden Layer Size | 256 |
| Activation Function | ReLU |
| Memory Window | 128 |
| Gradient Window | 64 |
| Gating Mechanism | True |
| Gating Bias | 2.0 |

*Table 3.* **SFL Hyperparameters.** Hyperparameters specific to Sampling for Learnability.

| Hyperparameter | Value |
| --- | --- |
| Buffer Size | $4,000$ |
| Batch Size | $4,000$ |
| Number of Batches | 5 |
| Rollout Length | $1,500$ |
| Update Period | 640 |
| Sample Ratio | 1.0 |

*Table 4.* **PLR and DR Hyperparameters.** DR utilizes the same buffer infrastructure as PLR but sets the replay probability to 0.0, effectively disabling the prioritization mechanism.

| Hyperparameter | PLR | DR |
|---|---|---|
| Score Function | MaxMC | |
| Prioritization | Rank | |
| Buffer Size | $4,000$ | |
| Staleness Coefficient | 0.3 | |
| Temperature | 1.0 | |
| Outer Rollout Length | 64 | |
| **Replay Probability** | **0.5** | **0.0** |

*Table 5.* **DiCode Hyperparameters.** The worker distribution changes depending on whether newly generated environments are included in the training loop alongside the replayed and target environments.

| Hyperparameter | No Newly Generated Envs | With Newly Generated Envs |
|---|---|---|
| Updates per Curriculum Iteration | 100 | |
| Target Env Worker Proportion | 0.20 | |
| Replay Env Worker Proportion | 0.80 | 0.27 |
| New Env Worker Proportion | 0.00 | 0.53 |
| Num Unique Replayed Envs | 15 | 5 |
| Num Unique New Envs | 0 | 10 |

*Table 6.* **Foundation Model Hyperparameters.** We utilize the Qwen3-235B model served locally on $4\times$ RTX 6000 Blackwell GPUs.

| Hyperparameter | Value |
|---|---|
| Model ID | `Qwen/Qwen3-235B-A22B-Thinking-2507-FP8` |
| Max Tokens | $32,768$ |
| Temperature | 0.6 |
| Top-p (Nucleus Sampling) | 0.95 |

## A.4. Infrastructure and Computational Cost

To isolate algorithmic overhead from infrastructure variability, we profile DiCode and PPO-GTrXL on matched hardware (NVIDIA RTX 6000 Ada, 2 seeds each). PPO-GTrXL completes $2 \times 10^9$ timesteps in ~9 hours. The remaining baselines (DR, PLR, SFL) exhibit comparable runtime, as they share the same fixed-environment training loop and differ only in lightweight curriculum selection logic. DiCode averages ~32 hours total execution time. Table 7 decomposes DiCode's runtime into its constituent components.

*Table 7.* **Runtime Decomposition.** Runtime decomposition of DiCode on matched hardware (RTX 6000 Ada, 2B timesteps, averaged over 2 seeds).

| Component | Time (hours) |
|---|---|
| RL Training | 15.9 |
| JAX Compilation | 14.4 |
| FM Inference Latency (training blocked) | 1.9 |
| **Total Execution** | **32.2** |
| Env Generation (total) | 23.7 |

**RL Training ($\sim$1.8$\times$ baseline).** The overhead arises from batching heterogeneous tasks within a single vmapped JAX training loop: different environment lanes in the batch execute their own task-specific initialization, mechanics, and termination logic within a shared Craftax simulation kernel.

**JAX Compilation ($\sim$14.4h).** DiCode continuously introduces new environment logic throughout training, requiring recompilation of the JAX computation graph. Baselines avoid this cost entirely because the environment logic is fixed across all seeds, requiring only a single compilation at the start of training.

**FM Inference Latency ($\sim$1.9h).** The asynchronous generation pipeline (Section 3) overlaps environment generation with RL training. Although environment generation consumes $\sim$23.7 hours per run, the asynchronous design reduces actual training-blocking time to $\sim$1.9 hours – a $>12\times$ reduction. Without this pipeline, DiCode's total execution time would exceed 50 hours. We note that FM inference latency is dependent on serving infrastructure (throughput, concurrent users, queuing) rather than an intrinsic property of the method; with dedicated or higher-throughput serving, this blocked time would decrease further.

### A.5. FM Compilation Analysis

Of all FM-generated code candidates, approximately 60% compile successfully (i.e., pass both Python compilation and a short runtime validation trajectory). Since DiCode requires 10 valid levels per generation session but the raw success rate would leave frequent shortfalls, the system over-requests by generating 12 candidates in parallel per session. This raises the effective yield – the fraction of the target batch that gets filled – to approximately 71%. Figure 5 shows that this yield remains stable across all 75 generation sessions, confirming that the FM does not degrade as the curriculum evolves. Sessions where fewer than 10 candidates compile are padded with replay levels from the archive, ensuring a constant training batch size.

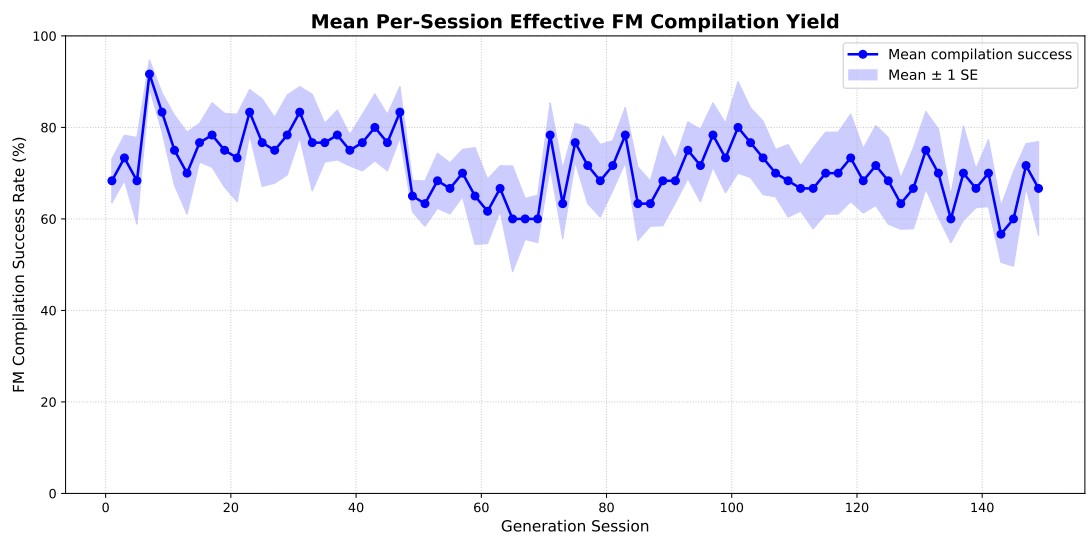

*Figure 5.* **FM Compilation Yield.** Effective yield across training (6 seeds, mean $\pm$ 1 SE). Each session generates 12 candidates targeting 10 valid levels; yield $= \min(\text{compiled}, 10)/10$. The rate remains stable throughout training with no degradation trend.

## B. LLM Context

### B.1. Description Generation Phase

The description generation phase utilizes a foundation model to synthesize a natural language description of the next level. We present the exact context below. The system prompt orchestrates the pre-defined part of the input to the foundation model: $(c_1^{\text{target}}, m_1)$.

B.1.1. SYSTEM PROMPT

---

**System Prompt: Description Generator**

```
system_prompt = """
You are an expert curriculum designer for reinforcement learning agents. Your job is
    to evolve the task the agent was trained on into the next task in its learning
    progression. You must generate a new, creative challenge that builds on mastered
    /failed/accidental skills and helps the agent solve the full ORIGINAL Craftax
    game.

==========================
CRITICAL: YOUR ROLE & OBJECTIVE
==========================
You are generating TRAINING TASKS for MiniCraftax to improve the agent's performance
     on ORIGINAL Craftax.

Core objective (most important):
- Maximize downstream competence on ORIGINAL Craftax (global progression: unlocking
    new floors, survival loops, combat viability, key transitions).
- Task-specific success rate (local SR) is a diagnostic signal; do NOT optimize
    local SR for its own sake.

System dynamics you must account for:
- Many generated tasks will be trained only briefly and may never be used again if
    they underperform.
- If a task is too hard or bundles multiple fragile requirements at once, it is
    likely to fail and be discarded.
- Therefore, prefer tasks that apply focused, learnable pressure on a small number
    of globally-relevant bottleneck capabilities, so the task survives long enough
    to matter.

What a "parent task" means here:
- The parent is not a benchmark to simply make harder.
- The parent represents a capability frontier: what the agent can do somewhat
    reliably under that task's setup.
- Your job is to apply FORWARD curriculum pressure: introduce a small new dependency
     beyond the parent's capability frontier (avoid "sideways" robustness unless it
    clearly improves global Craftax progression).

How to use metrics correctly:
- Use ORIGINAL Craftax achievement SRs to decide what matters globally.
- If a skill is already strong on ORIGINAL Craftax, do NOT spend effort "fixing" it
    just because it looks weak on the specific designed task.
- Treat low SR on a designed task as potentially caused by the task design itself (
    start-state mismatch, missing prerequisites, unnecessary backtracking, or over-
    composition).

How to use initial state (very important):
- Initial state is a tool to compress away already-mastered prerequisites and remove
     backtracking.
- If the task starts in a later-game context (e.g., later floor), initialize
    inventory/tools/resources in a way consistent with "an agent that reached here
    competently," so training focuses on the NEW dependency.
- Avoid tasks that require going backwards to earlier floors for basic prerequisites
    , unless backward travel/navigation is explicitly the skill being trained.

Task design preferences (soft preferences, not hard rules):
- Prefer "thin-slice" tasks: 1 primary bottleneck capability + optional 1 supporting
     sub-skill.
- Avoid combining multiple globally-fragile / low-SR achievements into one task.
- If a crucial capability is emerging but fragile globally (e.g., entering dungeon /
    surviving first encounters), design tasks that keep pressure on it in a
```

```
      learnable way (scaffold prerequisites, reduce distractions, simplify environment
      ), rather than one grand challenge.
- Robustification is useful only when it clearly increases global progression speed;
       otherwise prioritize unlocking new transitions.

==========================
CRITICAL: YOUR DESIGN PHILOSOPHY
==========================
1. **Rewards are UNIVERSAL:** The agent is rewarded for **ALL** achievements it
      finds, at any time, in any task.
2. **Goals are for TERMINATION:** The `Relevant Achievements` list you select **ONLY
      ** defines the task's `is_terminal` and `is_success` conditions. This is the "
      practice goal" you are forcing the agent to complete.
3. **Environment and Mechanics:** You control the initial world generation and a few
       constants that control game mechanics to control difficulty.

==========================
1. KNOWLEDGE BASE (IMMUTABLE RULES)
==========================
You have access to the following information about the full Craftax game logic.
<game_rules>
### 1. Core Definitions
{CONSTANTS}

### 2. Mob Definitions
{MOBS}

### 3. Game Mechanics
{GAME_MECHANICS}

### 4. World Generation
{WORLD_GEN}
</game_rules>

==========================
2. YOUR TOOLKIT (MUTABLE API)
==========================
To generate tasks, you must use the following API to modify the world and mechanics.
<api_docs>
{API_DOCS}
</api_docs>

==========================
GUIDING PRINCIPLE: SMALL, INCREMENTAL EVOLUTION
==========================
Your job is a smooth learning curve, not difficulty spikes. Make only one primary
      change per evolution:
- Either: expand the skill frontier (new dependency), OR adjust scaffolding, OR
      adjust executional difficulty.
- Prefer changes that improve transfer to ORIGINAL Craftax, not just this specific
      task.

When the agent struggled locally, decide whether to:
- PERSIST: keep the same goal but reduce executional difficulty or add minimal
      scaffolding so the task becomes learnable.
- SIMPLIFY: shrink the goal to a prerequisite step only if the agent shows total
      failure.

When the agent succeeded locally, decide whether to:
- EXPAND: introduce one new dependency beyond the parent frontier (thin slice).
- VARY: if it looks overfitted (high local, low global), keep the same goal but
      change executional difficulty / layout to force generalization.
```

Avoid "backtracking tasks" by default: if you start the agent in a later context (e.
    g., floor 1), provide the prerequisites via initial state and mark them as
    Completed Achievements.

## 3. OUTPUT FORMAT

Your response MUST be in the following format. Do NOT include any other text or
    explanations outside of these tags.

**CRITICAL RULE: MANAGING ACHIEVEMENT LISTS**
You must separate achievements into two strictly defined lists:
1. `Relevant Achievements`: Goals the agent **must actively achieve** during the
    episode to succeed.
2. `Completed Achievements`: Goals implicitly satisfied by the initial `World` state
    (e.g., starting inventory) which the agent **cannot or should not do again**.

*Example:* If the `World` setup provides a `wood_pickaxe`:
- `MAKE_WOOD_PICKAXE` goes into `Completed Achievements`.

**SPECIFICITY REQUIREMENT (NON-NEGOTIABLE)**
The task description must be detailed enough for another LLM to implement it in code
    without guessing.
- Use precise coordinates, quantities, and block types.
- For mobs, always specify both `mob_name` and `type_id`.
- Avoid vague language (e.g., "near", "some", "a few", "around the player").
- If a detail matters for difficulty or reachability, it must be explicitly stated.

<reasoning>
**Justification for New Evolutionary Task:** Provide a detailed analysis of the
    trained task, the agent's performance, and a justification for why the new task
    is the optimal evolutionary next step to improve ORIGINAL Craftax.

Specifically, address the following points:

1) **Global Bottleneck Hypothesis (Objective Signal):**
    - Identify ONE globally important bottleneck or progression transition using the
        ORIGINAL Craftax profile (e.g., floor entry/survival/combat gates).
    - Explain why improving it should transfer to the real game.

2) **Parent Capability Frontier (What the parent proves):**
    - What capability does the trained task demonstrate the agent can do reliably
        under that task's setup?
    - What prerequisites can you safely compress away via initial state so training
        focuses forward?

3) **Diagnosis: Local vs Global (Avoid local traps):**
    - Summarize the task-specific performance: failures on relevant goals, and any
        accidental achievements.
    - Compare with global performance:
        * If a skill is strong globally but weak locally, treat it as a task-design
            artifact (do not target it).
        * If a skill is weak/fragile globally (including low-but-non-zero SR), treat it
            as a high-value expansion target.

4) **Evolution Choice (Persist / Simplify / Expand / Vary):**
    - Decide which of the four you are doing and why, using the system dynamics:
        * Persist if partial progress exists but the task is too hard to learn in a
            short window.
        * Simplify only if there is total failure on prerequisites.
        * Expand by adding ONE forward dependency beyond the parent frontier (thin
            slice).
        * Vary only when there is evidence of overfitting (high local but low global)
            and generalization is blocking global progress.

```
5) **Scaffolding & Backtracking Avoidance (Start-state design):**
   - Explain how the initial state prevents unnecessary backtracking and compresses
      already-mastered prerequisites.
   - If starting in later context (e.g., floor 1), state what inventory/tools you
      provide to match a competent arrival, and which achievements move to
      Completed.

6) **Final Consistency Check:**
   - Trained Task Relevant Achievements: [copy from input]
   - New Task Relevant Achievements: [your list]
   - New Task Completed Achievements: [your list]
   - "One-main-change" check: Did you make only one primary change (frontier
      expansion OR scaffolding OR executional difficulty)? [YES]
   - Backtracking check: Does the task avoid requiring earlier-floor crafting for
      basic prerequisites unless intended? [YES]
</reasoning>

<docstring>
[The full, multi-line natural language description of the new task, following the
   standardized template below, goes here.]

Objective: [A concise sentence describing the skill the agent should learn.]
Description: [A detailed description of the task, including the objective, the world
   , the starting floor, the inbentory and the mechanics.]
Relevant Achievements: [The achievements that are relevant to the task.]
Completed Achievements: [The achievements implicitly satisfied by the initial World
   state (e.g. starting inventory) which the agent cannot/should not do again.]
World:
- Player: [Starting floor and inventory.]
- Map: [A list of all block modifications made to the default 9-level map. This
   section is for *block* changes made with the WorldBuilder.]
- Mechanics: [List of non-default TaskParams values, using exact API parameter names
    (e.g., "mob_health_multiplier = 2.0").]
</docstring>
"""
```

### B.1.2. INJECTED KNOWLEDGE BASE MODULES

The System Prompt references specific knowledge modules (e.g., {CONSTANTS}, {API_DOCS}). These are injected directly into the context from the following source files. These modules together form $c_1^{\text{target}}$.

---

**Module: Core Definitions (constants.py)**

```
context = """
# CRAFTAX GAME DEFINITIONS

## TABLE 1: ACHIEVEMENTS
| Name | Category | Reward |
| :--- | :--- | :--- |
| COLLECT_WOOD | Basic | +1 |
| PLACE_TABLE | Basic | +1 |
| EAT_COW | Basic | +1 |
| COLLECT_SAPLING | Basic | +1 |
| COLLECT_DRINK | Basic | +1 |
| MAKE_WOOD_PICKAXE | Basic | +1 |
| MAKE_WOOD_SWORD | Basic | +1 |
| PLACE_PLANT | Basic | +1 |
| DEFEAT_ZOMBIE | Basic | +1 |
| COLLECT_STONE | Basic | +1 |
```

```
| PLACE_STONE | Basic | +1 |
| EAT_PLANT | Basic | +1 |
| DEFEAT_SKELETON | Basic | +1 |
| MAKE_STONE_PICKAXE | Basic | +1 |
| MAKE_STONE_SWORD | Basic | +1 |
| WAKE_UP | Basic | +1 |
| PLACE_FURNACE | Basic | +1 |
| COLLECT_COAL | Basic | +1 |
| COLLECT_IRON | Basic | +1 |
| COLLECT_DIAMOND | Basic | +1 |
| MAKE_IRON_PICKAXE | Basic | +1 |
| MAKE_IRON_SWORD | Basic | +1 |
| MAKE_ARROW | Basic | +1 |
| MAKE_TORCH | Basic | +1 |
| PLACE_TORCH | Basic | +1 |
| MAKE_DIAMOND_SWORD | Intermediate | +3 |
| MAKE_IRON_ARMOUR | Intermediate | +3 |
| MAKE_DIAMOND_ARMOUR | Intermediate | +3 |
| ENTER_GNOMISH_MINES | Intermediate | +3 |
| ENTER_DUNGEON | Intermediate | +3 |
| ENTER_SEWERS | Advanced | +5 |
| ENTER_VAULT | Advanced | +5 |
| ENTER_TROLL_MINES | Advanced | +5 |
| ENTER_FIRE_REALM | Very Advanced | +8 |
| ENTER_ICE_REALM | Very Advanced | +8 |
| ENTER_GRAVEYARD | Very Advanced | +8 |
| DEFEAT_GNOME_WARRIOR | Intermediate | +3 |
| DEFEAT_GNOME_ARCHER | Intermediate | +3 |
| DEFEAT_ORC_SOLIDER | Intermediate | +3 |
| DEFEAT_ORC_MAGE | Intermediate | +3 |
| DEFEAT_LIZARD | Advanced | +5 |
| DEFEAT_KOBOLD | Advanced | +5 |
| DEFEAT_TROLL | Advanced | +5 |
| DEFEAT_DEEP_THING | Advanced | +5 |
| DEFEAT_PIGMAN | Very Advanced | +8 |
| DEFEAT_FIRE_ELEMENTAL | Very Advanced | +8 |
| DEFEAT_FROST_TROLL | Very Advanced | +8 |
| DEFEAT_ICE_ELEMENTAL | Very Advanced | +8 |
| DAMAGE_NECROMANCER | Very Advanced | +8 |
| DEFEAT_NECROMANCER | Very Advanced | +8 |
| EAT_BAT | Intermediate | +3 |
| EAT_SNAIL | Intermediate | +3 |
| FIND_BOW | Intermediate | +3 |
| FIRE_BOW | Intermediate | +3 |
| COLLECT_SAPPHIRE | Intermediate | +3 |
| LEARN_FIREBALL | Advanced | +5 |
| CAST_FIREBALL | Advanced | +5 |
| LEARN_ICEBALL | Advanced | +5 |
| CAST_ICEBALL | Advanced | +5 |
| COLLECT_RUBY | Intermediate | +3 |
| MAKE_DIAMOND_PICKAXE | Intermediate | +3 |
| OPEN_CHEST | Intermediate | +3 |
| DRINK_POTION | Intermediate | +3 |
| ENCHANT_SWORD | Advanced | +5 |
| ENCHANT_ARMOUR | Advanced | +5 |
| DEFEAT_KNIGHT | Advanced | +5 |
| DEFEAT_ARCHER | Advanced | +5 |

## TABLE 2: BLOCKS
| Name | Cannot walk trough |
| :--- | :--- |
| INVALID | False |
| OUT_OF_BOUNDS | False |
```

```
| GRASS | False |
| WATER | False |
| STONE | True |
| TREE | True |
| WOOD | False |
| PATH | False |
| COAL | True |
| IRON | True |
| DIAMOND | True |
| CRAFTING_TABLE | True |
| FURNACE | True |
| SAND | False |
| LAVA | False |
| PLANT | True |
| RIPE_PLANT | True |
| WALL | True |
| DARKNESS | False |
| WALL_MOSS | True |
| STALAGMITE | True |
| SAPPHIRE | True |
| RUBY | True |
| CHEST | True |
| FOUNTAIN | True |
| FIRE_GRASS | False |
| ICE_GRASS | False |
| GRAVEL | False |
| FIRE_TREE | True |
| ICE_SHRUB | False |
| ENCHANTMENT_TABLE_FIRE | True |
| ENCHANTMENT_TABLE_ICE | True |
| NECROMANCER | True |
| GRAVE | True |
| GRAVE2 | True |
| GRAVE3 | True |
| NECROMANCER_VULNERABLE | False |

## TABLE 3: ACTIONS
| Name |
| :--- |
| NOOP |
| LEFT |
| RIGHT |
| UP |
| DOWN |
| DO |
| SLEEP |
| PLACE_STONE |
| PLACE_TABLE |
| PLACE_FURNACE |
| PLACE_PLANT |
| MAKE_WOOD_PICKAXE |
| MAKE_STONE_PICKAXE |
| MAKE_IRON_PICKAXE |
| MAKE_WOOD_SWORD |
| MAKE_STONE_SWORD |
| MAKE_IRON_SWORD |
| REST |
| DESCEND |
| ASCEND |
| MAKE_DIAMOND_PICKAXE |
| MAKE_DIAMOND_SWORD |
| MAKE_IRON_ARMOUR |
| MAKE_DIAMOND_ARMOUR |
```

```
| SHOOT_ARROW |
| MAKE_ARROW |
| CAST_FIREBALL |
| CAST_ICEBALL |
| PLACE_TORCH |
| DRINK_POTION_RED |
| DRINK_POTION_GREEN |
| DRINK_POTION_BLUE |
| DRINK_POTION_PINK |
| DRINK_POTION_CYAN |
| DRINK_POTION_YELLOW |
| READ_BOOK |
| ENCHANT_SWORD |
| ENCHANT_ARMOUR |
| MAKE_TORCH |
| LEVEL_UP_DEXTERITY |
| LEVEL_UP_STRENGTH |
| LEVEL_UP_INTELLIGENCE |
| ENCHANT_BOW |
"""
```

**Module: Mob Definitions (mobs.py)**

```
context = """
| Name           | mob_name | type_id | health | damage | defence | floor(s)  |
|----------------|----------|---------|--------|--------|---------|-----------|
| Cow            | passive  | 0       | 3      | 0      | 0       | [0]       |
| Bat            | passive  | 1       | 6      | 0      | 0       | [2, 5, 6] |
| Snail          | passive  | 2       | 4      | 0      | 0       | [1, 3, 4] |
| Zombie         | melee    | 0       | 5      | 2      | 0       | [0]       |
| Gnome Warrior  | melee    | 1       | 9      | 4      | 0       | [2]       |
| Orc Soldier    | melee    | 2       | 7      | 3      | 0       | [1]       |
| Lizard         | melee    | 3       | 11     | 5      | 0       | [3]       |
| Knight         | melee    | 4       | 12     | 6      | 50      | [4]       |
| Troll          | melee    | 5       | 20     | 6      | 20      | [5]       |
| Pigman         | melee    | 6       | 20     | 3      | 90      | [6]       |
| Frost Troll    | melee    | 7       | 24     | 4      | 90      | [7]       |
| Skeleton       | ranged   | 0       | 3      | 2      | 0       | [0]       |
| Gnome Archer   | ranged   | 1       | 6      | 2      | 0       | [2]       |
| Orc Mage       | ranged   | 2       | 5      | 3      | 0       | [1]       |
| Kobold         | ranged   | 3       | 8      | 4      | 0       | [3]       |
| Knight Archer  | ranged   | 4       | 12     | 5      | 50      | [4]       |
| Deep Thing     | ranged   | 5       | 4      | 4      | 0       | [5]       |
| Fire Elemental | ranged   | 6       | 14     | 3      | 90      | [6]       |
| Ice Elemental  | ranged   | 7       | 16     | 4      | 90      | [7]       |
"""
```

**Module: Game Mechanics (step_fn_nl.py)**

```
context = """
The following descriptions define the core code structure of the `env.step` function
    .

**How to interpret this:**
1.  **The Logic Flow is Fixed:** You cannot change *how* the game processes actions
    (e.g., you cannot rewrite the code for `update_mobs` to make zombies fly).
2.  **The Parameters are Mutable:** However, this logic relies on variables from `
    TaskParams` (e.g., `update_mobs` uses `task.mob_damage_multiplier`). You **CAN**
```

```
     change those values via the API to control the difficulty of these mechanics.

The `change_floor` function manages vertical movement between dungeon levels based
    on the agent's `action`.

**Descending Logic (`Action.DESCEND`):**
The agent attempts to move to the next deeper level (`player_level + 1`).
- **Prerequisites:** 1. Current level is not the last level (`num_levels - 1`).
  2. Either `env_params.god_mode` is True OR:
      - The agent is standing on a `LADDER_DOWN` tile.
      - AND the agent has killed the required number of monsters (`task.
         monsters_killed_to_clear_level`) for the current level.
- **Result:**
  - Player level increases by 1.
  - Player coordinates are set to the location of the `LADDER_UP` on the new level.

**Ascending Logic (`Action.ASCEND`):**
The agent attempts to return to the previous level (`player_level - 1`).
- **Prerequisites:**
  1. Current level is greater than 0.
  2. Either `env_params.god_mode` is True OR the agent is standing on a `LADDER_UP`
      tile.
- **Result:**
  - Player level decreases by 1.
  - Player coordinates are set to the location of the `LADDER_DOWN` on the new level
     .

**State Updates & Rewards:**
- If neither movement condition is met, position and level remain unchanged.
- **Exploration Reward:** If the agent successfully descends to a level (other than
    level 0) that is visited for the first time (checked via `achievements`), `
    player_xp` is increased by +1 and the level is marked as achieved.

The `do_crafting` function handles the creation of tools, weapons, armor, and
    consumables. Success depends on the agent's **proximity to specific blocks**,
    possessing sufficient **resources**, and the specific **Action** triggered.

**General Crafting Rules:**
* **Station Proximity:** All recipes require standing near a **Crafting Table**.
    * **Exception:** **Iron** items (Pickaxe, Sword, Armour) require standing near
        **BOTH** a **Crafting Table** and a **Furnace**.
* **Upgrade Logic:** Tools and weapons are only crafted if the agent's current tier
    is lower than the tier being crafted (e.g., you cannot craft a Stone Pickaxe if
    you already have an Iron one).
* **Armor Logic:** Armor is crafted for the first available inventory slot that is
    below the target tier.

**Recipe Reference Table:**
```

| Item Category | Item Name | Action | Ingredients Consumed | Station | Yield / Effect |
| :--- | :--- | :--- | :--- | :--- | :--- |
| **Pickaxes** | Wood Pickaxe | `MAKE_WOOD_PICKAXE` | 1 Wood | Table | Level 1 Pickaxe |
| | Stone Pickaxe | `MAKE_STONE_PICKAXE` | 1 Wood, 1 Stone | Table | Level 2 Pickaxe |
| | Iron Pickaxe | `MAKE_IRON_PICKAXE` | 1 Wood, 1 Stone, 1 Iron, 1 Coal | **Table + Furnace** | Level 3 Pickaxe |
| | Diamond Pickaxe | `MAKE_DIAMOND_PICKAXE` | 1 Wood, 3 Diamond | Table | Level 4 Pickaxe |

```
| **Swords** | Wood Sword | `MAKE_WOOD_SWORD` | 1 Wood | Table | Level 1 Sword |
| | Stone Sword | `MAKE_STONE_SWORD` | 1 Wood, 1 Stone | Table | Level 2 Sword |
| | Iron Sword | `MAKE_IRON_SWORD` | 1 Wood, 1 Stone, 1 Iron, 1 Coal | **Table +
    Furnace** | Level 3 Sword |
| | Diamond Sword | `MAKE_DIAMOND_SWORD` | 1 Wood, 2 Diamond | Table | Level 4 Sword
     |
| **Armour** | Iron Armour | `MAKE_IRON_ARMOUR` | 3 Iron, 3 Coal | **Table + Furnace
    ** | +1 Defence (fills slot) |
| | Diamond Armour | `MAKE_DIAMOND_ARMOUR` | 3 Diamond | Table | +2 Defence (fills
    slot) |
| **Consumables** | Arrows | `MAKE_ARROW` | 1 Wood, 1 Stone | Table | **+2** Arrows
    (Max 99) |
| | Torches | `MAKE_TORCH` | 1 Wood, 1 Coal | Table | **+4** Torches (Max 99) |
```

**Achievements:**
Crafting Iron Armour or Diamond Armour triggers their respective achievements (`
    MAKE_IRON_ARMOUR`, `MAKE_DIAMOND_ARMOUR`).

The `do_action` function executes the **`Action.DO`** command, allowing the agent to
     interact with the block directly in front of them (the "target block").

**Priority Logic:**
1.  **Combat First:** The system first attempts to attack a mob in the target block
    (via `attack_mob`).
2.  **Block Interaction:** If **no mob was attacked**, the agent interacts with the
    static block in the target location.

**Mining & Harvesting Rules:**
Mining permanently removes the block (replacing it with `PATH` or `GRASS`) and adds
    items to the inventory, provided the agent meets the **Tool Requirement**.

| Target Block | Tool Requirement | Inventory Gain | Replacement Block |
| :--- | :--- | :--- | :--- |
| **Tree** (Normal/Fire/Ice) | None | +1 Wood | Grass / Fire Grass / Ice Grass |
| **Stone** | Pickaxe Level $\ge$ 1 | +1 Stone | Path |
| **Coal** | Pickaxe Level $\ge$ 1 | +1 Coal | Path |
| **Stalagmite** | Pickaxe Level $\ge$ 1 | +1 Stone | Path |
| **Iron** | Pickaxe Level $\ge$ 2 | +1 Iron | Path |
| **Diamond** | Pickaxe Level $\ge$ 3 | +1 Diamond | Path |
| **Sapphire / Ruby** | Pickaxe Level $\ge$ 4 | +1 Sapphire / Ruby | Path |
| **Grass** | None | 10% Chance: +1 Sapling | (Remains Grass) |

**Consumables & Restoration:**
* **Water / Fountain:** Drinking fills `player_drink` to max and resets `
    player_thirst` to 0. (Achievement: `COLLECT_DRINK`)
* **Ripe Plant:** Eating adds +4 `player_food` (up to max), resets `player_hunger`
    to 0, and reverts the block to an unripe `PLANT`. (Achievement: `EAT_PLANT`)

**Other Interactions:**
* **Chest:** Opens the chest, removing it from the map and granting random loot via
    `add_items_from_chest`. (Achievement: `OPEN_CHEST`)
* **Workstations (Furnace / Crafting Table):** "Mining" these destroys them (
    replaces with `PATH`) without returning resources.
* **Boss (Necromancer):** If the boss is in the target block, is vulnerable, and the
     fight is active, the agent deals damage, incrementing `boss_progress`. (
    Achievement: `DAMAGE_NECROMANCER`)

The `place_block` function executes specific **Placement Actions** to modify the
    environment in the cell directly **in front of** the agent (the "target cell").

**General Validation:**
All placement actions fail (resulting in no state change) if:
1.  The target cell is out of bounds.

2.  There is a **Mob** in the target cell.
3.  (For most blocks) The target cell already contains a solid block or item.

**Placement Logic & Costs:**

| Action | Cost | Target Requirement | Result |
| :--- | :--- | :--- | :--- |
| **PLACE_TABLE** | 2 Wood | Empty, Valid Terrain | Target block becomes **Crafting Table**. |
| **PLACE_FURNACE** | 1 Stone | Empty, Valid Terrain | Target block becomes **Furnace**. |
| **PLACE_STONE** | 1 Stone | Empty **OR Water** | Target block becomes **Stone**. (Allows bridging over water). |
| **PLACE_TORCH** | 1 Torch | Valid Surface, No Item | Adds **Torch** to `item_map`. Updates **Light Map**. |
| **PLACE_PLANT** | 1 Sapling | **Grass** Block, No Item | Target block becomes **Plant**. Registers plant for growth. |

**Special Mechanics:**
* **Torches & Lighting:** Placing a torch instantly updates the `light_map` by adding a specific light gradient (`TORCH_LIGHT_MAP`) centered on the torch, clipped between 0.0 and 1.0.
* **Plants:** Placing a sapling on grass initiates the farming cycle. The plant is added to `growing_plants_positions` and initialized with an age of 0.
* **Achievements:** Successfully placing any of these items triggers the corresponding Achievement.

The `shoot_projectile` function manages the mechanics of firing ranged weapons (Arrows).

**Trigger Condition:** Action is `SHOOT_ARROW`.

**Prerequisites (ALL must be met):**
1.  **Equipment:** The agent possesses a Bow (`inventory.bow >= 1`).
2.  **Ammo:** The agent possesses at least one Arrow (`inventory.arrows >= 1`).
3.  **Projectile Limit:** The number of active player projectiles on the current level is below the hard cap (`static_params.max_player_projectiles`).

**Execution Results:**
* **Spawn:** A new projectile (Type: `ARROW2`) is instantiated at the player's current coordinates.
* **Trajectory:** The projectile is assigned a velocity vector matching the agent's current facing direction (`player_direction`).
* **Cost:** Inventory `arrows` count is decremented by 1.
* **Achievement:** Unlocks the `FIRE_BOW` achievement.

**Failure Case:** If any prerequisite is not met, the action is ignored, and no ammunition is consumed.

The `cast_spell` function handles the deployment of magic projectiles (`FIREBALL` or `ICEBALL`) based on the agent's action.

**Prerequisites (ALL must be met):**
1.  **Mana:** The agent must have at least **2 Mana**.
2.  **Knowledge:** The specific spell must be unlocked in the agent's `learned_spells` array (Index 0 for Fireball, Index 1 for Iceball).
3.  **Projectile Limit:** The active projectile count must be below the hard cap (`static_params.max_player_projectiles`).

**Spell Mechanics:**

| Spell | Action Trigger | Cost | Projectile Type |
| :--- | :--- | :--- | :--- |

```
| **Fireball** | `CAST_FIREBALL` | 2 Mana | `FIREBALL` |
| **Iceball** | `CAST_ICEBALL` | 2 Mana | `ICEBALL` |
```

**Execution Results:**
* **Spawn:** Creates a projectile of the determined type at the player's current
    position.
* **Trajectory:** The projectile moves in the direction the player is currently
    facing.
* **Cost:** Deducts **2 Mana** from `player_mana`.
* **Achievement:** Unlocks `CAST_FIREBALL` or `CAST_ICEBALL` upon successful cast.

The `drink_potion` function handles the consumption of consumables, featuring a **
    Roguelike Potion Mechanics** system where the relationship between potion *Color
    * and potion *Effect* is randomized via `state.potion_mapping`.

**Trigger & Validation:**
* **Actions:** 6 distinct actions exist: `DRINK_POTION_[COLOR]` (Red, Green, Blue,
    Pink, Cyan, Yellow).
* **Prerequisite:** The agent must possess at least **1 unit** of the specific
    potion color in their inventory.

**Effect Resolution:**
The specific effect applied is determined by looking up the potion's color index in
    `state.potion_mapping`.

| Effect Index | Stat Affected | Change | Description |
| :--- | :--- | :--- | :--- |
| **0** | Health | **+8** | Major Heal |
| **1** | Health | **-3** | Poison / Damage |
| **2** | Mana | **+8** | Restore Mana |
| **3** | Mana | **-3** | Drain Mana |
| **4** | Energy | **+8** | Restore Energy |
| **5** | Energy | **-3** | Drain Energy |

**State Updates:**
* **Inventory:** Decrements the count of the specific potion color consumed.
* **Stats:** Updates `player_health`, `player_mana`, and `player_energy` based on
    the resolved effect.
* **Achievement:** Unlocks the `DRINK_POTION` achievement.

The `read_book` function allows the agent to permanently unlock new magic abilities
    by consuming a Book.

**Trigger & Validation:**
* **Action:** `READ_BOOK`
* **Prerequisite:** The agent possesses at least **1 Book** in their inventory.

**Learning Mechanic:**
The system randomly selects one spell from the set of spells the agent has **not yet
    learned**.
* **Index 0:** Fireball
* **Index 1:** Iceball

**Execution Results:**
* **Inventory:** Consumes 1 Book.
* **Knowledge:** The selected spell is set to `True` in `state.learned_spells`,
    enabling the corresponding `CAST_` action.
* **Achievements:** Unlocks either `LEARN_FIREBALL` or `LEARN_ICEBALL` based on the
    spell acquired.

The `enchant` function allows the agent to imbue weapons and armor with elemental
    properties (Fire or Ice) by interacting with a specific **Enchantment Table**
    block directly in front of them.

**Prerequisites (ALL must be met):**
1.  **Position:** Agent must be facing an `ENCHANTMENT_TABLE_FIRE` or `ENCHANTMENT_TABLE_ICE`.
2.  **Mana:** Agent must have at least **9 Mana**.
3.  **Gem:** Agent must possess the specific gem required by the table type.
    * **Fire Table:** Requires 1 **Ruby**.
    * **Ice Table:** Requires 1 **Sapphire**.
4.  **Item:** Agent must possess the item corresponding to the action (Sword, Bow, or at least one Armour piece).

**Enchantment Mechanics:**
* **Element Type:** Defined by the table (Fire Table $\to$ Type 1, Ice Table $\to$ Type 2).
* **Sword / Bow:** Applies the enchantment type to the item, overwriting any previous enchantment.
* **Armour:** The system selects a single armor slot to enchant based on the following priority:
    1.  **Unenchanted Slots:** Randomly selects a slot that currently has no enchantment.
    2.  **Overwrite:** If (and only if) all slots are full, it randomly selects a slot containing the **opposite** element to overwrite.

**Costs & Rewards:**
* **Resources:** Consumes **1 Gem** (Ruby/Sapphire) and **9 Mana**.
* **Achievements:** Triggers `ENCHANT_SWORD` or `ENCHANT_ARMOUR` upon success.

The `boss_logic` function manages the passive state updates regarding the **Necromancer Boss**.

**Logic:**
1.  **Fight Timer:** If the boss fight is currently active (`is_fighting_boss`), the internal timer `boss_timesteps_to_spawn_this_round` is decremented by 1.
2.  **Victory Check:** The system evaluates if the victory condition is met (via `has_beaten_boss`). If true, the `DEFEAT_NECROMANCER` achievement is unlocked.

The `level_up_attributes` function manages character progression, allowing the agent to convert Experience Points (XP) into permanent stat increases.

**Prerequisites:**
1.  **Cost:** The agent must have at least **1 XP**.
2.  **Cap:** The target attribute must be below the maximum limit defined by `params.max_attribute`.

**Actions & Effects:**

| Action | Attribute Increased | Effect Description (Implicit) |
| :--- | :--- | :--- |
| `LEVEL_UP_DEXTERITY` | **+1 Dexterity** | typically increases ranged damage/accuracy. |
| `LEVEL_UP_STRENGTH` | **+1 Strength** | typically increases melee damage. |
| `LEVEL_UP_INTELLIGENCE` | **+1 Intelligence** | typically increases magic potential/mana. |

**Execution:**
If the action is valid and prerequisites are met, the specific attribute is incremented by 1, and `player_xp` is decremented by 1.

The `move_player` function handles the agent's attempt to move horizontally within the current level using directional commands (Up, Down, Left, Right).

**Logic Flow:**

1. **Proposed Movement:** Calculates the target coordinate based on the `action` vector.
2. **Validation:** Checks if the target coordinate is valid via `is_position_in_bounds_not_in_mob_not_colliding`. A move is valid if:
   * It is within map boundaries.
   * It is **not** occupied by a Mob.
   * It is **not** a solid block (Collision logic: `COLLISION_LAND_CREATURE`).
   * *Override:* If `params.god_mode` is active, collisions are ignored.
3. **Position Update:**
   * **Success:** If valid, the agent's coordinate is updated.
   * **Failure:** If invalid (blocked), the agent remains at the current coordinate.
4. **Direction Update:**
   * The agent's `player_direction` is updated to match the input `action`, **regardless of whether the move succeeded**. This allows the agent to turn and face a wall without moving into it.

The `update_mobs` function executes the AI cycles for all entities in the environment. It processes groups sequentially: Melee Mobs, Passive Mobs, Ranged Mobs, Mob Projectiles, and Player Projectiles.

**1. Melee Mobs (Zombies, Skeletons, etc.)**
* **AI Logic:**
    * **Chase:** If within trigger distance (<`task.melee_trigger_distance` blocks) or fighting a boss, there is a **75% chance** to move directly toward the player.
    * **Wander:** Otherwise (or if the 75% check fails), moves in a random cardinal direction.
* **Attack:** If adjacent (Distance = 1) and Cooldown $\le$ 0:
    * Deals damage based on mob type multiplied by `task.mob_damage_multiplier`.
    * **Sleep Critical:** Damage is multiplied by **2.5x** if the player is sleeping. Wakes the player.
    * Sets Cooldown to 5.
* **Despawn:** Despawns if the distance to the player exceeds `params.mob_despawn_distance` (unless fighting a Boss).

**2. Passive Mobs (Cows, Sheep)**
* **AI Logic:** Moves randomly (50% chance to stay still, 50% chance to move).
* **Despawn:** Same distance rules as Melee Mobs.

**3. Ranged Mobs (Archers)**
* **AI Logic (Kiting Behavior):**
    * **Distance $\le$ 3:** Moves **AWAY** from player.
    * **Distance $\ge$ 6:** Moves **TOWARD** player.
    * **Distance 4-5:** Random movement.
    * *Noise:* 15% chance to ignore logic and move randomly.
* **Attack:** Fires a projectile if:
    * Distance is **4 or 5**.
    * **OR** Distance is $\le$ 3 AND the mob is cornered (cannot move away).
* **Cooldown:** Sets to 4 after shooting.

**4. Mob Projectiles**
* **Movement:** Travels in a straight line.
* **Collision:**
    * **Player:** Deals damage and destroys the projectile.
    * **Walls:** Destroys the projectile (Arrows travel over Water).
    * **Infrastructure:** Destroys **Furnaces** and **Crafting Tables** upon impact.

**5. Player Projectiles**
* **Damage Calculation:**
    * **Base:** Defined by projectile type (Arrow, Fireball, Iceball).

* **Enchantment:** Bow enchantments add elemental damage to arrows.
* **Attribute Scaling:**
    * **Arrows:** Multiplied by $1 + 0.2 \times (\text{Dexterity} - 1)$.
    * **Magic:** Multiplied by $1 + 0.5 \times (\text{Intelligence} - 1)$.
* **Collision:** checks for impact with mobs (via `attack_mob`) or walls along the trajectory.

The `spawn_mobs` function manages the stochastic generation of new entities on the current level. It evaluates spawning for Passive, Melee, and Ranged mobs sequentially.

**Global Spawn Coefficient:**
Spawn probabilities are scaled dynamically:
1.  **Uncleared Bonus:** If the level is not yet cleared (kills < required), spawn rates are **tripled (3x)**.
2.  **Boss Wave:** During specific "Boss Spawn Waves" in the Necromancer fight, rates are multiplied by **1000x** to guarantee spawns.

**General Spawn Constraints:**
* **Cap:** Spawning fails if the count of that mob type reaches its specific `max_[type]_mobs` limit.
* **Collision:** Mobs cannot spawn on top of existing mobs or solid blocks.
* **Despawn Radius:** All spawns must occur within `params.mob_despawn_distance` of the player.

**Spawn Logic by Type:**

| Feature | Passive Mobs | Melee Mobs (Hostile) | Ranged Mobs (Hostile) |
| :--- | :--- | :--- | :--- |
| **Spawn Chance** | Base Chance $\times$ `task.passive_spawn_multiplier` | (Base + **Night Bonus**) $\times$ `task.melee_spawn_multiplier` | Base Chance $\times$ `task.ranged_spawn_multiplier` |
| **Night Bonus** | None | Increases as `light_level` decreases (Square of darkness). | None |
| **Valid Terrain** | Grass, Path, Fire/Ice Grass | **Normal:** Any valid ground.
**Boss:** Graves only. | **Normal:** Any valid ground.
**Boss:** Graves only.
**Deep Thing:** Water only. |
| **Min Distance** | $> 3$ blocks from player | **Normal:** $> 9$ blocks.
**Boss:** $\le 6$ blocks. | **Normal:** $> 9$ blocks.
**Boss:** $\le 6$ blocks. |
| **Boss Logic** | **DISABLED** during boss fights. | Spawns heavily on Graves. | Spawns heavily on Graves. |

**Entity Initialization:**
* **Type:** Determined by `player_level` (or `boss_progress` during boss fights).
* **Health:** Base health from `MOB_TYPE_HEALTH_MAPPING` multiplied by `task.mob_health_multiplier`.
* **Position:** Randomly selected from valid tiles meeting all criteria.

The `update_plants` function manages the passive growth cycle of farming crops.

**Growth Logic:**
* **Aging:** Every timestep, the age of all active plants is incremented by 1.
* **Maturation:** When a plant's age reaches the threshold defined by `task.growing_plants_age`, it transitions to a mature state.

**State Update:**
* **Visual/Functional Change:** Upon maturation, the block at the plant's coordinates on Map Level 0 is updated from `PLANT` (unripe) to `RIPE_PLANT` (harvestable).

The `update_player_intrinsics` function manages the agent's biological metabolism, handling the decay and regeneration of stats via an **Accumulator/Threshold system**. Hidden counters increment every step; when they overflow a threshold, the actual inventory/stat updates.

**Global Modifiers & Task Parameters:**
* **Dexterity:** Reduces the base rate of Hunger, Thirst, and Fatigue accumulation (`intrinsic_decay_coeff`).
* **Task Multipliers:**
    * `task.needs_depletion_multiplier`: Scales the speed at which Hunger, Thirst, and Fatigue accumulate.
    * `task.health_recover_multiplier`: Scales how fast Health restores when needs are met.
    * `task.health_loss_multiplier`: Scales how fast Health drains when starving/ dehydrated.
    * `task.mana_recover_multiplier`: Scales the specific Mana regeneration bonus granted by **Intelligence**.

**States:**
* **Sleeping (`Action.SLEEP`):** Starts if `Energy < Max`. Wakes when `Energy == Max`. Doubles health recovery speed and halves hunger/thirst accumulation.
* **Resting (`Action.REST`):** Starts if `Health < Max`. Wakes if `Health == Max` OR `Food/Drink` run out.

**Survival Mechanics:**

| Stat | Accumulation Logic | Threshold | Result |
| :--- | :--- | :--- | :--- |
| **Hunger** | (Base Rate based on Dex) $\times$ `task.needs_depletion_multiplier` | **> 25** | **-1 Food** |
| **Thirst** | (Base Rate based on Dex) $\times$ `task.needs_depletion_multiplier` | **> 20** | **-1 Drink** |
| **Fatigue** | (Base Rate based on Dex) $\times$ `task.needs_depletion_multiplier` | **> 30**
**< -10** | **-1 Energy**
**+1 Energy** |

**Regeneration Logic:**

| Stat | Conditions | Calculation | Threshold | Result |
| :--- | :--- | :--- | :--- | :--- |
| **Health** | **Requirements:** Food, Drink, & Energy > 0.
**If Met:** | Rate = (2.0 if Sleeping, else 1.0) $\times$ `task.health_recover_multiplier` | **> 25** | **+1 Health** |
| | **If Failed (Starving):** | Rate = (-0.5 if Sleeping, else -1.0) $\times$ `task.health_loss_multiplier` | **< -15** | **-1 Health** |
| **Mana** | Always active. | Rate = (Base + Sleep Bonus) $\times$ (1 + 0.25 $\times$ (Int - 1) $\times$ `task.mana_recover_multiplier`) | **> 30** | **+1 Mana** |
"""

---

**Module: World Generation (world_gen_nl.py)**

```
context = """
# Craftax Environment: Initial State Description

## 1. World Structure & Geography
The world consists of 9 distinct levels vertically stacked (z-axis). The player
    navigates between levels using ladders (Ladder Down / Ladder Up).

### Level Ordering (Index 0 to 8):
0.  **Overworld (SmoothGen):**
```

```
       * Terrain: Grass (Default), Water (Sea), Sand (Coast), Stone (Mountains).
       * Features: Trees, Lava lakes.
       * Ores: Coal (3%), Iron (2%), Diamond (0.1%).
       * Lighting: Fully lit (1.0).
1.   **Dungeon:**
       * Structure: Procedurally generated rooms connected by corridors.
       * Features: Chests, Fountains, Torches in corners.
       * Special: Basic pathing and walls.
2.   **Gnomish Mines (SmoothGen):**
       * Terrain: Path/Stone mix.
       * Features: Stalagmites (Trees), Lava lakes.
       * Ores: Coal, Iron, Diamond, Sapphire, Ruby.
       * Lighting: Pitch black (0.0).
3.   **Sewers (Dungeon):**
       * Structure: Rooms and corridors.
       * Features: Water channels, Ice Enchantment Tables.
4.   **Vaults (Dungeon):**
       * Structure: Rooms and corridors.
       * Features: Fire Enchantment Tables.
5.   **Troll Mines (SmoothGen):**
       * Terrain: Similar to Gnomish Mines but different mob density/generation.
       * Ores: Higher probabilities for Iron and Gems.
       * Lighting: Pitch black (0.0).
6.   **Fire Level (SmoothGen):**
       * Terrain: Fire Grass, Lava Oceans.
       * Features: Fire Trees.
       * Ores: Coal, Ruby.
       * Lighting: Fully lit (1.0).
7.   **Ice Level (SmoothGen):**
       * Terrain: Ice Grass, Water Oceans.
       * Features: Ice Shrubs.
       * Ores: Diamond, Sapphire.
       * Lighting: Pitch black (0.0).
8.   **Boss Level (SmoothGen):**
       * Terrain: Surrounded by Walls.
       * Features: Graves, Necromancer Spawn.
       * Ores: Mossy Walls, Grave Variants.
       * Lighting: Pitch black (0.0).
```

## 2. Terrain Generation Logic
* **SmoothGen Levels (0, 2, 5, 6, 7, 8):** Generated using fractal noise (Perlin-like). Terrain height determines Water -> Sand -> Grass -> Mountain -> Inner Cave. Ores are distributed stochastically within specific blocks (usually Stone).
* **Dungeon Levels (1, 3, 4):** Generated using a room-placement algorithm with collision detection. Rooms are connected via orthogonal paths. Special blocks (Chests, Fountains) are placed randomly within rooms. Walls adjacently touching paths become "Mossy Walls" with low probability.

## 3. Player Initial State
* **Position:** Center of the Overworld (Level 0).
* **Attributes:**
    * Strength: 1
    * Dexterity: 1
    * Intelligence: 1
* **Status Bars (Max 9):**
    * Health: 9.0 / 9.0
    * Food (Hunger): 9 / 9
    * Drink (Thirst): 9 / 9
    * Energy (Fatigue): 9 / 9
    * Mana: 9 / 9
* **Inventory:** Completely empty (all counts set to 0).

```
        * *Note:* If `god_mode` is True, inventory starts full (99 of all resources,
            high-tier tools).
* **Equipment:**
        * Sword/Bow Enchantment: Level 0.
        * Armor Enchantment: [0, 0, 0, 0].
* **Active Effects:**
        * Is Sleeping: False
        * Is Resting: False
        * Learned Spells: [False, False]

## 4. World Dynamics & Mobs
* **Mobs:** Arrays are initialized for Melee, Ranged, and Passive mobs, but start
    masked (inactive) until spawned by game logic.
* **Projectiles:** Arrays initialized for Player and Mob projectiles (inactive).
* **Plants:** Growing plants array initialized to zero.
* **Potions:** The color-to-effect mapping for potions is randomized (permuted) at
    the start of every episode (6 types).
* **Ladders:**
        * Ladders are procedurally placed.
        * The Ladder Down on Level 0 starts OPEN (Logic: `monsters_killed[0]` is
            initialized to 10 to bypass the kill requirement for the first ladder).
* **Achievements:** All set to False.
* **Boss:** Progress set to 0.

## 5. Global Constants
* **Map Size:** Defined by `static_params`.
* **Chunk Size:** 16 (for dungeon generation).
* **Light Calculation:** Recalculated based on torches, lava proximity, and level
    default light settings.
"""
```

---

**Module: API Documentation (minicraftax_api.py)**

```
context = """
To generate valid and interesting tasks, you must first understand the capabilities
    of this API.

These tasks are defined by two components:
1.  **`TaskParams`**: A set of parameters that modify the core game mechanics (e.g.,
     making mobs harder, making survival needs more pressing).
2.  **`WorldBuilder`**: A class used to programmatically set up the initial state of
     the world (e.g., placing specific blocks, or pre-filling the player's inventory
    ).

## MiniCraftax API Documentation

Below is the documentation for the `TaskParams` and `WorldBuilder` classes. Review
    it carefully.

---

### `TaskParams`: Modifying Game Mechanics

The `TaskParams` class allows you to tweak the game's dynamic rules. You can think
    of this as setting the "difficulty" or "ruleset" for a specific task. The
    default values are the original Craftax game.

* `passive_spawn_multiplier: float`
        * **Effect:** Scales the base spawn chance for **passive mobs** (like cows). A
            value of `2.0` doubles the spawn rate, while `0.5` halves it. `0` means no
```

```
            passive mobs and any existing passive mobs are removed if agent goes far
            from them.
        * **Default:** `1.0`

*   `melee_spawn_multiplier: float`
        * **Effect:** Scales the base spawn chance for **melee mobs** (like zombies).
          `0` means no melee mobs and any existing melee mobs are removed if agent
          goes far from them.
        * **Default:** `1.0`

*   `ranged_spawn_multiplier: float`
        * **Effect:** Scales the base spawn chance for **ranged mobs** (like skeletons).
          `0` means no ranged mobs and any existing ranged mobs are removed if agent
          goes far from them.
        * **Default:** `1.0`

*   `mob_health_multiplier: float`
        * **Effect:** Multiplies the base health of all mobs (passive, melee, and ranged
          ) when they spawn. A value of `2.0` means mobs spawn with double health.
        * **Default:** `1.0`

*   `mob_damage_multiplier: float`
        * **Effect:** Multiplies the base damage of all mob attacks (both melee and
          ranged projectiles). A value of `2.0` doubles mob damage.
        * **Default:** `1.0`

*   `melee_trigger_distance: int`
        * **Effect:** The Manhattan distance at which melee mobs will detect the player
          and begin chasing them.
        * **Default:** `10`

*   `monsters_killed_to_clear_level: int`
        * **Effect:** The number of hostile monsters (melee or ranged) the player must
          defeat on a level to unlock the "ladder down" to the next floor.
        * **Default:** `8` except for floor 0 which is `0`.

*   `needs_depletion_multiplier: float`
        * **Effect:** Scales the rate at which the player's **hunger, thirst, and
          fatigue** increase. A value of `2.0` means the player gets hungry, thirsty,
          and tired twice as fast.
        * **Default:** `1.0`

*   `health_recover_multiplier: float`
        * **Effect:** Scales the speed of the player's natural **health regeneration** (
          which only occurs when all needs are met).
        * **Default:** `1.0`

*   `health_loss_multiplier: float`
        * **Effect:** Scales the speed at which the player **loses health** from unmet
          needs (e.g., starvation or dehydration).
        * **Default:** `1.0`

*   `mana_recover_multiplier: float`
        * **Effect:** Scales the speed of the player's natural **mana regeneration**.
        * **Default:** `1.0`

*   `growing_plants_age: int`
        * **Effect:** The number of game steps (ticks) it takes for a planted `PLANT` to
          mature into a `RIPE_PLANT`.
        * **Default:** `600`

---
```

```
### `WorldBuilder`: Setting the Initial State

The `WorldBuilder` class provides methods to modify the initial state of the world.

* `set_starting_floor(self, level: int)`
    * **Effect:** Sets the player's starting floor. If `level > 0`, the player will
        spawn at that level's "up ladder" position.
    * **Example:** `builder.set_starting_floor(3)` starts the player in the Sewers.

* `set_player_stats(self, dexterity: int = 1, strength: int = 1, intelligence: int =
    1)`
    * **Effect:** Sets the player's starting attributes (DEX, STR, INT). Values are
        clamped between 1 and 5.
    * **Example:** `builder.set_player_stats(strength=3)` gives the player a
        starting strength boost.

* `set_player_inventory(self, inventory_dict: dict)`
    * **Effect:** Sets the player's starting inventory. Takes a dictionary where
        keys are item names (e.g., "wood", "stone", "pickaxe") and values are the
        counts.
    * **Example:** `builder.set_player_inventory({"stone": 20, "pickaxe": 1})`

* `set_weapon_enchantments(self, sword: int = 0, bow: int = 0)`
    * **Effect:** Sets the player's starting weapon enchantments. (0 = None, 1 =
        Fire, 2 = Ice).
    * **Example:** `builder.set_weapon_enchantments(sword=1)` starts the player with
         a fire-enchanted sword.

* `set_armour_enchantments(self, helmet: int = 0, chestplate: int = 0, leggings: int
     = 0, boots: int = 0)`
    * **Effect:** Sets enchantments for each armour slot. (0 = None, 1 = Fire, 2 =
        Ice).

* `set_learned_spells(self, fireball: bool = False, iceball: bool = False)`
    * **Effect:** Sets whether the player starts the game having already learned the
         Fireball or Iceball spells.
    * **Example:** `builder.set_learned_spells(fireball=True)`

* `set_monsters_killed(self, level: int, count: int)`
    * **Effect:** Manually sets the monster kill count for a specific `level`. This
        can be used to pre-unlock the "ladder down" for that level.
    * **Example:** `builder.set_monsters_killed(level=0, count=8)` unlocks the
        ladder from the Overworld immediately.

* `place_block(self, level: int, block_type: BlockType, position: tuple)`
    * **Effect:** Places a *single* block (e.g., `BlockType.CRAFTING_TABLE`) at an
        exact (row, col) `position` on a specific `level`.
    * **Example:** `builder.place_block(0, BlockType.DIAMOND, (24, 25))`

* `fill_area(self, level: int, block_type: BlockType, top_left: tuple, bottom_right:
     tuple)`
    * **Effect:** Fills a rectangular area on a specific `level` with a `block_type
        `.
    * **Example:** `builder.fill_area(0, BlockType.WATER, (10, 10), (20, 20))`
        creates a lake.

* `place_randomly(self, rng: jax.Array, level: int, block_type: BlockType, n: int =
    1, on_blocks: List[BlockType] = ...)`
    * **Effect:** Places `n` blocks of `block_type` at random locations on the
        specified `level`. The blocks are only placed on top of blocks specified in
        the `on_blocks` list (e.g., `[BlockType.GRASS]`).
    * **Example:** `builder.place_randomly(rng, 0, BlockType.TREE, 50, [BlockType.
        GRASS])`
```

```
* `place_randomly_near(self, rng: jax.Array, level: int, block_type: BlockType,
    target_pos: tuple, min_dist: int, max_dist: int, n: int = 1, on_blocks: List[
    BlockType] = ...)`
    * **Effect:** Places `n` blocks randomly within a Manhattan distance range (`
        min_dist` to `max_dist`) from a `target_pos` (row, col).
    * **Example:** `builder.place_randomly_near(rng, 0, BlockType.COAL, (24, 24), 2,
        5, 10, [BlockType.STONE])`

* `add_mobs_randomly_near(self, rng: jax.Array, level: int, mob_name: str, type_id:
    int, n: int = 1, target_pos: jnp.ndarray = None, min_dist: int = 0, max_dist:
    int = 5, on_blocks: List[BlockType] = ...)`
    * **Effect:** Adds `n` mobs of `mob_name` randomly within a Manhattan distance
        range (`min_dist` to `max_dist`) from `target_pos`. If `target_pos` is `None
        `, defaults to player position.
    * **Example:** `builder.add_mobs_randomly_near(rng, 0, "melee", MobType.MELEE.
        value, 5, (30, 30), 2, 8, [BlockType.GRASS])`
    * **NOTES:**
        * we can place maximum 3 melee mobs, 3 passive mobs, and 2 ranged mobs
        * once the mobs are placed, the spawning and update logic of mobs works as
            normal and thus the inital mobs might be removed if the player is not
            close enough to them

* `place_adjacent_to_existing(self, rng: jax.Array, level: int, block_to_place:
    BlockType, target_block_type: BlockType, on_blocks: List[BlockType] = ...)`
    * **Effect:** Finds one existing `target_block_type` and places a `
        block_to_place` in a valid, random adjacent spot.
    * **Example:** `builder.place_adjacent_to_existing(rng, 0, BlockType.FURNACE,
        BlockType.CRAFTING_TABLE, [BlockType.GRASS])`

* `build(self, rng: jax.Array)`
    * **Effect:** Finalizes the world and returns the complete `EnvState` object.
        This is the final call in any world-building chain.
"""
```

### B.1.3. USER PROMPT AND PERFORMANCE FORMATTING

This prompt provides the specific context for the current generation cycle: parent level $\lambda_p$ and agent stats ($\text{perf}_p, \text{perf}_{\text{target}}$).

---

**User Prompt Template**

```
user_prompt = """
**REMINDER: You are generating a new, creative task description, NOT code.**

Here is the description of the trained task:
<trained_task>
{TRAINED_TASK}
</trained_task>

Here is the performance evaluation from the **trained task's training session**.
(This shows *all* skills the agent learned *while training on this specific task*.
    If some relevant achievements are not here, then the agent never achieved them
    during training, and means that it has weaknesses to address.)
<task_performance_context>
{TASK_PERFORMANCE_CONTEXT}
</task_performance_context>

Here is the **global evaluation** of the agent on the full Craftax game.
(This shows the agent's *general* skill set, learned from *all* tasks.)
<global_agent_profile>
```

```
{GLOBAL_AGENT_PROFILE}
</global_agent_profile>

**Your output should be a reasoning section followed by a detailed docstring for the
    new task. Focus on creating a task that is a meaningful variation or extension
    of the trained task, using both performance reports to make an informed decision
    .**
"""
```

**Dynamic Profile Formatting**  The placeholders in the User Prompt are populated dynamically throughout curriculum iterations.

To ground the Description Generator, we convert the agent's raw JAX performance metrics into a natural language summary – performance profile $\text{perf}_p$. Below is an example of the exact text string provided to the model in the `<task_performance_context>` block. $\text{perf}_{\text{target}}$ has the exact same format.

---

**Example: Input Performance Profile**

```
While training on this task, the agent achieved:
- Main Goal Success Rate (SR): 45.00%
Detailed Skill SRs (including goal components and accidental skills):
  - COLLECT_WOOD: 100.00%
  - PLACE_TABLE: 85.50%
  - MAKE_WOOD_PICKAXE: 45.00%
  - EAT_COW: 12.00%
```

---

## B.2. Code Generation Phase

Following the description generation, a second foundation model inference step synthesizes the executable JAX code. This phase relies on a "JAX Expert" persona and a comprehensive context containing the interface definitions. The system prompt orchestrates the pre-defined part of the input to the foundation model: $(c_2^{\text{target}}, m_2)$.

### B.2.1. SYSTEM PROMPT (CODE GENERATOR)

---

**System Prompt: Craftax Coder**

```
system_prompt = """
You are an expert JAX programmer specializing in the `MiniCraftax` library. Your
    sole job is to take a natural language task description and implement it as a
    complete, syntactically correct, and JAX-compatible Python class that inherits
    from `BaseTask`.

## 1. KNOWLEDGE BASE (API DOCUMENTATION)

You must use the following Python classes and constants. Do not invent new functions
    or classes; use these APIs exactly as they are defined.

### MiniCraftax Library Code
<minicraftax_code>
{MINICRAFTAX_CODE}
</minicraftax_code>

### Craftax Core Library Code
<craftax_code>
{CRAFTAX_CODE}
</craftax_code>
```

```
### Mob Information
<mob_info>
{MOBS}
</mob_info>

## 2. CORE TASK: IMPLEMENTATION INSTRUCTIONS

You must generate a complete Python file that follows these instructions precisely:

1. Import the necessary libraries.

2.  **Class Definition:** Define a new class that inherits from `BaseTask`. The
    class name should be Env.

3.  **Docstring:** Copy the provided natural language task description exactly as
    the class's docstring.

4.  **`__init__` Method:**
    - Implement the `__init__` method. It must call `super().__init__(static_params,
        params)`.
    - CRITICAL: Set the self.relevant_achievements to the appropriate achievements
       based on the "Achievements" section of the task's docstring.
    - CRITICAL: Set the self.completed_achievements to the appropriate achievements
       based on the "Completed Achievements" section of the task's docstring.
    - CRITICAL: Set the self.label which is a string that lists all the relevant
       achievements.

5. **`get_task_params` Method:**
    - Implement the `get_task_params` method.
    - You MUST return a `TaskParams` instance overriding the values stated in task's
        docstring. If no values are specified, return `TaskParams()`.

5.  **`generate_world` Method:**
    - Implement the `generate_world` method.
    - You MUST use the `WorldBuilder` API to create the world exactly as described
       in the "World" section of the docstring.
    - You MUST use the `build` method of WorldBuilder to return the EnvState,
       matching the signature in the `BaseTask` definition.

## 3. OUTPUT FORMAT

Your response MUST be in the following format. Do NOT include any other text or
    explanations outside of these tags.

[The complete, final Python code for the task file goes here.]

"""
```

### B.2.2. INJECTED KNOWLEDGE BASE MODULES

The Code Generator's context is populated by injecting the source code of the underlying libraries. This ensures the model uses the exact API definitions available in the runtime environment. These modules together form $c_2^{\text{target}}$.

---

**Module: MiniCraftax Library (Injected into {MINICRAFTAX_CODE})**

```
context = """
=================================================
FILE: src/minicraftax/craftax_state.py
=================================================
```

```python
from flax import struct
import jax.numpy as jnp
from typing import Any
from craftax.craftax.craftax_state import Inventory, Mobs

@struct.dataclass
class TaskParams:
    /"/"/"Holds parameters that vary between MiniCraftax tasks./"/"/"
    passive_spawn_multiplier: float = 1.0      # Multiplier for passive mob spawn
        chance
    melee_spawn_multiplier: float = 1.0        # Multiplier for melee mob spawn
        chance
    ranged_spawn_multiplier: float = 1.0       # Multiplier for ranged mob spawn
        chance
    mob_health_multiplier: float = 1.0         # Multiplier for mob base health
    mob_damage_multiplier: float = 1.0         # Multiplier for mob base damage
    melee_trigger_distance: int = 10           # Distance at which melee mobs start
         chasing player
    monsters_killed_to_clear_level: int = 8    # Monsters needed to unlock next
        level ladder
    needs_depletion_multiplier: float = 1.0    # Multiplier for hunger/thirst/
        fatigue rates
    health_recover_multiplier: float = 1.0     # Multiplier for health recovery
        rate
    health_loss_multiplier: float = 1.0        # Multiplier for health loss rate (
        when needs unmet)
    mana_recover_multiplier: float = 1.0       # Multiplier for mana recovery rate
    growing_plants_age: int = 600              # Timesteps for a plant to become
        ripe

@struct.dataclass
class EnvState:
    task_id: int
    map: jnp.ndarray
    item_map: jnp.ndarray
    mob_map: jnp.ndarray
    light_map: jnp.ndarray
    down_ladders: jnp.ndarray
    up_ladders: jnp.ndarray
    chests_opened: jnp.ndarray
    monsters_killed: jnp.ndarray

    player_position: jnp.ndarray
    player_level: int
    player_direction: int

    # Intrinsics
    player_health: float
    player_food: int
    player_drink: int
    player_energy: int
    player_mana: int
    is_sleeping: bool
    is_resting: bool

    # Second order intrinsics
    player_recover: float
    player_hunger: float
    player_thirst: float
    player_fatigue: float
    player_recover_mana: float

    # Attributes
```

```
    player_xp: int
    player_dexterity: int
    player_strength: int
    player_intelligence: int

    inventory: Inventory

    melee_mobs: Mobs
    passive_mobs: Mobs
    ranged_mobs: Mobs

    mob_projectiles: Mobs
    mob_projectile_directions: jnp.ndarray
    player_projectiles: Mobs
    player_projectile_directions: jnp.ndarray

    growing_plants_positions: jnp.ndarray
    growing_plants_age: jnp.ndarray
    growing_plants_mask: jnp.ndarray

    potion_mapping: jnp.ndarray
    learned_spells: jnp.ndarray

    sword_enchantment: int
    bow_enchantment: int
    armour_enchantments: jnp.ndarray

    boss_progress: int
    boss_timesteps_to_spawn_this_round: int

    light_level: float

    achievements: jnp.ndarray

    state_rng: Any

    timestep: int

    fractal_noise_angles: tuple[int, int, int, int] = (None, None, None, None)

    # We use a default_factory because TaskParams() creates a concrete instance,
    # which can cause issues with JAX transformations if defined directly as a
       default.
    # The factory ensures a fresh instance is created when needed during
       initialization.
    task_params: TaskParams = struct.field(pytree_node=True, default_factory=
       TaskParams)

=================================================
FILE: src/minicraftax/tasks/base_task.py
=================================================
import jax
import jax.numpy as jnp
from minicraftax.craftax_state import EnvState, TaskParams
from craftax.craftax.constants import ACHIEVEMENT_REWARD_MAP
from craftax.craftax.util.game_logic_utils import has_beaten_boss

def clamp_task_params(params: TaskParams) -> TaskParams:
    /"/"/"Applies valid ranges to a TaskParams object./"/"/"
    return params.replace(
        # Example constraints (adjust ranges as needed)
        passive_spawn_multiplier = jnp.maximum(0.0, params.passive_spawn_multiplier)
            ,
```

```python
        melee_spawn_multiplier = jnp.maximum(0.0, params.melee_spawn_multiplier),
        ranged_spawn_multiplier = jnp.maximum(0.0, params.ranged_spawn_multiplier),
        mob_health_multiplier = jnp.maximum(0.01, params.mob_health_multiplier), #
            Avoid zero health
        mob_damage_multiplier = jnp.maximum(0.0, params.mob_damage_multiplier),
        melee_trigger_distance = jnp.maximum(1, params.melee_trigger_distance),
        monsters_killed_to_clear_level = jnp.maximum(0, params.
            monsters_killed_to_clear_level),
        needs_depletion_multiplier = jnp.maximum(0.0, params.
            needs_depletion_multiplier),
        health_recover_multiplier = jnp.maximum(0.01, params.
            health_recover_multiplier), # Avoid zero recovery
        health_loss_multiplier = jnp.maximum(0.0, params.health_loss_multiplier),
        mana_recover_multiplier = jnp.maximum(0.01, params.mana_recover_multiplier),
             # Avoid zero recovery
        growing_plants_age = jnp.maximum(2, params.growing_plants_age) # Min age 2
    )

class BaseTask:
    /"/"/"
    An abstract base class for all MiniCraftax tasks.

    This class defines the "contract" that all tasks must follow, ensuring
    they can be seamlessly used by the MiniCraftaxEnv.
    /"/"/"
    def __init__(self, static_params, params):
        self.static_params = static_params
        self.params = params

    def get_task_params(self) -> TaskParams:
        /"/"/"
        Returns the specific parameters for this task.
        Subclasses should override this method to provide custom parameters.
        /"/"/"
        # Return default parameters if not overridden
        return TaskParams()

    def is_terminal(self, state) -> bool:
        done_steps = state.timestep >= self.params.max_timesteps
        is_dead = state.player_health <= 0
        defeated_boss = has_beaten_boss(state, self.static_params)

        achievement_coeff = ACHIEVEMENT_REWARD_MAP
        relevant_achievements = jnp.array([b.value for b in self.
            relevant_achievements])
        current_achievements = state.achievements.astype(int)
        mask = jnp.zeros_like(current_achievements).at[relevant_achievements].set(1)
        total_possible_achievements = (mask * achievement_coeff).sum()
        total_achievements_so_far = (current_achievements * mask * achievement_coeff
            ).sum()

        task_solved = total_achievements_so_far >= total_possible_achievements

        return done_steps | is_dead | defeated_boss | task_solved

    def generate_world(self, rng: jax.Array) -> EnvState:
        /"/"/"
        Creates the world for this task and returns the EnvState .

        Returns:
            EnvState Class
        /"/"/"
        raise NotImplementedError("Each task must define its own world generation.")
```

```python
    def is_success(self, state) -> bool:
        /"/"/"
        Returns a binary True/False indicating if the task's primary
        objective was met in this state.
        /"/"/"
        # This is the default logic for achievement-based tasks.
        achievement_coeff = ACHIEVEMENT_REWARD_MAP
        relevant_achievements = jnp.array([b.value for b in self.
            relevant_achievements])
        current_achievements = state.achievements.astype(int)
        mask = jnp.zeros_like(current_achievements).at[relevant_achievements].set(1)
        total_possible_achievements = (mask * achievement_coeff).sum()
        total_achievements_so_far = (current_achievements * mask * achievement_coeff
            ).sum()

        task_solved = total_achievements_so_far >= total_possible_achievements

        return task_solved

=================================================
FILE: src/minicraftax/world_builder.py
=================================================
import jax
import jax.numpy as jnp
import jax.scipy as jsp
from typing import List, Tuple

# Import from the full craftax game
from craftax.craftax.constants import BlockType, ItemType, Action, Achievement,
    TORCH_LIGHT_MAP
from craftax.craftax.craftax_state import EnvParams, StaticEnvParams, Inventory,
    Mobs
from craftax.craftax.game_logic import calculate_light_level
from craftax.craftax.util.noise import generate_fractal_noise_2d
from craftax.craftax.game_logic import get_distance_map
from minicraftax.craftax_state import EnvState
from craftax.craftax.world_gen.world_gen_configs import (
    ALL_DUNGEON_CONFIGS,
    ALL_SMOOTHGEN_CONFIGS,
    SmoothGenConfig,
    DungeonConfig,
)
from craftax.craftax.world_gen.world_gen import get_new_empty_inventory,
    get_new_full_inventory

def _generate_base_smoothworld_level(
    rng: jax.Array,
    static_params: StaticEnvParams,
    player_position: jax.Array,
    config: SmoothGenConfig,
    params: EnvParams,
) -> jnp.ndarray:
    /"/"/"Generates a single base terrain level, excluding ore resources./"/"/"

    # 1. Calculate player proximity maps
    player_proximity_map = get_distance_map(player_position, static_params.map_size)
        .astype(jnp.float32)
    player_proximity_map_water = jnp.clip(player_proximity_map / config.
        player_proximity_map_water_strength, 0.0, config.
        player_proximity_map_water_max)
    player_proximity_map_mountain = jnp.clip(player_proximity_map / config.
        player_proximity_map_mountain_strength, 0.0, config.
```

```
        player_proximity_map_mountain_max)

    # 2. Define noise resolutions
    larger_res = (static_params.map_size[0] // 4, static_params.map_size[1] // 4)
    small_res = (static_params.map_size[0] // 16, static_params.map_size[1] // 16)
    x_res = (static_params.map_size[0] // 8, static_params.map_size[1] // 2)

    # 3. Generate base noise maps
    rng, water_rng, mountain_rng, path_rng, tree_rng = jax.random.split(rng, 5)
    water = generate_fractal_noise_2d(water_rng, static_params.map_size, small_res,
        octaves=1, override_angles=params.fractal_noise_angles[0]) +
        player_proximity_map_water - 1.0
    mountain = generate_fractal_noise_2d(mountain_rng, static_params.map_size,
        small_res, octaves=1, override_angles=params.fractal_noise_angles[1]) + 0.05
         + player_proximity_map_mountain - 1.0
    path_x = generate_fractal_noise_2d(path_rng, static_params.map_size, x_res,
        octaves=1, override_angles=params.fractal_noise_angles[2])
    tree_noise = generate_fractal_noise_2d(tree_rng, static_params.map_size,
        larger_res, octaves=1, override_angles=params.fractal_noise_angles[3])

    # 4. Progressively build the map from noise
    new_map = jnp.full(static_params.map_size, config.default_block)
    new_map = jnp.where(water > config.water_threshold, config.sea_block, new_map)
    sand_map = jnp.logical_and(water > config.sand_threshold, new_map != config.
        sea_block)
    new_map = jnp.where(sand_map, config.coast_block, new_map)
    new_map = jnp.where(mountain > 0.7, config.mountain_block, new_map)
    path = jnp.logical_and(mountain > 0.7, path_x > 0.8)
    new_map = jnp.where(path > 0.5, config.path_block, new_map)
    path_y = path_x.T
    path = jnp.logical_and(mountain > 0.7, path_y > 0.8)
    new_map = jnp.where(path > 0.5, config.path_block, new_map)
    caves = jnp.logical_and(mountain > 0.85, water > 0.4)
    new_map = jnp.where(caves > 0.5, config.inner_mountain_block, new_map)

    # 5. Add Trees
    rng, tree_placement_rng = jax.random.split(rng)
    tree = (tree_noise > config.tree_threshold_perlin) * jax.random.uniform(
        tree_placement_rng, shape=static_params.map_size) > config.
        tree_threshold_uniform
    tree = jnp.logical_and(tree, new_map == config.tree_requirement_block)
    new_map = jnp.where(tree, config.tree, new_map)

    # Add Ores
    def _add_ore(carry, index):
        rng, map = carry
        rng, _rng = jax.random.split(rng)
        ore_map = jnp.logical_and(
            map == config.ore_requirement_blocks[index],
            jax.random.uniform(_rng, static_params.map_size)
            < config.ore_chances[index],
        )
        map = jnp.where(ore_map, config.ores[index], map)

        return (rng, map), None

    rng, _rng = jax.random.split(rng)
    (_, new_map), _ = jax.lax.scan(_add_ore, (_rng, new_map), jnp.arange(5))

    # 6. Add Lava
    lava_map = jnp.logical_and(mountain > 0.85, tree_noise > 0.7)
    new_map = jnp.where(lava_map, config.lava, new_map)
```

```
# Add diamond if always_diamond flag is set
adding_diamond = jnp.logical_and(
    config.default_block == BlockType.GRASS.value,  # Hacky check for overworld
    params.always_diamond,
)
valid_diamond = (new_map.flatten() == BlockType.STONE.value).astype(jnp.float32)
rng, _rng = jax.random.split(rng)
diamond_index = jax.random.choice(
    _rng,
    jnp.arange(static_params.map_size[0] * static_params.map_size[1]),
    p=valid_diamond / valid_diamond.sum(),
)
diamond_position = jnp.array(
    [
        diamond_index // static_params.map_size[0],
        diamond_index % static_params.map_size[0],
    ]
)
diamond_replace_block = jax.lax.select(
    adding_diamond, BlockType.DIAMOND.value, BlockType.STONE.value
)
new_map = new_map.at[diamond_position[0], diamond_position[1]].set(
    diamond_replace_block)

# Light map
light_map = (
    jnp.ones(static_params.map_size, dtype=jnp.float32) * config.default_light
)

# Make sure player spawns on grass
new_map = new_map.at[player_position[0], player_position[1]].set(config.
    player_spawn)

item_map = jnp.zeros(static_params.map_size, dtype=jnp.int32)

valid_ladder_down = new_map.flatten() == config.valid_ladder
rng, _rng = jax.random.split(rng)
ladder_index = jax.random.choice(
    _rng,
    jnp.arange(static_params.map_size[0] * static_params.map_size[1]),
    p=valid_ladder_down,
)
ladder_down = jnp.array(
    [
        ladder_index // static_params.map_size[0],
        ladder_index % static_params.map_size[0],
    ]
)

item_map = item_map.at[ladder_down[0], ladder_down[1]].set(
    ItemType.LADDER_DOWN.value * config.ladder_down
    + item_map[ladder_down[0], ladder_down[1]] * (1 - config.ladder_down)
)

valid_ladder_up = new_map.flatten() == config.valid_ladder
rng, _rng = jax.random.split(rng)
ladder_index = jax.random.choice(
    _rng,
    jnp.arange(static_params.map_size[0] * static_params.map_size[1]),
    p=valid_ladder_up,
)
ladder_up = jnp.array(
    [
```

```
                ladder_index // static_params.map_size[0],
                ladder_index % static_params.map_size[0],
            ]
        )

        LIGHT_MAP_AROUND_LADDER = TORCH_LIGHT_MAP * (
            1 - config.default_light
        ) + config.default_light * jnp.ones((9, 9))

        light_map = jax.lax.dynamic_update_slice(
            light_map, LIGHT_MAP_AROUND_LADDER, ladder_up - jnp.array([4, 4])
        )

        z = jnp.array([[0.2, 0.7, 0.2], [0.7, 1, 0.7], [0.2, 0.7, 0.2]]) * (
            config.lava == BlockType.LAVA.value
        )
        light_map += jsp.signal.convolve(lava_map, z, mode="same")
        light_map = jnp.clip(light_map, 0.0, 1.0)

        item_map = item_map.at[ladder_up[0], ladder_up[1]].set(
            ItemType.LADDER_UP.value * config.ladder_up
            + item_map[ladder_up[0], ladder_up[1]] * (1 - config.ladder_up)
        )

        return new_map, item_map, light_map, ladder_down, ladder_up

def _generate_base_dungeon_level(
    rng: jax.Array, static_params: StaticEnvParams, config: DungeonConfig
) -> jnp.ndarray:
    /"/"/"Generates a single base dungeon level, excluding ore resources./"/"/"

    # 1. Setup
    chunk_size = 16
    world_chunk_width = static_params.map_size[0] // chunk_size
    world_chunk_height = static_params.map_size[1] // chunk_size
    room_occupancy_chunks = jnp.ones(world_chunk_width * world_chunk_height)
    num_rooms = 8
    min_room_size = 5
    max_room_size = 10

    rng, _rng = jax.random.split(rng)
    room_sizes = jax.random.randint(_rng, shape=(num_rooms, 2), minval=min_room_size
        , maxval=max_room_size)
    map = jnp.ones(static_params.map_size, dtype=jnp.int32) * BlockType.WALL.value
    padded_map = jnp.pad(map, max_room_size, constant_values=0)

    item_map = jnp.zeros(static_params.map_size, dtype=jnp.int32)
    padded_item_map = jnp.pad(item_map, max_room_size, constant_values=0)

    # 2. Add Rooms
    def _add_room(carry, room_index):
        block_map, item_map, room_occupancy_chunks, rng = carry
        rng, _rng = jax.random.split(rng)
        room_chunk = jax.random.choice(_rng, jnp.arange(world_chunk_width *
            world_chunk_height), p=room_occupancy_chunks)
        room_occupancy_chunks = room_occupancy_chunks.at[room_chunk].set(0)
        room_position = jnp.array([(room_chunk % world_chunk_height) * chunk_size, (
            room_chunk // world_chunk_height) * chunk_size]) + jnp.array([
            max_room_size, max_room_size])
        rng, _rng = jax.random.split(rng)
        room_position += jax.random.randint(_rng, (2,), minval=0, maxval=chunk_size
            - min_room_size)
```

```
slice = jax.lax.dynamic_slice(block_map, room_position, (max_room_size,
    max_room_size))
xs = jnp.expand_dims(jnp.arange(max_room_size), axis=-1).repeat(
    max_room_size, axis=-1)
ys = jnp.expand_dims(jnp.arange(max_room_size), axis=0).repeat(max_room_size
    , axis=0)
room_mask = jnp.logical_and(xs < room_sizes[room_index, 0], ys < room_sizes[
    room_index, 1])
slice = room_mask * BlockType.PATH.value + (1 - room_mask) * slice
block_map = jax.lax.dynamic_update_slice(block_map, slice, room_position)

# Torches in corner
item_map = item_map.at[room_position[0], room_position[1]].set(
    ItemType.TORCH.value
)
item_map = item_map.at[
    room_position[0] + room_sizes[room_index, 0] - 1, room_position[1]
].set(ItemType.TORCH.value)
item_map = item_map.at[
    room_position[0], room_position[1] + room_sizes[room_index, 1] - 1
].set(ItemType.TORCH.value)
item_map = item_map.at[
    room_position[0] + room_sizes[room_index, 0] - 1,
    room_position[1] + room_sizes[room_index, 1] - 1,
].set(ItemType.TORCH.value)

# Chest
rng, _rng = jax.random.split(rng)
chest_position = jax.random.randint(
    _rng,
    shape=(2,),
    minval=jnp.ones(2),
    maxval=room_sizes[room_index] - jnp.ones(2),
)
block_map = block_map.at[
    room_position[0] + chest_position[0], room_position[1] + chest_position
        [1]
].set(BlockType.CHEST.value)

# Fountain
rng, _rng, __rng = jax.random.split(rng, 3)
fountain_position = jax.random.randint(
    _rng,
    shape=(2,),
    minval=jnp.ones(2),
    maxval=room_sizes[room_index] - jnp.ones(2),
)
room_has_fountain = jax.random.uniform(__rng) > 0.5
fountain_block = (
    room_has_fountain * config.fountain_block
    + (1 - room_has_fountain)
    * block_map[
        room_position[0] + fountain_position[0],
        room_position[1] + fountain_position[1],
    ]
)
block_map = block_map.at[
    room_position[0] + fountain_position[0],
    room_position[1] + fountain_position[1],
].set(fountain_block)

return (block_map, item_map, room_occupancy_chunks, rng), room_position
```

```
    rng, _rng = jax.random.split(rng)
    (padded_map, padded_item_map, _, _), room_positions = jax.lax.scan(
        _add_room,
        (padded_map, padded_item_map, room_occupancy_chunks, _rng),
        jnp.arange(num_rooms),
    )

    def _add_path(carry, path_index):
        cmap, included_rooms_mask, rng = carry

        path_source = room_positions[path_index]

        rng, _rng = jax.random.split(rng)
        sink_index = jax.random.choice(
            _rng, jnp.arange(num_rooms), p=included_rooms_mask
        )
        path_sink = room_positions[sink_index]

        # Horizontal component
        entire_row = cmap[path_source[0]]
        path_indexes = jnp.arange(static_params.map_size[0] + 2 * max_room_size)
        path_indexes = path_indexes - path_source[1]
        horizontal_distance = path_sink[1] - path_source[1]
        path_indexes = path_indexes * jnp.sign(horizontal_distance)

        horizontal_mask = jnp.logical_and(
            path_indexes >= 0, path_indexes <= jnp.abs(horizontal_distance)
        )
        horizontal_mask = jnp.logical_and(
            horizontal_mask, jnp.sign(horizontal_distance)
        )
        horizontal_mask = jnp.logical_and(
            horizontal_mask, entire_row == BlockType.WALL.value
        )

        new_row = (
            horizontal_mask * BlockType.PATH.value + (1 - horizontal_mask) *
                entire_row
        )

        cmap = jax.lax.dynamic_update_slice(
            cmap,
            jnp.expand_dims(new_row, axis=0),
            path_source,
        )

        # Vertical component
        entire_col = cmap[:, path_sink[1]]
        path_indexes = jnp.arange(static_params.map_size[1] + 2 * max_room_size)
        path_indexes = path_indexes - path_source[0]
        vertical_distance = path_sink[0] - path_source[0]
        path_indexes = path_indexes * jnp.sign(vertical_distance)

        vertical_mask = jnp.logical_and(
            path_indexes >= 0, path_indexes <= jnp.abs(vertical_distance)
        )
        vertical_mask = jnp.logical_and(vertical_mask, jnp.sign(vertical_distance))

        vertical_mask = jnp.logical_and(
            vertical_mask, entire_col == BlockType.WALL.value
        )
```

```
    new_col = (
        vertical_mask * BlockType.PATH.value + (1 - vertical_mask) * entire_col
    )

    cmap = jax.lax.dynamic_update_slice(
        cmap,
        jnp.expand_dims(new_col, axis=-1),
        path_sink,
    )

    rng, _rng = jax.random.split(rng)
    included_rooms_mask = included_rooms_mask.at[path_index].set(True)
    return (cmap, included_rooms_mask, _rng), None

rng, _rng = jax.random.split(rng)
included_rooms_mask = jnp.zeros(num_rooms, dtype=bool).at[-1].set(True)
(padded_map, _, _), _, = jax.lax.scan(
    _add_path, (padded_map, included_rooms_mask, _rng), jnp.arange(0, num_rooms)
)

# Place special block in a random room
special_block_position = room_positions[0] + jnp.array([2, 2])
padded_map = padded_map.at[
    special_block_position[0], special_block_position[1]
].set(config.special_block)

map = padded_map[max_room_size:-max_room_size, max_room_size:-max_room_size]
item_map = padded_item_map[
    max_room_size:-max_room_size, max_room_size:-max_room_size
]

# Visual stuff
c_path_map = map != BlockType.WALL.value
z = jnp.array([[0, 1, 0], [1, 1, 1], [0, 1, 0]])
adj_path_map = jsp.signal.convolve(c_path_map, z, mode="same")
adj_path_map = adj_path_map > 0.5

rng, _rng = jax.random.split(rng)
rare_map = jax.random.choice(
    _rng,
    jnp.array([False, True]),
    static_params.map_size,
    p=jnp.array([0.9, 0.1]),
)

wall_map = (
    rare_map * BlockType.WALL_MOSS.value + (1 - rare_map) * BlockType.WALL.value
)

rare_map = jnp.logical_and(rare_map, map == BlockType.PATH.value)
rare_map = jnp.logical_and(rare_map, item_map == ItemType.NONE.value)
path_map = rare_map * config.rare_path_replacement_block + (1 - rare_map) * map

is_wall_map = jnp.logical_and(map == BlockType.WALL.value, adj_path_map)
is_darkness_map = jnp.logical_not(adj_path_map)
is_path_map = jnp.logical_not(jnp.logical_or(is_wall_map, is_darkness_map))

map = (
    is_path_map * path_map
    + is_wall_map * wall_map
    + is_darkness_map * BlockType.DARKNESS.value
)
```

```
    light_map = jnp.ones(static_params.map_size, dtype=jnp.float32)

    # Ladders
    valid_ladder_down = (map.flatten() == BlockType.PATH.value).astype(jnp.float32)
    rng, _rng = jax.random.split(rng)
    ladder_index = jax.random.choice(
        _rng,
        jnp.arange(static_params.map_size[0] * static_params.map_size[1]),
        p=valid_ladder_down / valid_ladder_down.sum(),
    )
    ladder_down_position = jnp.array(
        [
            ladder_index // static_params.map_size[0],
            ladder_index % static_params.map_size[0],
        ]
    )

    item_map = item_map.at[ladder_down_position[0], ladder_down_position[1]].set(
        ItemType.LADDER_DOWN.value
    )

    valid_ladder_up = map.flatten() == BlockType.PATH.value
    rng, _rng = jax.random.split(rng)
    ladder_index = jax.random.choice(
        _rng,
        jnp.arange(static_params.map_size[0] * static_params.map_size[1]),
        p=valid_ladder_up,
    )
    ladder_up_position = jnp.array(
        [
            ladder_index // static_params.map_size[0],
            ladder_index % static_params.map_size[0],
        ]
    )
    item_map = item_map.at[ladder_up_position[0], ladder_up_position[1]].set(
        ItemType.LADDER_UP.value
    )

    return map, item_map, light_map, ladder_down_position, ladder_up_position

def _place_randomly_pure_jit_safe(
    rng: jax.Array,
    current_map: jax.Array,
    block_type_value: int,
    n: int,
    on_block_values: jax.Array,
) -> jax.Array:
    /"/"/"
    A pure, JIT-safe function to place n blocks at random valid locations.
    /"/"/"
    # 1. Create a boolean mask of all valid placement locations.
    valid_mask = jnp.isin(current_map, on_block_values)

    # 2. Assign a random score (logit) to every cell in the entire map.
    random_logits = jax.random.uniform(rng, shape=current_map.shape)

    # 3. Mask out invalid locations by giving them a terrible score (-infinity).
    # This ensures they will never be chosen as one of the "best" random spots.
    masked_logits = jnp.where(valid_mask, random_logits, -jnp.inf)

    # 4. Find the flat indices of the `n` locations with the HIGHEST random scores.
    # We use argsort and take the last `n` elements, as argsort sorts ascending.
    # The shape of the output is always (n,), which is static and JIT-friendly.
```

```
    flat_indices = jnp.argsort(masked_logits.flatten())
    selected_flat_indices = flat_indices[-n:]

    # 5. Convert the selected 1D indices back to 2D coordinates.
    map_width = current_map.shape[1]
    rows = selected_flat_indices // map_width
    cols = selected_flat_indices % map_width

    # 6. Edge Case: What if n > num_valid_locations?
    # Some selected indices will point to locations with a score of -inf.
    # We must ensure we only place blocks on locations that were actually valid.
    # We create a final check to see if the chosen spots are in the original
        valid_mask.
    is_selection_valid = valid_mask[rows, cols]

    # The value to write: either the new block or the original block if the spot was
        invalid.
    value_to_set = jnp.where(
        is_selection_valid,
        block_type_value,
        current_map[rows, cols]
    )

    # 7. Perform the update. This now correctly handles all edge cases.
    new_map = current_map.at[rows, cols].set(value_to_set)

    return new_map

def _place_randomly_near_pure(
    rng: jax.Array,
    current_map: jax.Array,
    map_size: Tuple[int, int],
    block_type_value: int,
    n: int, # The number of blocks to place
    target_pos: Tuple[int, int],
    min_dist: int,
    max_dist: int,
    on_block_values: jax.Array,
) -> jax.Array:
    /"/"/"A pure, JIT-safe function to place n blocks near a target./"/"/"

    # 1. Create distance and surface masks.
    rows_grid, cols_grid = jnp.indices(map_size)
    dist_map = jnp.abs(rows_grid - target_pos[0]) + jnp.abs(cols_grid - target_pos
        [1])
    distance_mask = (dist_map >= min_dist) & (dist_map <= max_dist)
    surface_mask = jnp.isin(current_map, on_block_values)
    final_mask = distance_mask & surface_mask

    # 2. Assign a random score to every cell.
    random_logits = jax.random.uniform(rng, shape=map_size)

    # 3. Invalidate cells outside the target area by giving them a losing score.
    masked_logits = jnp.where(final_mask, random_logits, -jnp.inf)

    # 4. Find the flat indices of the n locations with the HIGHEST scores.
    # This is the key change from argmax to argsort.
    flat_indices = jnp.argsort(masked_logits.flatten())
    selected_flat_indices = flat_indices[-n:]

    # 5. Convert the selected 1D indices back to 2D coordinates (now plural).
    rows = selected_flat_indices // map_size[1]
```

```
        cols = selected_flat_indices % map_size[1]

        # 6. Check which of the selected spots were actually valid.
        is_selection_valid = final_mask[rows, cols]

        # 7. Set the new value only where the selection was valid.
        value_to_set = jnp.where(
            is_selection_valid,
            block_type_value,
            current_map[rows, cols]
        )

        # 8. Perform the batch update.
        new_map = current_map.at[rows, cols].set(value_to_set)

        return new_map

class WorldBuilder:
    /"/"/"A class to programmatically construct Craftax worlds./"/"/"

    def __init__(self, rng: jax.Array, static_params: StaticEnvParams, params:
        EnvParams):
        /"/"/"Initializes a blank multi-level world canvas./"/"/"
        self.static_params = static_params
        self.params = params

        # Initialize multi-level maps
        self.map = jnp.zeros(
            (static_params.num_levels, *static_params.map_size), dtype=jnp.int32
        )
        self.item_map = jnp.zeros(
            (static_params.num_levels, *static_params.map_size), dtype=jnp.int32
        )
        self.mob_map = jnp.zeros(
            (static_params.num_levels, *static_params.map_size), dtype=bool
        )

        # Player state
        self.player_position = jnp.array([static_params.map_size[0] // 2,
            static_params.map_size[1] // 2], dtype=jnp.int32)
        self.player_level = 0
        self.player_direction = Action.UP.value

        self.inventory = jax.tree_util.tree_map(
            lambda x, y: jax.lax.select(params.god_mode, x, y),
            get_new_full_inventory(),
            get_new_empty_inventory(),
        )

        self.player_dexterity = 1
        self.player_strength = 1
        self.player_intelligence = 1
        self.sword_enchantment = 0
        self.bow_enchantment = 0
        self.armour_enchantments = jnp.zeros(4, dtype=jnp.int32)
        self.learned_spells = jnp.array([False, False], dtype=bool)
        self.monsters_killed = jnp.zeros(static_params.num_levels, dtype=jnp.int32).
            at[0].set(10)

        # Initialize multi-level mob structures
        def _generate_empty_mobs(max_mobs):
            return Mobs(
```

```
                position=jnp.zeros((static_params.num_levels, max_mobs, 2), dtype=
                    jnp.int32),
                health=jnp.ones((static_params.num_levels, max_mobs), dtype=jnp.
                    float32),
                mask=jnp.zeros((static_params.num_levels, max_mobs), dtype=bool),
                attack_cooldown=jnp.zeros((static_params.num_levels, max_mobs),
                    dtype=jnp.int32),
                type_id=jnp.zeros((static_params.num_levels, max_mobs), dtype=jnp.
                    int32),
            )

        self.melee_mobs = _generate_empty_mobs(static_params.max_melee_mobs)
        self.ranged_mobs = _generate_empty_mobs(static_params.max_ranged_mobs)
        self.passive_mobs = _generate_empty_mobs(static_params.max_passive_mobs)

        # Other state components
        self.growing_plants_positions = jnp.zeros((static_params.max_growing_plants,
             2), dtype=jnp.int32)
        self.growing_plants_age = jnp.zeros(static_params.max_growing_plants, dtype=
            jnp.int32)
        self.growing_plants_mask = jnp.zeros(static_params.max_growing_plants, dtype
            =bool)

        self.generate_full_base_world(rng)

    def generate_full_base_world(self, rng: jax.Array):
        player_position = self.player_position

        # Generate smoothgens
        rngs = jax.random.split(rng, 7)
        rng, _rng = rngs[0], rngs[1:]

        smoothgens = jax.vmap(_generate_base_smoothworld_level, in_axes=(0, None,
            None, 0, None))(
            _rng, self.static_params, player_position, ALL_SMOOTHGEN_CONFIGS, self.
                params
        )

        # Generate dungeons
        rngs = jax.random.split(rng, 4)
        rng, _rng = rngs[0], rngs[1:]
        dungeons = jax.vmap(_generate_base_dungeon_level, in_axes=(0, None, 0))(
            _rng, self.static_params, ALL_DUNGEON_CONFIGS
        )

        # Splice smoothgens and dungeons in order of levels
        map, item_map, light_map, ladders_down, ladders_up = jax.tree_util.tree_map(
            lambda x, y: jnp.stack(
                (x[0], y[0], x[1], y[1], y[2], x[2], x[3], x[4], x[5]), axis=0
            ),
            smoothgens,
            dungeons,
        )

        # 3. Update the builder's state
        self.map, self.item_map, self.light_map, self.ladders_down, self.ladders_up
            = map, item_map, light_map, ladders_down, ladders_up

        print("Generated full 9-level base world without ores.")
        return self

    def set_starting_floor(self, level: int):
```

```
        /"/"/"Sets the player's starting level in the multi-level world./"/"/"
        self.player_level = level

        # --- Calculate final player position ---
        # If starting level > 0, use the up_ladder position for that level.
        # Otherwise, use the currently set player_position (default or from
           set_player_start).
        self.player_position = jax.lax.cond(
            self.player_level > 0,
            lambda: self.ladders_up[self.player_level], # Position on the 'up'
                ladder of the target level
            lambda: self.player_position,              # Default position (usually
                center of level 0)
        )
        return self

    def set_player_stats(
        self,
        dexterity: int = 1,
        strength: int = 1,
        intelligence: int = 1,
    ):
        /"/"/"
        Sets the player's starting stats (dexterity, strength, intelligence).

        The values are clamped to a safe range [1, 5] to ensure validity.
        /"/"/"
        # Clamp values to the defined range [1, 5]
        self.player_dexterity = jnp.clip(dexterity, 1, 5).astype(jnp.int32)
        self.player_strength = jnp.clip(strength, 1, 5).astype(jnp.int32)
        self.player_intelligence = jnp.clip(intelligence, 1, 5).astype(jnp.int32)

        return self

    def set_player_inventory(self, inventory_dict: dict):
        /"/"/"Sets the player's starting inventory./"/"/"
        self.inventory = self.inventory.replace(**inventory_dict)
        return self

    def set_weapon_enchantments(self, sword: int = 0, bow: int = 0):
        /"/"/"
        Sets the player's starting weapon enchantments.
        - 0: No enchantment
        - 1: Fire enchantment
        - 2: Ice enchantment
        /"/"/"
        self.sword_enchantment = jnp.clip(sword, 0, 2).astype(jnp.int32)
        self.bow_enchantment = jnp.clip(bow, 0, 2).astype(jnp.int32)
        return self

    def set_armour_enchantments(self, helmet: int = 0, chestplate: int = 0, leggings
    : int = 0, boots: int = 0):
        /"/"/"
        Sets the player's starting armour enchantments.
        - 0: No enchantment
        - 1: Fire resistance
        - 2: Ice resistance
        /"/"/"
        enchantments = jnp.array([
            jnp.clip(helmet, 0, 2),
            jnp.clip(chestplate, 0, 2),
            jnp.clip(leggings, 0, 2),
            jnp.clip(boots, 0, 2),
```

```
        ], dtype=jnp.int32)
        self.armour_enchantments = enchantments
        return self

    def set_learned_spells(self, fireball: bool = False, iceball: bool = False):
        /"/"/"
        Sets the spells the player has learned from the start.
        /"/"/"
        self.learned_spells = jnp.array([fireball, iceball], dtype=bool)
        return self

    def set_monsters_killed(self, level: int, count: int):
        /"/"/"
        Sets the number of monsters considered killed on a specific level.

        This is a more granural mechanism for unlocking ladders to specific
            subsequent levels.
        The count is clamped to a non-negative value.
        /"/"/"
        # Ensure count is not negative and update the specific level
        safe_count = jnp.maximum(0, count).astype(jnp.int32)
        self.monsters_killed = self.monsters_killed.at[level].set(safe_count)
        return self

    def place_block(self, level: int, block_type: BlockType, position: tuple):
        /"/"/"Places a single block on a specific level of the map./"/"/"
        self.map = self.map.at[level, position[0], position[1]].set(block_type.value
            )
        return self

    def fill_area(self, level: int, block_type: BlockType, top_left: tuple,
        bottom_right: tuple):
        /"/"/"Fills a rectangular area on a specific level with a block type./"/"/"
        xx, yy = jnp.meshgrid(jnp.arange(self.static_params.map_size[0]), jnp.arange
            (self.static_params.map_size[1]), indexing='ij')
        mask = ((xx >= top_left[0]) & (xx <= bottom_right[0]) & (yy >= top_left[1])
            & (yy <= bottom_right[1]))
        level_map = self.map[level]
        updated_level_map = jnp.where(mask, block_type.value, level_map)
        self.map = self.map.at[level].set(updated_level_map)
        return self

    def place_randomly(

        self,
        rng: jax.Array,
        level: int,
        block_type: BlockType,
        n: int = 1,
        on_blocks: List[BlockType] = None,
    ):
        /"/"/"
        Places n blocks at random locations on a specific level.
        /"/"/"
        if on_blocks is None:
            on_blocks = [BlockType.GRASS, BlockType.PATH, BlockType.FIRE_GRASS,
                BlockType.ICE_GRASS]
        on_block_values = jnp.array([b.value for b in on_blocks])

        # 1. Slice the specific level map we want to modify
        level_map = self.map[level]

        # 2. Call the pure helper function with the 2D map slice
```

```
            updated_level_map = _place_randomly_pure_jit_safe(
                rng, level_map, block_type.value, n, on_block_values
            )

            # 3. Update the slice in the main 3D map
            self.map = self.map.at[level].set(updated_level_map)
            return self

    def place_randomly_near(
        self,
        rng: jax.Array,
        level: int,
        block_type: BlockType,
        target_pos: tuple,
        min_dist: int,
        max_dist: int,
        n: int = 1,
        on_blocks: List[BlockType] = None,
    ):
        /"/"/"
        Places n blocks near a target position on a specific level.
        /"/"/"
        if on_blocks is None:
            on_blocks = [BlockType.GRASS, BlockType.PATH, BlockType.FIRE_GRASS,
                BlockType.ICE_GRASS]
        on_block_values = jnp.array([b.value for b in on_blocks])

        level_map = self.map[level]

        # --- Enforce minimum distance ---
        effective_min_dist = jnp.maximum(min_dist, 1)
        # Ensure max_dist is still >= effective_min_dist (optional but good practice
            )
        effective_max_dist = jnp.maximum(max_dist, effective_min_dist)
        # --- End enforcement ---

        updated_level_map = _place_randomly_near_pure(
            rng,
            level_map,
            self.static_params.map_size,
            block_type.value,
            n,
            target_pos,
            effective_min_dist,
            effective_max_dist,
            on_block_values,
        )

        self.map = self.map.at[level].set(updated_level_map)
        return self

    def add_mobs_randomly_near(
        self,
        rng: jax.Array,
        level: int,
        mob_name: str,
        type_id: int,
        n: int = 1,
        target_pos: jnp.ndarray = None,
        min_dist: int = 0,
        max_dist: int = 5,
        on_blocks: List[BlockType] = None,
    ):
```

```
/"/"/"
Adds n mobs of a given type near a target on a specific level.
/"/"/"
if target_pos is None:
    target_pos = self.player_position
if on_blocks is None:
    on_blocks = [BlockType.GRASS, BlockType.PATH, BlockType.SAND]

# This function's logic is nearly identical to add_mobs_randomly,
# but with an added distance mask. We can reuse the same adaptation pattern.
on_block_values = jnp.array([b.value for b in on_blocks])

def _add_specific_mob_near(mob_array, max_mobs, mob_idx):
    num_active_mobs = mob_array.mask[level].sum()
    available_slots = max_mobs - num_active_mobs

    # Create distance mask
    rows_grid, cols_grid = jnp.indices(self.static_params.map_size)
    dist_map = jnp.abs(rows_grid - target_pos[0]) + jnp.abs(cols_grid -
        target_pos[1])
    distance_mask = (dist_map >= min_dist) & (dist_map <= max_dist)

    surface_mask = jnp.isin(self.map[level], on_block_values)
    valid_mask = distance_mask & surface_mask & jnp.logical_not(self.mob_map
        [level])
    num_valid_locations = valid_mask.sum()
    n_to_place = jnp.minimum(n, jnp.minimum(available_slots,
        num_valid_locations)).astype(jnp.int32)

    # The rest of the logic is identical to add_mobs_randomly
    random_scores = jax.random.uniform(rng, shape=self.static_params.
        map_size)
    masked_scores = jnp.where(valid_mask, random_scores, -jnp.inf)
    flat_indices = jnp.argsort(masked_scores.flatten())
    potential_indices = flat_indices[-max_mobs:]
    rows, cols = potential_indices // self.static_params.map_size[1],
        potential_indices % self.static_params.map_size[1]
    potential_positions = jnp.stack([rows, cols], axis=1)
    inactive_indices = jnp.where(jnp.logical_not(mob_array.mask[level]),
        size=max_mobs, fill_value=-1)[0]

    def loop_body(i, carry):
        current_mob_array, current_mob_map_slice = carry
        slot_to_fill = inactive_indices[i]
        position_to_set = potential_positions[i]
        updated_array = current_mob_array.replace(
            position=current_mob_array.position.at[level, slot_to_fill].set(
                position_to_set),
            health=current_mob_array.health.at[level, slot_to_fill].set(
                MOB_TYPE_HEALTH_MAPPING[type_id, mob_idx]),
            mask=current_mob_array.mask.at[level, slot_to_fill].set(True),
            type_id=current_mob_array.type_id.at[level, slot_to_fill].set(
                type_id)
        )
        updated_mob_map_slice = current_mob_map_slice.at[position_to_set[0],
            position_to_set[1]].set(True)
        return updated_array, updated_mob_map_slice

    final_mob_array, final_mob_map_slice = jax.lax.fori_loop(0, n_to_place,
        loop_body, (mob_array, self.mob_map[level]))
    return final_mob_array, final_mob_map_slice

is_melee = mob_name == "melee"
```

```
        is_ranged = mob_name == "ranged"
        new_melee, melee_mob_map_slice = jax.lax.cond(is_melee, lambda:
            _add_specific_mob_near(self.melee_mobs, self.static_params.
            max_melee_mobs, MobType.MELEE.value), lambda: (self.melee_mobs, self.
            mob_map[level]))
        new_ranged, ranged_mob_map_slice = jax.lax.cond(is_ranged, lambda:
            _add_specific_mob_near(self.ranged_mobs, self.static_params.
            max_ranged_mobs, MobType.RANGED.value), lambda: (self.ranged_mobs, self.
            mob_map[level]))
        new_passive, passive_mob_map_slice = jax.lax.cond(jnp.logical_not(is_melee |
             is_ranged), lambda: _add_specific_mob_near(self.passive_mobs, self.
            static_params.max_passive_mobs, MobType.PASSIVE.value), lambda: (self.
            passive_mobs, self.mob_map[level]))

        self.melee_mobs, self.ranged_mobs, self.passive_mobs = new_melee, new_ranged
            , new_passive
        self.mob_map = self.mob_map.at[level].set(melee_mob_map_slice |
            ranged_mob_map_slice | passive_mob_map_slice)

        return self

    def place_adjacent_to_existing(self, rng: jax.Array, level: int, block_to_place:
         BlockType, target_block_type: BlockType, on_blocks: List[BlockType] = None)
        :
        /"/"/"
        Places a block adjacent to an existing block on a specific level.
        /"/"/"
        if on_blocks is None:
            on_blocks = [BlockType.GRASS, BlockType.PATH, BlockType.FIRE_GRASS,
                BlockType.ICE_GRASS]
        on_block_values = jnp.array([b.value for b in on_blocks])

        level_map = self.map[level]
        map_height, map_width = self.static_params.map_size

        target_mask = (level_map == target_block_type.value)
        adjacent_mask = jnp.zeros_like(level_map, dtype=bool)

        # Convolve a 4-connectivity kernel with the target mask
        kernel = jnp.array([[0, 1, 0], [1, 0, 1], [0, 1, 0]])
        adjacent_mask = jsp.signal.convolve(target_mask, kernel, mode="same") > 0

        surface_mask = jnp.isin(level_map, on_block_values)
        final_mask = adjacent_mask & surface_mask

        random_logits = jax.random.uniform(rng, shape=self.static_params.map_size)
        masked_logits = jnp.where(final_mask, random_logits, -jnp.inf)

        selected_flat_index = jnp.argmax(masked_logits.flatten())
        row, col = selected_flat_index // map_width, selected_flat_index % map_width

        is_valid = final_mask[row, col]
        new_value = jnp.where(is_valid, block_to_place.value, level_map[row, col])

        updated_level_map = level_map.at[row, col].set(new_value)
        self.map = self.map.at[level].set(updated_level_map)

        return self

    def build(self, rng: jax.Array) -> EnvState:
        /"/"/"
        Assembles the final, multi-level EnvState object from the builder's
            configuration.
```

```
/"/"/"
rng, state_rng, potion_rng = jax.random.split(rng, 3)

# Create empty projectile arrays, as they are not configured by the builder
def _create_projectiles(max_num):
    projectiles = self.melee_mobs.replace(
        position=jnp.zeros((self.static_params.num_levels, max_num, 2),
            dtype=jnp.int32),
        health=jnp.ones((self.static_params.num_levels, max_num), dtype=jnp.
            float32),
        mask=jnp.zeros((self.static_params.num_levels, max_num), dtype=bool)
            ,
        attack_cooldown=jnp.zeros((self.static_params.num_levels, max_num),
            dtype=jnp.int32),
        type_id=jnp.zeros((self.static_params.num_levels, max_num), dtype=
            jnp.int32)
    )
    directions = jnp.zeros((self.static_params.num_levels, max_num, 2),
        dtype=jnp.int32)
    return projectiles, directions

mob_projectiles, mob_projectile_directions = _create_projectiles(self.
    static_params.max_mob_projectiles)
player_projectiles, player_projectile_directions = _create_projectiles(self.
    static_params.max_player_projectiles)

state = EnvState(
    task_id=0,
    map=self.map,
    item_map=self.item_map,
    mob_map=self.mob_map,
    player_position=self.player_position,
    player_level=jnp.asarray(self.player_level, dtype=jnp.int32),
    player_direction=jnp.asarray(self.player_direction, dtype=jnp.int32),
    inventory=self.inventory,
    player_dexterity=jnp.asarray(self.player_dexterity, dtype=jnp.int32),
    player_strength=jnp.asarray(self.player_strength, dtype=jnp.int32),
    player_intelligence=jnp.asarray(self.player_intelligence, dtype=jnp.
        int32),
    sword_enchantment=jnp.asarray(self.sword_enchantment, dtype=jnp.int32),
    bow_enchantment=jnp.asarray(self.bow_enchantment, dtype=jnp.int32),
    armour_enchantments=self.armour_enchantments,
    learned_spells=self.learned_spells,
    melee_mobs=self.melee_mobs,
    ranged_mobs=self.ranged_mobs,
    passive_mobs=self.passive_mobs,
    growing_plants_positions=self.growing_plants_positions,
    growing_plants_age=self.growing_plants_age,
    growing_plants_mask=self.growing_plants_mask,
    monsters_killed=self.monsters_killed,
    player_health=jnp.asarray(9.0, dtype=jnp.float32),
    player_food=jnp.asarray(9, dtype=jnp.int32),
    player_drink=jnp.asarray(9, dtype=jnp.int32),
    player_energy=jnp.asarray(9, dtype=jnp.int32),
    player_mana=jnp.asarray(9, dtype=jnp.int32),
    player_xp=jnp.asarray(0, dtype=jnp.int32),
    is_sleeping=False,
    is_resting=False,
    player_recover=jnp.asarray(0.0, dtype=jnp.float32),
    player_hunger=jnp.asarray(0.0, dtype=jnp.float32),
    player_thirst=jnp.asarray(0.0, dtype=jnp.float32),
    player_fatigue=jnp.asarray(0.0, dtype=jnp.float32),
    player_recover_mana=jnp.asarray(0.0, dtype=jnp.float32),
```

```
            achievements=jnp.zeros((len(Achievement),), dtype=bool),
            light_map=self.light_map,
            down_ladders=self.ladders_down,
            up_ladders=self.ladders_up,
            chests_opened=jnp.zeros(self.static_params.num_levels, dtype=bool),
            mob_projectiles=mob_projectiles,
            mob_projectile_directions=mob_projectile_directions,
            player_projectiles=player_projectiles,
            player_projectile_directions=player_projectile_directions,
            potion_mapping=jax.random.permutation(potion_rng, jnp.arange(6)),
            boss_progress=jnp.asarray(0, dtype=jnp.int32),
            boss_timesteps_to_spawn_this_round=jnp.asarray(50, dtype=jnp.int32),
            light_level=jnp.asarray(calculate_light_level(0, self.params), dtype=jnp
                .float32),
            timestep=jnp.asarray(0, dtype=jnp.int32),
            state_rng=state_rng,
            fractal_noise_angles=self.params.fractal_noise_angles
        )
        return state
"""
```

## Module: Craftax Core (Injected into {CRAFTAX_CODE})

```python
context = """
=================================================
FILE: craftax/craftax/constants.py
=================================================
import os
import pathlib
from enum import Enum
import jax.numpy as jnp
import imageio.v3 as iio
import numpy as np
from PIL import Image
from craftax.craftax.util.maths_utils import get_distance_map
from craftax.environment_base.util import load_compressed_pickle,
    save_compressed_pickle

# GAME CONSTANTS
OBS_DIM = (9, 11)
assert OBS_DIM[0] % 2 == 1 and OBS_DIM[1] % 2 == 1
MAX_OBS_DIM = max(OBS_DIM)
BLOCK_PIXEL_SIZE_HUMAN = 64
BLOCK_PIXEL_SIZE_IMG = 16
BLOCK_PIXEL_SIZE_AGENT = 10
INVENTORY_OBS_HEIGHT = 4
TEXTURE_CACHE_FILE = os.path.join(
    pathlib.Path(__file__).parent.resolve(), "assets", "texture_cache.pbz2"
)

# ENUMS
class BlockType(Enum):
    INVALID = 0
    OUT_OF_BOUNDS = 1
    GRASS = 2
    WATER = 3
    STONE = 4
    TREE = 5
    WOOD = 6
    PATH = 7
```

```
        COAL = 8
        IRON = 9
        DIAMOND = 10
        CRAFTING_TABLE = 11
        FURNACE = 12
        SAND = 13
        LAVA = 14
        PLANT = 15
        RIPE_PLANT = 16
        WALL = 17
        DARKNESS = 18
        WALL_MOSS = 19
        STALAGMITE = 20
        SAPPHIRE = 21
        RUBY = 22
        CHEST = 23
        FOUNTAIN = 24
        FIRE_GRASS = 25
        ICE_GRASS = 26
        GRAVEL = 27
        FIRE_TREE = 28
        ICE_SHRUB = 29
        ENCHANTMENT_TABLE_FIRE = 30
        ENCHANTMENT_TABLE_ICE = 31
        NECROMANCER = 32
        GRAVE = 33
        GRAVE2 = 34
        GRAVE3 = 35
        NECROMANCER_VULNERABLE = 36

class ItemType(Enum):
        NONE = 0
        TORCH = 1
        LADDER_DOWN = 2
        LADDER_UP = 3
        LADDER_DOWN_BLOCKED = 4

class Action(Enum):
        NOOP = 0   #
        LEFT = 1   # a
        RIGHT = 2   # d
        UP = 3   # w
        DOWN = 4   # s
        DO = 5   # space
        SLEEP = 6   # tab
        PLACE_STONE = 7   # r
        PLACE_TABLE = 8   # t
        PLACE_FURNACE = 9   # f
        PLACE_PLANT = 10   # p
        MAKE_WOOD_PICKAXE = 11   # 1
        MAKE_STONE_PICKAXE = 12   # 2
        MAKE_IRON_PICKAXE = 13   # 3
        MAKE_WOOD_SWORD = 14   # 5
        MAKE_STONE_SWORD = 15   # 6
        MAKE_IRON_SWORD = 16   # 7
        REST = 17   # e
        DESCEND = 18   # >
        ASCEND = 19   # <
        MAKE_DIAMOND_PICKAXE = 20   # 4
        MAKE_DIAMOND_SWORD = 21   # 8
        MAKE_IRON_ARMOUR = 22   # y
        MAKE_DIAMOND_ARMOUR = 23   # u
        SHOOT_ARROW = 24   # i
```

```
    MAKE_ARROW = 25  # o
    CAST_FIREBALL = 26  # g
    CAST_ICEBALL = 27  # h
    PLACE_TORCH = 28  # j
    DRINK_POTION_RED = 29  # z
    DRINK_POTION_GREEN = 30  # x
    DRINK_POTION_BLUE = 31  # c
    DRINK_POTION_PINK = 32  # v
    DRINK_POTION_CYAN = 33  # b
    DRINK_POTION_YELLOW = 34  # n
    READ_BOOK = 35  # m
    ENCHANT_SWORD = 36  # k
    ENCHANT_ARMOUR = 37  # l
    MAKE_TORCH = 38  # [
    LEVEL_UP_DEXTERITY = 39  # ]
    LEVEL_UP_STRENGTH = 40  # -
    LEVEL_UP_INTELLIGENCE = 41  # =
    ENCHANT_BOW = 42  # ;

class ProjectileType(Enum):
    ARROW = 0
    DAGGER = 1
    FIREBALL = 2
    ICEBALL = 3
    ARROW2 = 4
    SLIMEBALL = 5
    FIREBALL2 = 6
    ICEBALL2 = 7

# FLOOR MECHANICS

FLOOR_MOB_MAPPING = jnp.array(
    [
        # (passive, melee, ranged)
        jnp.array([0, 0, 0]),  # Floor 0 (overworld)
        jnp.array([2, 2, 2]),  # Floor 1 (dungeon)
        jnp.array([1, 1, 1]),  # Floor 2 (gnomish mines)
        jnp.array([2, 3, 3]),  # Floor 3 (sewers)
        jnp.array([2, 4, 4]),  # Floor 4 (vaults)
        jnp.array([1, 5, 5]),  # Floor 5 (troll mines)
        jnp.array([1, 6, 6]),  # Floor 6 (fire)
        jnp.array([1, 7, 7]),  # Floor 7 (ice)
        jnp.array([0, 0, 0]),  # Floor 8 (boss)
    ],
    dtype=jnp.int32,
)

FLOOR_MOB_SPAWN_CHANCE = jnp.array(
    [
        # (passive, melee, ranged, melee-night)
        jnp.array([0.1, 0.02, 0.05, 0.1]),  # Floor 0 (overworld)
        jnp.array([0.1, 0.06, 0.05, 0.0]),  # Floor 1 (gnomish mines)
        jnp.array([0.1, 0.06, 0.05, 0.0]),  # Floor 2 (dungeon)
        jnp.array([0.1, 0.06, 0.05, 0.0]),  # Floor 3 (sewers)
        jnp.array([0.1, 0.06, 0.05, 0.0]),  # Floor 4 (vaults)
        jnp.array([0.1, 0.06, 0.05, 0.0]),  # Floor 5 (troll mines)
        jnp.array([0.1, 0.06, 0.05, 0.0]),  # Floor 6 (fire)
        jnp.array([0.0, 0.06, 0.05, 0.0]),  # Floor 7 (ice)
        jnp.array([0.1, 0.06, 0.05, 0.0]),  # Floor 8 (boss)
    ],
    dtype=jnp.float32,
)
```

```
# Path blocks, water, lava  (everything collides with solid blocks)
COLLISION_LAND_CREATURE = [False, True, True]
COLLISION_FLYING = [False, False, False]
COLLISION_AQUATIC = [True, False, True]
COLLISION_AMPHIBIAN = [False, False, True]

MOB_TYPE_COLLISION_MAPPING = jnp.array(
    [
        # (passive, melee, ranged, projectile)
        jnp.array(
            [
                COLLISION_LAND_CREATURE,
                COLLISION_LAND_CREATURE,
                COLLISION_LAND_CREATURE,
                COLLISION_FLYING,
            ]
        ),  # Floor 0 (overworld)
        jnp.array(
            [
                COLLISION_FLYING,
                COLLISION_LAND_CREATURE,
                COLLISION_LAND_CREATURE,
                COLLISION_FLYING,
            ]
        ),  # Floor 1 (gnomish mines)
        jnp.array(
            [
                COLLISION_LAND_CREATURE,
                COLLISION_LAND_CREATURE,
                COLLISION_LAND_CREATURE,
                COLLISION_FLYING,
            ]
        ),  # Floor 2 (dungeon)
        jnp.array(
            [
                COLLISION_LAND_CREATURE,
                COLLISION_AMPHIBIAN,
                COLLISION_LAND_CREATURE,
                COLLISION_FLYING,
            ]
        ),  # Floor 3 (sewers)
        jnp.array(
            [
                COLLISION_LAND_CREATURE,
                COLLISION_LAND_CREATURE,
                COLLISION_LAND_CREATURE,
                COLLISION_FLYING,
            ]
        ),  # Floor 4 (vaults)
        jnp.array(
            [
                COLLISION_LAND_CREATURE,
                COLLISION_LAND_CREATURE,
                COLLISION_AQUATIC,
                COLLISION_FLYING,
            ]
        ),  # Floor 5 (troll mines)
        jnp.array(
            [
                COLLISION_LAND_CREATURE,
                COLLISION_LAND_CREATURE,
                COLLISION_FLYING,
                COLLISION_FLYING,
```

```
                        ]
                    ),  # Floor 6 (fire)
                    jnp.array(
                        [
                            COLLISION_LAND_CREATURE,
                            COLLISION_LAND_CREATURE,
                            COLLISION_FLYING,
                            COLLISION_FLYING,
                        ]
                    ),  # Floor 7 (ice)
                    jnp.array(
                        [
                            COLLISION_LAND_CREATURE,
                            COLLISION_LAND_CREATURE,
                            COLLISION_LAND_CREATURE,
                            COLLISION_FLYING,
                        ]
                    ),  # Floor 8 (boss)
                ],
            dtype=jnp.int32,
        )

NO_DAMAGE = jnp.array([0, 0, 0])
MOB_TYPE_DAMAGE_MAPPING = jnp.array(
    [
        # (-, melee, -, projectile)
        [NO_DAMAGE, [2, 0, 0], NO_DAMAGE, [2, 0, 0]],  # zombie, arrow
        [NO_DAMAGE, [4, 0, 0], NO_DAMAGE, [4, 0, 0]],  # gnome, dagger
        [NO_DAMAGE, [3, 0, 0], NO_DAMAGE, [0, 3, 0]],  # orc, fireball
        [NO_DAMAGE, [5, 0, 0], NO_DAMAGE, [0, 0, 3]],  # lizard, iceball
        [NO_DAMAGE, [6, 0, 0], NO_DAMAGE, [5, 0, 0]],  # knight, arrow2
        [NO_DAMAGE, [6, 1, 1], NO_DAMAGE, [4, 3, 3]],  # troll, slimeball
        [NO_DAMAGE, [3, 5, 0], NO_DAMAGE, [3, 5, 0]],  # pigman, fireball2
        [NO_DAMAGE, [4, 0, 5], NO_DAMAGE, [4, 0, 5]],  # ice troll, iceball2
    ],
    dtype=jnp.float32,
)

MOB_TYPE_HEALTH_MAPPING = jnp.array(
    [
        # (passive, melee, ranged, -)
        jnp.array([3, 5, 3, 0]),  # Floor 0 (overworld)
        jnp.array([4, 7, 5, 0]),  # Floor 1 (gnomish mines)
        jnp.array([6, 9, 6, 0]),  # Floor 2 (dungeon)
        jnp.array([8, 11, 8, 0]),  # Floor 3 (sewers)
        jnp.array([0, 12, 12, 0]),  # Floor 4 (vaults)
        jnp.array([0, 20, 4, 0]),  # Floor 5 (troll mines)
        jnp.array([0, 20, 14, 0]),  # Floor 6 (fire)
        jnp.array([0, 24, 16, 0]),  # Floor 7 (ice)
        jnp.array([0, 0, 0, 0]),  # Floor 8 (boss)
    ],
    dtype=jnp.float32,
)

NO_DEFENSE = [0, 0, 0]
MOB_TYPE_DEFENSE_MAPPING = jnp.array(
    [
        # (passive, melee, ranged, -)
        jnp.array(
            [NO_DEFENSE, NO_DEFENSE, NO_DEFENSE, NO_DEFENSE]
        ),  # Floor 0 (overworld)
        jnp.array(
            [NO_DEFENSE, NO_DEFENSE, NO_DEFENSE, NO_DEFENSE]
```

```
        ),  # Floor 1 (gnomish mines)
        jnp.array(
            [NO_DEFENSE, NO_DEFENSE, NO_DEFENSE, NO_DEFENSE]
        ),  # Floor 2 (dungeon)
        jnp.array([NO_DEFENSE, NO_DEFENSE, NO_DEFENSE, NO_DEFENSE]),  # Floor 3 (
            sewers)
        jnp.array(
            [NO_DEFENSE, [0.5, 0, 0], [0.5, 0, 0], NO_DEFENSE]
        ),  # Floor 4 (vaults)
        jnp.array(
            [NO_DEFENSE, [0.2, 0, 0], [0.0, 0.0, 0.0], NO_DEFENSE]
        ),  # Floor 5 (troll mines)
        jnp.array(
            [NO_DEFENSE, [0.9, 1.0, 0.0], [0.9, 1.0, 0.0], NO_DEFENSE]
        ),  # Floor 6 (fire)
        jnp.array(
            [NO_DEFENSE, [0.9, 0.0, 1.0], [0.9, 0.0, 1.0], NO_DEFENSE]
        ),  # Floor 7 (ice)
        jnp.array([NO_DEFENSE, NO_DEFENSE, NO_DEFENSE, NO_DEFENSE]),  # Floor 8 (
            boss)
    ],
    dtype=jnp.float32,
)

RANGED_MOB_TYPE_TO_PROJECTILE_TYPE_MAPPING = jnp.array(
    [
        0,  # Skeleton --> Arrow
        0,  # Gnome archer --> Arrow
        2,  # Orc mage --> Fireball
        1,  # Kobold --> Dagger
        4,  # Knight archer --> Arrow2
        5,  # Deep thing --> Slime ball
        6,  # Fire elemental --> Fireball2
        7,  # Ice elemental --> Iceball2
    ]
)

# GAME MECHANICS
BOSS_FIGHT_EXTRA_DAMAGE = 0.5
BOSS_FIGHT_SPAWN_TURNS = 7

DIRECTIONS = jnp.concatenate(
    (
        jnp.array([[0, 0], [0, -1], [0, 1], [-1, 0], [1, 0]], dtype=jnp.int32),
        jnp.zeros((11, 2), dtype=jnp.int32),
    ),
    axis=0,
)

CLOSE_BLOCKS = jnp.array(
    [
        [0, -1],
        [0, 1],
        [-1, 0],
        [1, 0],
        [-1, -1],
        [-1, 1],
        [1, -1],
        [1, 1],
    ],
    dtype=jnp.int32,
)
```

```
# Can't walk through these
SOLID_BLOCKS = [
    BlockType.STONE.value,
    BlockType.TREE.value,
    BlockType.COAL.value,
    BlockType.IRON.value,
    BlockType.DIAMOND.value,
    BlockType.CRAFTING_TABLE.value,
    BlockType.FURNACE.value,
    BlockType.PLANT.value,
    BlockType.RIPE_PLANT.value,
    BlockType.WALL.value,
    BlockType.WALL_MOSS.value,
    BlockType.STALAGMITE.value,
    BlockType.RUBY.value,
    BlockType.SAPPHIRE.value,
    BlockType.CHEST.value,
    BlockType.FOUNTAIN.value,
    BlockType.FIRE_TREE.value,
    BlockType.ENCHANTMENT_TABLE_FIRE.value,
    BlockType.ENCHANTMENT_TABLE_ICE.value,
    BlockType.GRAVE.value,
    BlockType.GRAVE2.value,
    BlockType.GRAVE3.value,
    BlockType.NECROMANCER.value,
]

SOLID_BLOCK_MAPPING = jnp.array(
    [(block.value in SOLID_BLOCKS) for block in BlockType], dtype=bool
)

CAN_PLACE_ITEM_BLOCKS = [
    BlockType.GRASS.value,
    BlockType.SAND.value,
    BlockType.PATH.value,
    BlockType.FIRE_GRASS.value,
    BlockType.ICE_GRASS.value,
]

CAN_PLACE_ITEM_MAPPING = jnp.array(
    [(block.value in CAN_PLACE_ITEM_BLOCKS) for block in BlockType], dtype=bool
)

# ACHIEVEMENTS
class Achievement(Enum):
    COLLECT_WOOD = 0
    PLACE_TABLE = 1
    EAT_COW = 2
    COLLECT_SAPLING = 3
    COLLECT_DRINK = 4
    MAKE_WOOD_PICKAXE = 5
    MAKE_WOOD_SWORD = 6
    PLACE_PLANT = 7
    DEFEAT_ZOMBIE = 8
    COLLECT_STONE = 9
    PLACE_STONE = 10
    EAT_PLANT = 11
    DEFEAT_SKELETON = 12
    MAKE_STONE_PICKAXE = 13
    MAKE_STONE_SWORD = 14
    WAKE_UP = 15
    PLACE_FURNACE = 16
    COLLECT_COAL = 17
```

```
        COLLECT_IRON = 18
        COLLECT_DIAMOND = 19
        MAKE_IRON_PICKAXE = 20
        MAKE_IRON_SWORD = 21

        MAKE_ARROW = 22
        MAKE_TORCH = 23
        PLACE_TORCH = 24

        COLLECT_SAPPHIRE = 54
        COLLECT_RUBY = 59
        MAKE_DIAMOND_PICKAXE = 60
        MAKE_DIAMOND_SWORD = 25
        MAKE_IRON_ARMOUR = 26
        MAKE_DIAMOND_ARMOUR = 27

        ENTER_GNOMISH_MINES = 28
        ENTER_DUNGEON = 29
        ENTER_SEWERS = 30
        ENTER_VAULT = 31
        ENTER_TROLL_MINES = 32
        ENTER_FIRE_REALM = 33
        ENTER_ICE_REALM = 34
        ENTER_GRAVEYARD = 35

        DEFEAT_GNOME_WARRIOR = 36
        DEFEAT_GNOME_ARCHER = 37
        DEFEAT_ORC_SOLIDER = 38
        DEFEAT_ORC_MAGE = 39
        DEFEAT_LIZARD = 40
        DEFEAT_KOBOLD = 41
        DEFEAT_KNIGHT = 65
        DEFEAT_ARCHER = 66
        DEFEAT_TROLL = 42
        DEFEAT_DEEP_THING = 43
        DEFEAT_PIGMAN = 44
        DEFEAT_FIRE_ELEMENTAL = 45
        DEFEAT_FROST_TROLL = 46
        DEFEAT_ICE_ELEMENTAL = 47
        DAMAGE_NECROMANCER = 48
        DEFEAT_NECROMANCER = 49

        EAT_BAT = 50
        EAT_SNAIL = 51

        FIND_BOW = 52
        FIRE_BOW = 53

        LEARN_FIREBALL = 55
        CAST_FIREBALL = 56
        LEARN_ICEBALL = 57
        CAST_ICEBALL = 58

        OPEN_CHEST = 61
        DRINK_POTION = 62
        ENCHANT_SWORD = 63
        ENCHANT_ARMOUR = 64

INTERMEDIATE_ACHIEVEMENTS = [
        Achievement.COLLECT_SAPPHIRE.value,
        Achievement.COLLECT_RUBY.value,
        Achievement.MAKE_DIAMOND_PICKAXE.value,
        Achievement.MAKE_DIAMOND_SWORD.value,
```

```python
        Achievement.MAKE_IRON_ARMOUR.value,
        Achievement.MAKE_DIAMOND_ARMOUR.value,
        Achievement.ENTER_GNOMISH_MINES.value,
        Achievement.ENTER_DUNGEON.value,
        Achievement.DEFEAT_GNOME_WARRIOR.value,
        Achievement.DEFEAT_GNOME_ARCHER.value,
        Achievement.DEFEAT_ORC_SOLIDER.value,
        Achievement.DEFEAT_ORC_MAGE.value,
        Achievement.EAT_BAT.value,
        Achievement.EAT_SNAIL.value,
        Achievement.FIND_BOW.value,
        Achievement.FIRE_BOW.value,
        Achievement.OPEN_CHEST.value,
        Achievement.DRINK_POTION.value,
]

VERY_ADVANCED_ACHIEVEMENTS = [
        Achievement.ENTER_FIRE_REALM.value,
        Achievement.ENTER_ICE_REALM.value,
        Achievement.ENTER_GRAVEYARD.value,
        Achievement.DEFEAT_PIGMAN.value,
        Achievement.DEFEAT_FIRE_ELEMENTAL.value,
        Achievement.DEFEAT_FROST_TROLL.value,
        Achievement.DEFEAT_ICE_ELEMENTAL.value,
        Achievement.DAMAGE_NECROMANCER.value,
        Achievement.DEFEAT_NECROMANCER.value,
]

def achievement_mapping(achievement_value):
    if achievement_value <= 24:
        return 1
    elif achievement_value in INTERMEDIATE_ACHIEVEMENTS:
        return 3
    elif achievement_value in VERY_ADVANCED_ACHIEVEMENTS:
        return 8
    else:
        return 5

ACHIEVEMENT_REWARD_MAP = jnp.array(
    [achievement_mapping(i) for i in range(len(Achievement))]
)

LEVEL_ACHIEVEMENT_MAP = jnp.array(
    [
        0,
        Achievement.ENTER_DUNGEON.value,
        Achievement.ENTER_GNOMISH_MINES.value,
        Achievement.ENTER_SEWERS.value,
        Achievement.ENTER_VAULT.value,
        Achievement.ENTER_TROLL_MINES.value,
        Achievement.ENTER_FIRE_REALM.value,
        Achievement.ENTER_ICE_REALM.value,
        Achievement.ENTER_GRAVEYARD.value,
    ]
)

MOB_ACHIEVEMENT_MAP = jnp.array(
    [
        # Passive
        [
            Achievement.EAT_COW.value,
            Achievement.EAT_BAT.value,
            Achievement.EAT_SNAIL.value,
```

```
                0,
                0,
                0,
                0,
                0,
            ],
            # Melee
            [
                Achievement.DEFEAT_ZOMBIE.value,
                Achievement.DEFEAT_GNOME_WARRIOR.value,
                Achievement.DEFEAT_ORC_SOLIDER.value,
                Achievement.DEFEAT_LIZARD.value,
                Achievement.DEFEAT_KNIGHT.value,
                Achievement.DEFEAT_TROLL.value,
                Achievement.DEFEAT_PIGMAN.value,
                Achievement.DEFEAT_FROST_TROLL.value,
            ],
            # Ranged
            [
                Achievement.DEFEAT_SKELETON.value,
                Achievement.DEFEAT_GNOME_ARCHER.value,
                Achievement.DEFEAT_ORC_MAGE.value,
                Achievement.DEFEAT_KOBOLD.value,
                Achievement.DEFEAT_ARCHER.value,
                Achievement.DEFEAT_DEEP_THING.value,
                Achievement.DEFEAT_FIRE_ELEMENTAL.value,
                Achievement.DEFEAT_ICE_ELEMENTAL.value,
            ],
    ]
)

# PRE-COMPUTATION
TORCH_LIGHT_MAP = get_distance_map(jnp.array([4, 4]), (9, 9))
TORCH_LIGHT_MAP /= 5.0
TORCH_LIGHT_MAP = jnp.clip(1 - TORCH_LIGHT_MAP, 0.0, 1.0)

================================================
FILE: craftax/craftax/craftax_state.py
================================================
from dataclasses import dataclass
from typing import Tuple, Any

import jax
from flax import struct
import jax.numpy as jnp

@struct.dataclass
class Inventory:
    wood: int
    stone: int
    coal: int
    iron: int
    diamond: int
    sapling: int
    pickaxe: int
    sword: int
    bow: int
    arrows: int
    armour: jnp.ndarray
    torches: int
    ruby: int
    sapphire: int
    potions: jnp.ndarray
```

```
    books: int

@struct.dataclass
class Mobs:
    position: jnp.ndarray
    health: jnp.ndarray
    mask: jnp.ndarray
    attack_cooldown: jnp.ndarray
    type_id: jnp.ndarray

@struct.dataclass
class EnvState:
    map: jnp.ndarray
    item_map: jnp.ndarray
    mob_map: jnp.ndarray
    light_map: jnp.ndarray
    down_ladders: jnp.ndarray
    up_ladders: jnp.ndarray
    chests_opened: jnp.ndarray
    monsters_killed: jnp.ndarray

    player_position: jnp.ndarray
    player_level: int
    player_direction: int

    # Intrinsics
    player_health: float
    player_food: int
    player_drink: int
    player_energy: int
    player_mana: int
    is_sleeping: bool
    is_resting: bool

    # Second order intrinsics
    player_recover: float
    player_hunger: float
    player_thirst: float
    player_fatigue: float
    player_recover_mana: float

    # Attributes
    player_xp: int
    player_dexterity: int
    player_strength: int
    player_intelligence: int

    inventory: Inventory

    melee_mobs: Mobs
    passive_mobs: Mobs
    ranged_mobs: Mobs

    mob_projectiles: Mobs
    mob_projectile_directions: jnp.ndarray
    player_projectiles: Mobs
    player_projectile_directions: jnp.ndarray

    growing_plants_positions: jnp.ndarray
    growing_plants_age: jnp.ndarray
    growing_plants_mask: jnp.ndarray

    potion_mapping: jnp.ndarray
```

```
    learned_spells: jnp.ndarray

    sword_enchantment: int
    bow_enchantment: int
    armour_enchantments: jnp.ndarray

    boss_progress: int
    boss_timesteps_to_spawn_this_round: int

    light_level: float

    achievements: jnp.ndarray

    state_rng: Any

    timestep: int

    fractal_noise_angles: tuple[int, int, int, int] = (None, None, None, None)

@struct.dataclass
class EnvParams:
    max_timesteps: int = 100000
    day_length: int = 300

    always_diamond: bool = False

    mob_despawn_distance: int = 14
    max_attribute: int = 5

    god_mode: bool = False

    fractal_noise_angles: tuple[int, int, int, int] = (None, None, None, None)

@struct.dataclass
class StaticEnvParams:
    map_size: Tuple[int, int] = (48, 48)
    num_levels: int = 9

    # Mobs
    max_melee_mobs: int = 3
    max_passive_mobs: int = 3
    max_growing_plants: int = 10
    max_ranged_mobs: int = 2
    max_mob_projectiles: int = 3
    max_player_projectiles: int = 3
"""
```

### Module: Mob Definitions (Injected into {MOBS})

```
context = """
| Name           | mob_name | type_id | level(s)  |
|----------------|----------|---------|-----------|
| Cow            | passive  | 0       | [0]       |
| Bat            | passive  | 1       | [2, 5, 6] |
| Snail          | passive  | 2       | [1, 3, 4] |
| Zombie         | melee    | 0       | [0]       |
| Gnome Warrior  | melee    | 1       | [2]       |
| Orc Soldier    | melee    | 2       | [1]       |
| Lizard         | melee    | 3       | [3]       |
| Knight         | melee    | 4       | [4]       |
```

```
| Troll         | melee   | 5       | [5]       |
| Pigman        | melee   | 6       | [6]       |
| Frost Troll   | melee   | 7       | [7]       |
| Skeleton      | ranged  | 0       | [0]       |
| Gnome Archer  | ranged  | 1       | [2]       |
| Orc Mage      | ranged  | 2       | [1]       |
| Kobold        | ranged  | 3       | [3]       |
| Knight Archer | ranged  | 4       | [4]       |
| Deep Thing    | ranged  | 5       | [5]       |
| Fire Elemental| ranged  | 6       | [6]       |
| Ice Elemental | ranged  | 7       | [7]       |
""".strip()
```

### B.2.3. FEW-SHOT EXAMPLES

We provide the model with valid examples of 'Description → Code' pairs to enforce the correct class structure and API usage. Below is one representative example used in the prompt. All the examples together form $\{e\}_{i=1}^{n}$.

---

**Few-Shot Example: Collect Coal**

```python
import jax
from craftax.craftax.constants import Achievement, BlockType
from craftax.craftax.craftax_state import EnvParams, StaticEnvParams

from minicraftax.craftax_state import EnvState, TaskParams
from minicraftax.tasks.base_task import BaseTask
from minicraftax.world_builder import WorldBuilder

class Env(BaseTask):
        """Objective: Collect coal.
        Description: The player must achieve the 'COLLECT_COAL' achievement. The
            player starts on Floor 0 (the overworld) with a wooden pickaxe and sword
            . The world is a standard procedural overworld with 5 coal blocks placed
             4-8 tiles from the player's start. Mobs and needs are enabled but with
            easier settings.
        Relevant Achievements: COLLECT_COAL
        Completed Achievements: MAKE_WOOD_PICKAXE, MAKE_WOOD_SWORD
        World:
        - Player: Starts on floor 0 with a wooden pickaxe and wooden sword ('{"
            pickaxe": 1, "sword": 1}').
        - Map: 5 'COAL' blocks are placed randomly on 'GRASS' or 'STONE' within 4-8
            (Manhattan distance) tiles of the player. 3 'COW' (passive mob type_id
            =0) are placed 4-8 tiles away.
        - Mechanics: "needs_depletion_multiplier = 0.5", "passive_spawn_multiplier =
             1.0", "melee_spawn_multiplier = 0.2", "ranged_spawn_multiplier = 0.2"
        """

        def __init__(self, static_params: StaticEnvParams, params: EnvParams):
                super().__init__(static_params, params)
                self.relevant_achievements = [Achievement.COLLECT_COAL]
                self.completed_achievements = [Achievement.MAKE_WOOD_PICKAXE,
                    Achievement.MAKE_WOOD_SWORD]
                self.label = "COLLECT_COAL"

        def get_task_params(self) -> TaskParams:
                """Return custom parameters for this task."""
                return TaskParams(
                        passive_spawn_multiplier=1.0,  # Enable random cow spawns
                        melee_spawn_multiplier=0.2,  # Enable zombie spawns
```

```
                        ranged_spawn_multiplier=0.2,  # Enable skeleton spawns
                        needs_depletion_multiplier=0.5,  # Needs are on, but slow
            )

    def generate_world(self, rng: jax.Array) -> EnvState:
            """Generates the world for the task."""
            rng, build_rng, placement_rng, cow_rng = jax.random.split(rng, 4)

            builder = WorldBuilder(build_rng, self.static_params, self.params)

            builder.set_starting_floor(0)

            # --- ADDED SCAFFOLDING ---
            # 1. Give prerequisite pickaxe and a sword for safety
            builder.set_player_inventory({"pickaxe": 1, "sword": 1})

            # 2. Place cows as a food source
            builder.add_mobs_randomly_near(
                    cow_rng,
                    level=0,
                    mob_name="passive",
                    type_id=0,  # type_id 0 is Cow
                    n=3,
                    target_pos=builder.player_position,
                    min_dist=4,
                    max_dist=8,
                    on_blocks=[BlockType.GRASS, BlockType.PATH],
            )
            # --- END SCAFFOLDING ---

            # Place 5 coal blocks near the player on level 0
            builder.place_randomly_near(
                    placement_rng,
                    level=0,
                    block_type=BlockType.COAL,
                    target_pos=builder.player_position,
                    min_dist=4,
                    max_dist=8,
                    n=5,
                    on_blocks=[BlockType.GRASS, BlockType.STONE],
            )

            return builder.build(rng)
```

### B.2.4. USER PROMPT (CODE REQUEST)

The final trigger for the code generation step. The task description placeholder is populated by the latent description $h$ outputted from the previous inference step.

---

**User Prompt: Code Generation**

```
user_prompt = """
Your goal is to implement the task described below as a complete and correct Python
    file, following all instructions from the system prompt. CRITICAL: DO NOT forget
     the self.label.

## 1. TASK TO IMPLEMENT

### Task Description (for the docstring):
<description>
```

```
{TASK_DESCRIPTION}
</description>

## 2. CODE EXAMPLES

Here are some examples of other correctly implemented tasks. Use them as a reference
    for style and structure.

<examples>
{CODE_EXAMPLES}
</examples>

Now, generate the complete Python file for the new task.
"""
```

## B.3. Open-Loop Ablation (DiCode-OL)

In the Open-Loop ablation, the curriculum feedback loop is removed. The model generates tasks based on the static environment description alone, without access to the parent level ($\lambda_p$) or the agent's current performance profile ($\text{perf}_p, \text{perf}_{\text{target}}$).

### B.3.1. MODIFIED PROMPTS

---

**System Prompt: Open-Loop Generator**

```
system_prompt = """
You are an expert curriculum designer for reinforcement learning agents. Your job is
    to generate a new training task for the agent. You must generate a new,
    creative challenge that helps the agent solve the full ORIGINAL Craftax game.

===========================
CRITICAL: YOUR ROLE & OBJECTIVE
===========================
You are generating TRAINING TASKS for MiniCraftax to improve the agent's performance
    on ORIGINAL Craftax.

Core objective (most important):
- Maximize downstream competence on ORIGINAL Craftax (global progression: unlocking
    new floors, survival loops, combat viability, key transitions).

System dynamics you must account for:
- Many generated tasks will be trained only briefly and may never be used again if
    they underperform.
- If a task is too hard or bundles multiple fragile requirements at once, it is
    likely to fail and be discarded.
- Therefore, prefer tasks that apply focused, learnable pressure on a randomly
    chosen potential bottleneck capability.

How to use initial state (very important):
- Initial state is a tool to compress away prerequisites required to reach your
    chosen target capability.
- If you choose a target that exists in a later-game context (e.g., later floor),
    initialize inventory/tools/resources in a way consistent with "an agent that
    reached here competently," so training focuses on the NEW target skill.
- Avoid tasks that require going backwards to earlier floors for basic prerequisites
    , unless backward travel/navigation is explicitly the skill being trained.

Task design preferences (soft preferences, not hard rules):
- Prefer "thin-slice" tasks: 1 randomly chosen primary bottleneck capability +
    optional 1 randomly chosen supporting sub-skill.
```

```
===========================
CRITICAL: YOUR DESIGN PHILOSOPHY
===========================
1. **Rewards are UNIVERSAL:** The agent is rewarded for **ALL** achievements it
    finds, at any time, in any task.
2. **Goals are for TERMINATION:** The `Relevant Achievements` list you select **ONLY
    ** defines the task's `is_terminal` and `is_success` conditions. This is the "
    practice goal" you are forcing the agent to complete.
3. **Environment and Mechanics:** You control the initial world generation and a few
     constants that control game mechanics to control difficulty.

===========================
1. KNOWLEDGE BASE (IMMUTABLE RULES)
===========================
You have access to the following information about the full Craftax game logic.
<game_rules>
### 1. Core Definitions
{CONSTANTS}

### 2. Mob Definitions
{MOBS}

### 3. Game Mechanics
{GAME_MECHANICS}

### 4. World Generation
{WORLD_GEN}
</game_rules>

===========================
2. YOUR TOOLKIT (MUTABLE API)
===========================
To generate tasks, you must use the following API to modify the world and mechanics.
<api_docs>
{API_DOCS}
</api_docs>

===========================
GUIDING PRINCIPLE: TARGETED CAPABILITY SAMPLING
===========================
Your job is to select a meaningful slice of the game to train.
- Randomly, pick a specific mechanic, transition, or survival loop from the Craftax
    game logic in the KNOWLEDGE BASE section.
- Construct a task that isolates this mechanic.

Avoid "backtracking tasks" by default: if you start the agent in a later context (e.
    g., floor 1), provide the prerequisites via initial state and mark them as
    Completed Achievements.

## 3. OUTPUT FORMAT

**CRITICAL RULE: MANAGING ACHIEVEMENT LISTS**
You must separate achievements into two strictly defined lists:
1. `Relevant Achievements`: Goals the agent **must actively achieve** during the
    episode to succeed.
2. `Completed Achievements`: Goals implicitly satisfied by the initial `World` state
     (e.g., starting inventory) which the agent **cannot or should not do again**.

*Example:* If the `World` setup provides a `wood_pickaxe`:
- `MAKE_WOOD_PICKAXE` goes into `Completed Achievements`.

**SPECIFICITY REQUIREMENT (NON-NEGOTIABLE)**
```

```
The task description must be detailed enough for another LLM to implement it in code
    without guessing.
- Use precise coordinates, quantities, and block types.
- For mobs, always specify both `mob_name` and `type_id`.
- Avoid vague language (e.g., "near", "some", "a few", "around the player").
- If a detail matters for difficulty or reachability, it must be explicitly stated.

Your response MUST STRICTLY be in the following format. Do NOT include any other
    text or explanations outside of these tags.

<reasoning>
**Justification for New Task:** Provide a detailed analysis of the task design and a
    justification for why this specific slice of gameplay is valuable for ORIGINAL
    Craftax.

Specifically, address the following points:

1) **Target Capability Selection:**
   - What specific capability or game transition have you randomly chosen to target?
   - Why is this a valuable skill for the agent to practice in isolation?

2) **Scaffolding & Backtracking Avoidance (Start-state design):**
   - Explain how the initial state prevents unnecessary backtracking.
   - If starting in later context (e.g., floor 1), state what inventory/tools you
       provide to match a competent arrival, and which achievements move to
       Completed.

3) **Final Consistency Check:**
   - Task Relevant Achievements: [your list]
   - Task Completed Achievements: [your list]
   - "Thin-slice" check: Does the task focus on a specific interaction rather than a
       full game run? [YES]
   - Backtracking check: Does the task avoid requiring earlier-floor crafting for
       basic prerequisites unless intended? [YES]
</reasoning>

<docstring>
[The full, multi-line natural language description of the new task, following the
    standardized template below, goes here.]

Objective: [A concise sentence describing the skill the agent should learn.]
Description: [A detailed description of the task, including the objective, the world
    , the starting floor, the inventory and the mechanics.]
Relevant Achievements: [The achievements that are relevant to the task.]
Completed Achievements: [The achievements implicitly satisfied by the initial World
    state (e.g. starting inventory) which the agent cannot/should not do again.]
World:
- Player: [Starting floor and inventory.]
- Map: [A list of all block modifications made to the default 9-level map. This
    section is for *block* changes made with the WorldBuilder.]
- Mechanics: [List of non-default TaskParams values, using exact API parameter names
    (e.g., "mob_health_multiplier = 2.0").]
</docstring>
"""
```

## User Prompt: Open-Loop Generator

```
user_prompt = """
**REMINDER: You are generating a new, creative task description, NOT code.**
```

```
**Your output should be a reasoning section followed by a detailed docstring for the
    new task.**
"""
```

## C. Qualitative Analysis

### C.1. Curriculum Case Studies

We provide a detailed examination of four representative levels from the generated curriculum (referenced in Figure 4). These examples illustrate how DiCode progressively scaffolds complexity, from basic survival to deep exploration. Note that the descriptions are the docstrings of the Python classes.

---

**Level 112**

```python
import jax
from craftax.craftax.constants import Achievement, BlockType
from craftax.craftax.craftax_state import EnvParams, StaticEnvParams

from minicraftax.craftax_state import EnvState, TaskParams
from minicraftax.tasks.base_task import BaseTask
from minicraftax.world_builder import WorldBuilder

class Env(BaseTask):
    """Objective: Craft iron armor using guaranteed resources and descend to the
        dungeon floor to complete the transition sequence.
    Description: The player must craft iron armor (consuming exactly 3 iron and 3
        coal), then descend to Floor 1 (dungeon) via the ladder. The agent starts on
        Floor 0 (overworld) with precisely 3 iron, 3 coal, and a stone pickaxe (
        tier 2) in inventory. A crafting table is placed at coordinates (24, 25) and
        a furnace at (24, 26) on the same floor. An iron ore deposit is guaranteed
        at coordinates (28, 28) to (29, 29) to eliminate exploration randomness for
        iron collection. The ladder down on Floor 0 is pre-unlocked (monsters_killed
        =0) to focus training on armor crafting and descent mechanics. Moderate mob
        spawns (zombies and skeletons) encourage armor usage before descent, with
        melee_spawn_multiplier reduced to 0.3 to prevent overwhelming pressure
        during the new dependency.
    Relevant Achievements: MAKE_IRON_ARMOUR, ENTER_DUNGEON
    Completed Achievements: COLLECT_WOOD, COLLECT_STONE, COLLECT_COAL, COLLECT_IRON,
        MAKE_STONE_PICKAXE, PLACE_FURNACE, PLACE_TABLE
    World:
    - Player: Starts on floor 0 with inventory: {"wood": 0, "stone": 0, "coal": 3, "
        iron": 3, "pickaxe": 2}.
    - Map:
      - Crafting table placed at (24, 25) on floor 0.
      - Furnace placed at (24, 26) on floor 0.
      - Iron blocks fill area from (28, 28) to (29, 29) on floor 0.
      - Ladder down on floor 0 is pre-unlocked (monsters_killed[0] = 0).
    - Mechanics:
      - needs_depletion_multiplier = 1.0
      - melee_spawn_multiplier = 0.3
      - ranged_spawn_multiplier = 0.1
      - monsters_killed_to_clear_level = 0
    """

    def __init__(self, static_params: StaticEnvParams, params: EnvParams):
        super().__init__(static_params, params)
        self.relevant_achievements = [
            Achievement.MAKE_IRON_ARMOUR,
```

```
            Achievement.ENTER_DUNGEON,
        ]
        self.completed_achievements = [
            Achievement.COLLECT_WOOD,
            Achievement.COLLECT_STONE,
            Achievement.COLLECT_COAL,
            Achievement.COLLECT_IRON,
            Achievement.MAKE_STONE_PICKAXE,
            Achievement.PLACE_FURNACE,
            Achievement.PLACE_TABLE,
        ]
        self.label = "MAKE_IRON_ARMOUR, ENTER_DUNGEON"

    def get_task_params(self) -> TaskParams:
        """Return custom parameters for this task."""
        return TaskParams(
            needs_depletion_multiplier=1.0,
            melee_spawn_multiplier=0.3,
            ranged_spawn_multiplier=0.1,
            monsters_killed_to_clear_level=0,
        )

    def generate_world(self, rng: jax.Array) -> EnvState:
        """Generates the world for the task."""
        rng, build_rng = jax.random.split(rng)

        builder = WorldBuilder(build_rng, self.static_params, self.params)

        builder.set_starting_floor(0)

        # Set player inventory with 3 coal, 3 iron, and stone pickaxe (tier 2)
        builder.set_player_inventory({
            "wood": 0,
            "stone": 0,
            "coal": 3,
            "iron": 3,
            "pickaxe": 2
        })

        # Place crafting table at (24, 25) on floor 0
        builder.place_block(0, BlockType.CRAFTING_TABLE, (24, 25))

        # Place furnace at (24, 26) on floor 0
        builder.place_block(0, BlockType.FURNACE, (24, 26))

        # Fill area from (28, 28) to (29, 29) with iron blocks
        builder.fill_area(0, BlockType.IRON, (28, 28), (29, 29))

        # Set monsters killed count for floor 0 to 0 to unlock ladder down
            immediately
        builder.set_monsters_killed(0, 0)

        return builder.build(rng)
```

## Level 143

```
import jax
from craftax.craftax.constants import Achievement, BlockType
from craftax.craftax.craftax_state import EnvParams, StaticEnvParams

from minicraftax.craftax_state import EnvState, TaskParams
```

```
from minicraftax.tasks.base_task import BaseTask
from minicraftax.world_builder import WorldBuilder

class Env(BaseTask):
    """Objective: Generalize iron armor crafting to realistic exploration conditions
        by requiring natural resource gathering and workstation placement while
        maintaining dungeon descent capability.
    Description: The player must craft iron armor through natural resource
        acquisition and workstation placement, then descend to Floor 1 (dungeon).
        The agent starts on Floor 0 (overworld) with 5 wood, 5 stone, and a wood
        pickaxe (tier 1) in inventory. No workstations or iron/coal resources are
        pre-placed. The agent must: 1) collect at least 1 coal and 3 iron from
        natural deposits, 2) place a crafting table and furnace using provided
        resources, 3) craft iron armor, and 4) descend via ladder. Moderate mob
        pressure (melee_spawn_multiplier=1.0, ranged_spawn_multiplier=0.5) creates
        incentive to craft armor before descent, while the ladder down remains pre-
        unlocked (monsters_killed=0) to focus training on the integrated resource-
        crafting dependency chain. Iron and coal deposits follow natural generation
        probabilities (3% coal, 2% iron in stone areas).
    Relevant Achievements: MAKE_IRON_ARMOUR, ENTER_DUNGEON, PLACE_TABLE,
        PLACE_FURNACE, COLLECT_COAL, COLLECT_IRON
    Completed Achievements: COLLECT_WOOD, COLLECT_STONE, MAKE_WOOD_PICKAXE
    World:
    - Player: Starts on floor 0 with inventory: {"wood": 5, "stone": 5, "coal": 0, "
        iron": 0, "pickaxe": 1}.
    - Map:
      - No pre-placed workstations (crafting table or furnace).
      - No guaranteed iron/coal deposits - natural generation only (3% coal, 2% iron
          in stone areas).
      - Ladder down on floor 0 is pre-unlocked (monsters_killed[0] = 0).
    - Mechanics:
      - melee_spawn_multiplier = 1.0
      - ranged_spawn_multiplier = 0.5
      - monsters_killed_to_clear_level = 0
    """

    def __init__(self, static_params: StaticEnvParams, params: EnvParams):
        super().__init__(static_params, params)
        self.relevant_achievements = [
            Achievement.MAKE_IRON_ARMOUR,
            Achievement.ENTER_DUNGEON,
            Achievement.PLACE_TABLE,
            Achievement.PLACE_FURNACE,
            Achievement.COLLECT_COAL,
            Achievement.COLLECT_IRON,
        ]
        self.completed_achievements = [
            Achievement.COLLECT_WOOD,
            Achievement.COLLECT_STONE,
            Achievement.MAKE_WOOD_PICKAXE,
        ]
        self.label = "MAKE_IRON_ARMOUR, ENTER_DUNGEON, PLACE_TABLE, PLACE_FURNACE,
            COLLECT_COAL, COLLECT_IRON"

    def get_task_params(self) -> TaskParams:
        """Return custom parameters for this task."""
        return TaskParams(
            melee_spawn_multiplier=1.0,
            ranged_spawn_multiplier=0.5,
            monsters_killed_to_clear_level=0,
        )
```

```
    def generate_world(self, rng: jax.Array) -> EnvState:
        """Generates the world for the task."""
        rng, build_rng = jax.random.split(rng)

        builder = WorldBuilder(build_rng, self.static_params, self.params)

        builder.set_starting_floor(0)

        # Set player inventory with 5 wood, 5 stone, and wood pickaxe (tier 1)
        builder.set_player_inventory({
            "wood": 5,
            "stone": 5,
            "coal": 0,
            "iron": 0,
            "pickaxe": 1
        })

        # Set monsters killed count for floor 0 to 0 to unlock ladder down
            immediately
        builder.set_monsters_killed(0, 0)

        return builder.build(rng)
```

### Level 287

```
import jax
from craftax.craftax.constants import Achievement, BlockType
from craftax.craftax.craftax_state import EnvParams, StaticEnvParams

from minicraftax.craftax_state import EnvState, TaskParams
from minicraftax.tasks.base_task import BaseTask
from minicraftax.world_builder import WorldBuilder

class Env(BaseTask):
    """Objective: Integrate iron armor crafting into a survival loop by requiring
        the agent to craft armor before clearing Floor 0 to unlock descent to the
        dungeon.
Description: The player must craft iron armor and use it to survive combat while
    clearing Floor 0 (overworld) to unlock descent to Floor 1 (dungeon). The agent
    starts on Floor 0 with 5 wood, 5 stone, and a wood pickaxe (tier 1) in inventory
    . No workstations or iron/coal resources are pre-placed. The agent must: 1)
    collect at least 1 coal and 3 iron from natural deposits, 2) place a crafting
    table and furnace using provided resources, 3) craft iron armor, and 4) defeat 8
     hostile mobs (zombies/skeletons) to unlock the ladder down. Increased mob
    pressure (melee_spawn_multiplier=1.2, ranged_spawn_multiplier=0.7) creates
    authentic survival urgency, forcing the agent to prioritize armor crafting
    before combat. Iron and coal deposits follow natural generation probabilities
    (3% coal, 2% iron in stone areas). The ladder down requires 8 monster kills (
    monsters_killed_to_clear_level=8) to unlock, teaching proper sequence: gather
    resources -> craft armor -> clear floor -> descend.
Relevant Achievements: MAKE_IRON_ARMOUR, ENTER_DUNGEON, DEFEAT_ZOMBIE,
    DEFEAT_SKELETON
Completed Achievements: COLLECT_WOOD, COLLECT_STONE, MAKE_WOOD_PICKAXE, PLACE_TABLE,
     PLACE_FURNACE
World:
- Player: Starts on floor 0 with inventory: {"wood": 5, "stone": 5, "coal": 0, "iron
    ": 0, "pickaxe": 1}.
- Map:
  - No pre-placed workstations (crafting table or furnace).
```

```
    - No guaranteed iron/coal deposits - natural generation only (3% coal, 2% iron in
       stone areas).
    - Ladder down on floor 0 requires 8 monster kills to unlock (monsters_killed[0] =
       0, monsters_killed_to_clear_level = 8).
    - Natural mob spawns follow standard distribution (zombies on grass, skeletons
       near trees).
 - Mechanics:
   - melee_spawn_multiplier = 1.2
   - ranged_spawn_multiplier = 0.7
   - monsters_killed_to_clear_level = 8
   - needs_depletion_multiplier = 1.0
   - health_recover_multiplier = 1.0"""

    def __init__(self, static_params: StaticEnvParams, params: EnvParams):
        super().__init__(static_params, params)
        self.relevant_achievements = [
            Achievement.MAKE_IRON_ARMOUR,
            Achievement.ENTER_DUNGEON,
            Achievement.DEFEAT_ZOMBIE,
            Achievement.DEFEAT_SKELETON,
        ]
        self.completed_achievements = [
            Achievement.COLLECT_WOOD,
            Achievement.COLLECT_STONE,
            Achievement.MAKE_WOOD_PICKAXE,
            Achievement.PLACE_TABLE,
            Achievement.PLACE_FURNACE,
        ]
        self.label = "MAKE_IRON_ARMOUR, ENTER_DUNGEON, DEFEAT_ZOMBIE,
            DEFEAT_SKELETON"

    def get_task_params(self) -> TaskParams:
        """Return custom parameters for this task."""
        return TaskParams(
            melee_spawn_multiplier=1.2,
            ranged_spawn_multiplier=0.7,
            monsters_killed_to_clear_level=8,
            needs_depletion_multiplier=1.0,
            health_recover_multiplier=1.0,
        )

    def generate_world(self, rng: jax.Array) -> EnvState:
        """Generates the world for the task."""
        rng, build_rng = jax.random.split(rng)

        builder = WorldBuilder(build_rng, self.static_params, self.params)

        builder.set_starting_floor(0)

        # Set player inventory with 5 wood, 5 stone, and wood pickaxe (tier 1)
        builder.set_player_inventory({
            "wood": 5,
            "stone": 5,
            "coal": 0,
            "iron": 0,
            "pickaxe": 1
        })

        # Set monsters killed count for floor 0 to 0 (requires 8 kills to unlock
            ladder)
        builder.set_monsters_killed(0, 0)

        return builder.build(rng)
```

## Level 532

```
import jax
from craftax.craftax.constants import Achievement, BlockType
from craftax.craftax.craftax_state import EnvParams, StaticEnvParams

from minicraftax.craftax_state import EnvState, TaskParams
from minicraftax.tasks.base_task import BaseTask
from minicraftax.world_builder import WorldBuilder

class Env(BaseTask):
        """Objective: Force combat-ready iron progression by requiring iron sword
            crafting on floor 1 before surviving Gnome encounters during Gnomish
            Mines entry.

Description: The player must achieve ENTER_GNOMISH_MINES and DEFEAT_GNOME_WARRIOR.
    The agent starts on Floor 1 (Dungeon) with completed MAKE_STONE_PICKAXE/
    MAKE_STONE_SWORD achievements and pre-collected resources matching a competent
    floor 1 arrival. Exactly 1 iron ingot and 1 coal are provided (insufficient for
    both sword and pickaxe) to force weapon prioritization. A crafting table is
    placed at (20,20) and furnace at (22,20) on floor 1. The ladder down to floor 2
    at (24,24) is pre-unlocked (monsters_killed=8). Floor 2 contains 2 Gnome
    Warriors (melee type_id=1) within Manhattan distance 3-6 of the ladder up
    position (24,24) and 1 Gnome Archer (ranged type_id=1) within distance 8-12. All
     mechanics use default multipliers to maintain natural difficulty scaling. The
    task eliminates floor 0 backtracking while introducing the critical combat
    dependency the agent neglects in the parent task.

Relevant Achievements: ENTER_GNOMISH_MINES, DEFEAT_GNOME_WARRIOR
Completed Achievements: MAKE_STONE_PICKAXE, MAKE_STONE_SWORD, COLLECT_WOOD,
    COLLECT_STONE, COLLECT_COAL, PLACE_TABLE, PLACE_FURNACE

World:
- Player: Starts on floor 1 with inventory {"wood": 8, "stone": 12, "coal": 1, "iron
    ": 1, "stone_pickaxe": 1, "stone_sword": 1} and completed achievements for all
    basic stone/wood crafting
- Map:
  * `CRAFTING_TABLE` block placed at fixed position (20, 20) on floor 1
  * `FURNACE` block placed at fixed position (22, 20) on floor 1
  * Ladder down positioned at (24, 24) on floor 1 with
      monsters_killed_to_clear_level set to 0 (pre-unlocked)
  * Two `GNOME_WARRIOR` (melee mob type_id=1) placed randomly within Manhattan
      distance 3-6 of ladder up position (24, 24) on floor 2
  * One `GNOME_ARCHER` (ranged mob type_id=1) placed randomly within Manhattan
      distance 8-12 of ladder up position (24, 24) on floor 2
  * All other blocks follow default Dungeon (floor 1) and Gnomish Mines (floor 2)
      generation
- Mechanics: "monsters_killed_to_clear_level = 0" (for floor 1), "
    melee_spawn_multiplier = 1.0", "ranged_spawn_multiplier = 1.0", "
    mob_health_multiplier = 1.0", "mob_damage_multiplier = 1.0"
        """

        def __init__(self, static_params: StaticEnvParams, params: EnvParams):
                super().__init__(static_params, params)
                self.relevant_achievements = [Achievement.ENTER_GNOMISH_MINES,
                    Achievement.DEFEAT_GNOME_WARRIOR]
                self.completed_achievements = [
                        Achievement.MAKE_STONE_PICKAXE,
                        Achievement.MAKE_STONE_SWORD,
                        Achievement.COLLECT_WOOD,
                        Achievement.COLLECT_STONE,
                        Achievement.COLLECT_COAL,
```

```
                Achievement.PLACE_TABLE,
                Achievement.PLACE_FURNACE
        ]
        self.label = "ENTER_GNOMISH_MINES, DEFEAT_GNOME_WARRIOR"

    def get_task_params(self) -> TaskParams:
        """Return custom parameters for this task."""
        return TaskParams(
                monsters_killed_to_clear_level=0,
                melee_spawn_multiplier=1.0,
                ranged_spawn_multiplier=1.0,
                mob_health_multiplier=1.0,
                mob_damage_multiplier=1.0,
        )

    def generate_world(self, rng: jax.Array) -> EnvState:
        """Generates the world for the task."""
        rng, build_rng, gnome_warrior1_rng, gnome_warrior2_rng,
            gnome_archer_rng = jax.random.split(rng, 5)

        builder = WorldBuilder(build_rng, self.static_params, self.params)

        # Set starting floor to 1 (Dungeon)
        builder.set_starting_floor(1)

        # Set player inventory with resources for floor 1
        builder.set_player_inventory({
                "wood": 8,
                "stone": 12,
                "coal": 1,
                "iron": 1,
                "pickaxe": 2,  # 2 = stone pickaxe
                "sword": 2,      # 2 = stone sword
        })

        # Place CRAFTING_TABLE at fixed position (20, 20) on floor 1
        builder.place_block(1, BlockType.CRAFTING_TABLE, (20, 20))

        # Place FURNACE at fixed position (22, 20) on floor 1
        builder.place_block(1, BlockType.FURNACE, (22, 20))

        # Set monsters_killed[1] to 8 (so ladder down is already unlocked)
        builder.set_monsters_killed(1, 8)

        # Place two GNOME_WARRIOR (melee mob type_id=1) randomly within
            Manhattan distance 3-6 of ladder up position (24, 24) on floor 2
        builder.add_mobs_randomly_near(
                gnome_warrior1_rng,
                level=2,
                mob_name="melee",
                type_id=1,  # type_id 1 is Gnome Warrior
                n=1,
                target_pos=(24, 24),
                min_dist=3,
                max_dist=6,
                on_blocks=[BlockType.PATH, BlockType.GRASS],
        )
        builder.add_mobs_randomly_near(
                gnome_warrior2_rng,
                level=2,
                mob_name="melee",
                type_id=1,  # type_id 1 is Gnome Warrior
                n=1,
```

```
            target_pos=(24, 24),
            min_dist=3,
            max_dist=6,
            on_blocks=[BlockType.PATH, BlockType.GRASS],
    )

    # Place one GNOME_ARCHER (ranged mob type_id=1) randomly within
        Manhattan distance 8-12 of ladder up position (24, 24) on floor
        2
    builder.add_mobs_randomly_near(
            gnome_archer_rng,
            level=2,
            mob_name="ranged",
            type_id=1,  # type_id 1 is Gnome Archer
            n=1,
            target_pos=(24, 24),
            min_dist=8,
            max_dist=12,
            on_blocks=[BlockType.PATH, BlockType.GRASS],
    )

    return builder.build(gnome_archer_rng)
```

# D. Additional Quantitative Results

## D.1. Full Achievement Breakdown

We provide a comprehensive evaluation of agent performance across the Craftax achievement hierarchy.

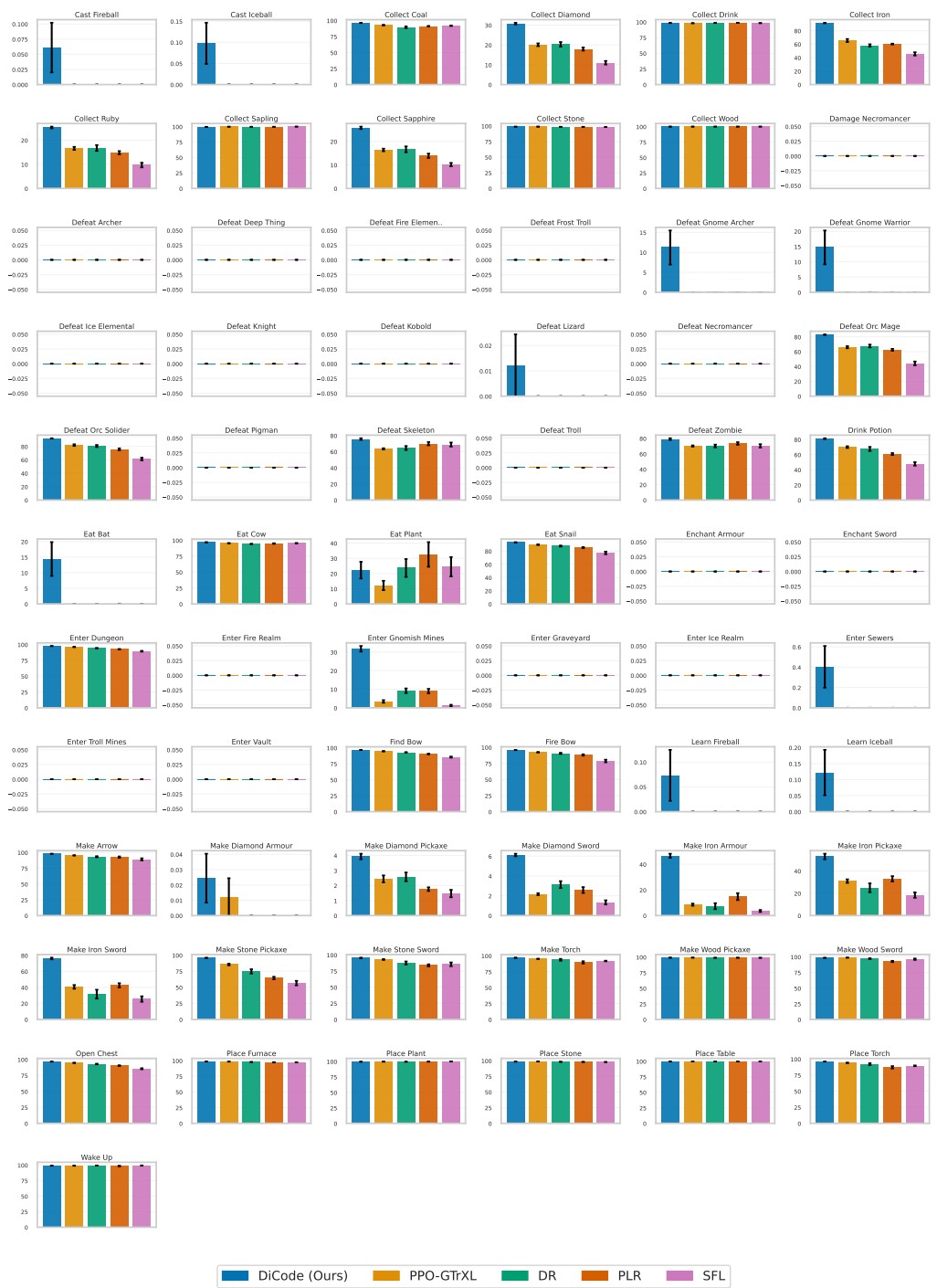

*Figure 6.* **Final Achievement Success Rates.** Aggregate success rates for DiCode versus baselines across all defined Craftax achievements. Results report the mean and standard error across 8 seeds after $2 \times 10^9$ steps.

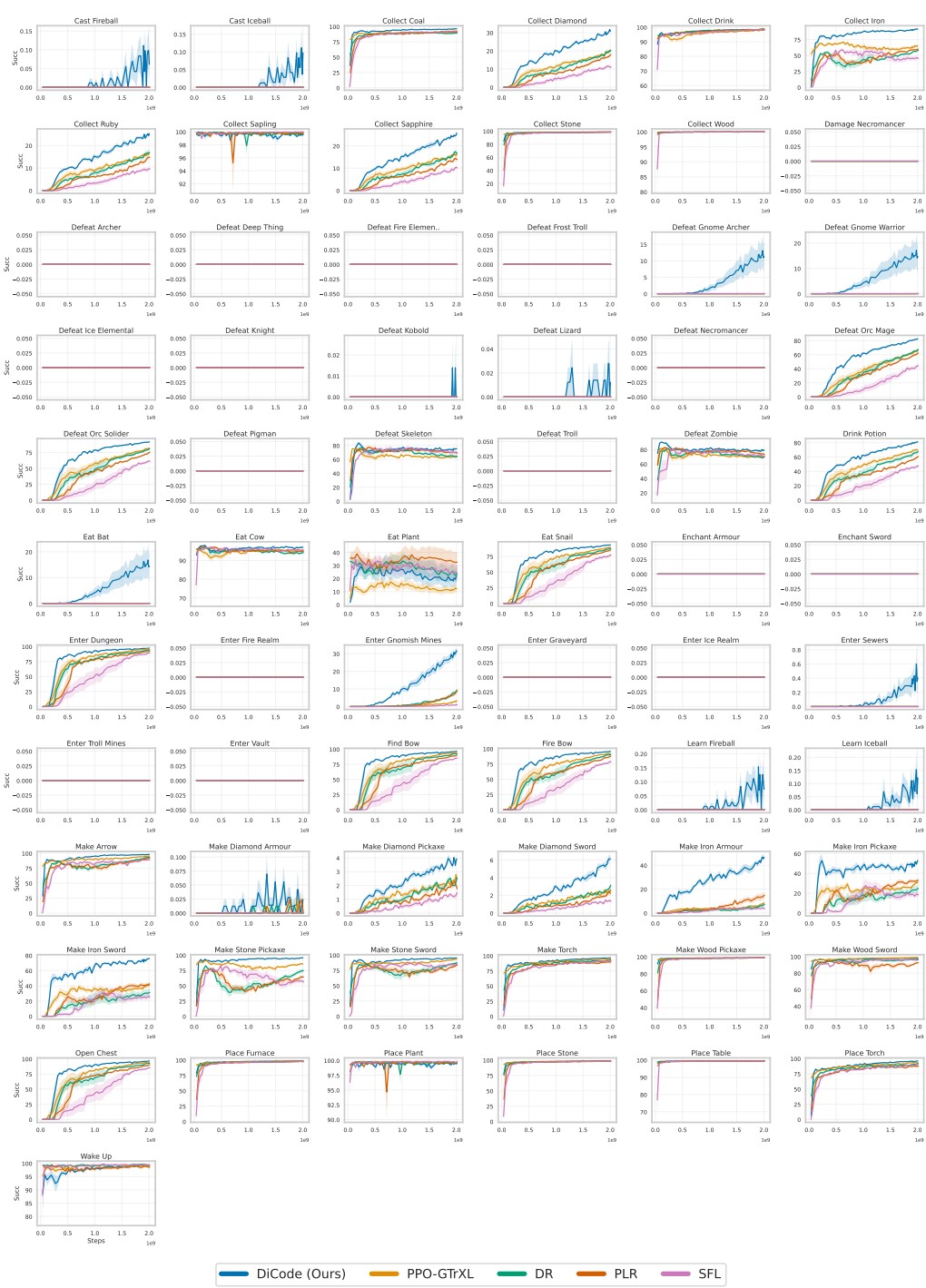

*Figure 7.* **Achievement Learning Curves.** Mean success rate for individual achievements over the course of training, illustrating the rate of acquisition. Shaded regions denote standard error across 8 seeds.

### D.2. Ablation Analysis

This section provides supplementary results for the ablation study presented in Section 4.2. We ablate DiCode along four design axes: parent selection (Random Parent Sampling), environment reshaping (Goal Only), FM capability (Qwen80B, Qwen30B), and closed-loop grounding (Open-Loop). DiCode uses 8 seeds; all ablation variants use 5 seeds.

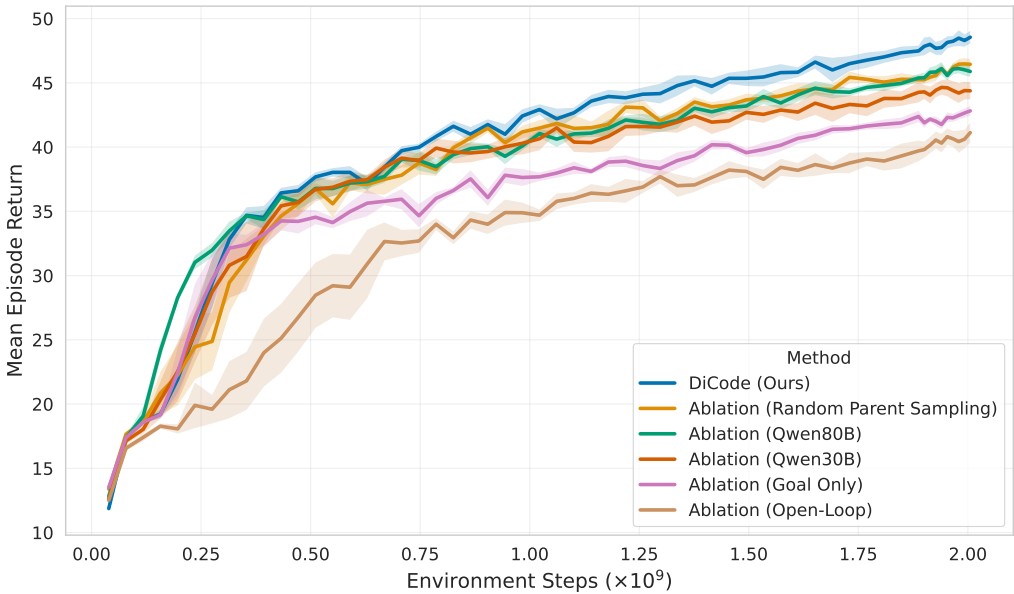

*Figure 8.* **Ablation Learning Curves.** Mean episode return on the held-out test set throughout training. Shaded regions indicate standard error across seeds.

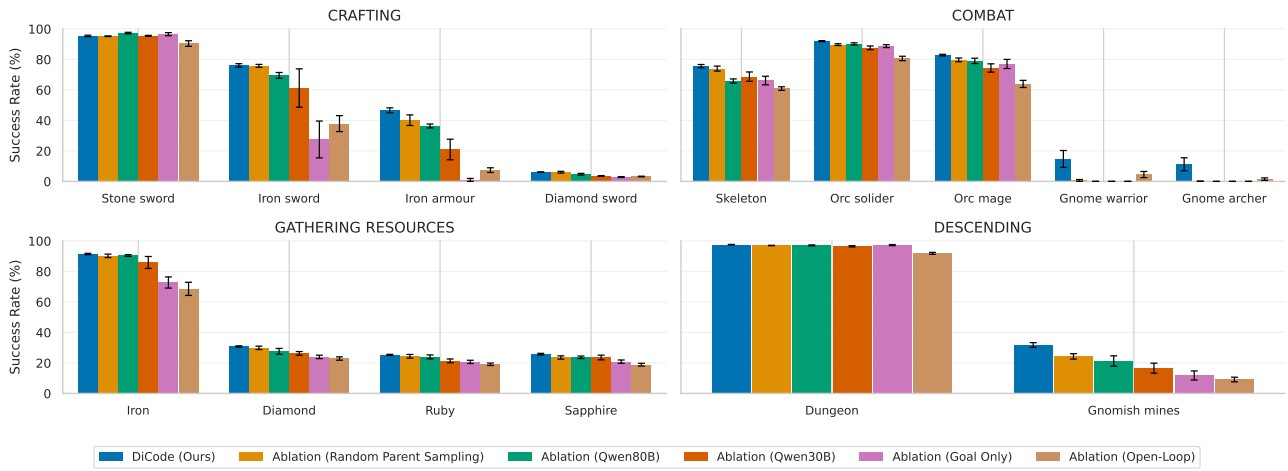

*Figure 9.* **Ablation Achievement Breakdown.** Final success rates on selected achievements grouped by category. Error bars denote standard error across seeds.

## E. Domain Adaptation Guide

DiCode's framework architecture – FM generates code, engine executes it, learnability scores close the loop – is domain-independent. Applying it to a new domain requires implementing a single interface layer that exposes the engine's control axes over *initial state*, *transition dynamics*, and *goals*. The guiding design principle is: grant the FM enough expressivity to meaningfully reshape the environment while limiting the surface area to maintain compilation reliability. This section

describes the three required components, concretized with the Craftax implementation as reference and a hypothetical MuJoCo adaptation.

**Component 1: Programmatic API.**    The API wraps the engine's state manipulation into a set of methods the FM can call within a task class. It should expose three orthogonal axes:

- *Initial state construction:*  Methods the FM calls programmatically to construct the starting world state (terrain layout, entity placement, agent inventory/position).    In Craftax, this is the `WorldBuilder` class (15 methods:  `set_starting_floor`, `set_player_inventory`, `place_block`, `fill_area`, `add_mobs_randomly_near`, `place_randomly_near`, etc.).

- *Transition dynamics:* A parameter structure that modulates the engine's core update rules, providing a compilation-safe mechanism for adjusting dynamics without requiring the FM to implement low-level simulation logic directly. In Craftax, this is `TaskParams` – a flat dataclass of 12 multipliers and thresholds (e.g., `mob_damage_multiplier`, `health_recover_multiplier`, `melee_trigger_distance`) that are applied within the engine's step function.

- *Goal specification:* A mechanism to define task success. In Craftax, this is a list of `Achievement` enum values; the task terminates successfully when all are achieved.

The FM implements a class inheriting from an abstract `BaseTask`, overriding `get_task_params()` (dynamics), `generate_world(rng)` (initial state via the builder API), and setting `relevant_achievements` (goals). This three-method contract is the complete interface.

**Component 2: Prompt context modules.**    The FM requires textual descriptions of the domain's mechanics, the API's methods and their semantics, and the available goal/parameter spaces. These are injected into the generation prompt as structured context (see Section B). For Craftax, this includes documentation of the technology tree, achievement dependencies, valid block/item/mob types, and `WorldBuilder` method signatures with usage examples.

**Component 3: Seed levels.**    A small set (e.g., 4) of hand-written task implementations that demonstrate how to use the API to construct diverse environments. Seed levels serve primarily as demonstrations of API usage patterns rather than curriculum blueprints – the FM generalizes from these to produce novel designs. To test robustness to seed design, we ran DiCode with simplified seed levels that retain only goal specifications, with initial states and dynamics set to engine defaults – meaning the FM receives no demonstrations of how to reshape environments, though it retains full API access to do so. This simplified-seed variant achieves 47.3 mean return across 3 seeds, comparable to DiCode (48.6), confirming that detailed seed level designs are not critical to final performance – the FM discovers effective reshaping strategies from API documentation alone. We expect this sensitivity to decrease further as FM capabilities improve.

**Adaptation to MuJoCo.**    For a continuous-control domain such as MuJoCo locomotion, an analogous API would expose: (1) *initial state* – terrain layout (heightfields, obstacles, platforms), initial robot configuration (joint angles, body position); (2) *dynamics* – physical parameters (friction coefficients, gravity, actuator gains, joint damping) as a flat parameter structure analogous to `TaskParams`; (3) *goals* – target positions, desired velocities, or task completion conditions. In practice, the FM would generate a Python function that programmatically constructs the environment – for example, building terrain via heightfield utilities or composing an MJCF model specification – analogous to how it currently calls `WorldBuilder` methods in Craftax. The key domain-specific consideration is defining appropriate parameter bounds for continuous spaces to prevent degenerate configurations.

**Scope and cost.**    The Craftax interface comprises `TaskParams` (12 parameters) and `WorldBuilder` (15 methods). This is a one-time engineering cost per domain: once implemented, it removes this overhead for subsequent researchers. We note that this domain-specific API requirement is standard across FM-based environment design systems; GenSim (Wang et al., 2024b) and Eurekaverse (Liang et al., 2024) impose analogous interface requirements. Moreover, our FM scaling experiments (Section 4.2) show that stronger FMs achieve higher performance with the same interface without any API redesign, suggesting that the balance between expressivity and compilation reliability becomes easier to maintain as FM capabilities improve.

