# OpenReview forum: "Dreaming in Code for Curriculum Learning in Open-Ended Worlds"
_ICML.cc/2026/Conference — ICML 2026 regular_

### Official Review · Reviewer_vMj3 · 2026-02-24

**Soundness:** 2
**Presentation:** 2
**Significance:** 2
**Originality:** 2
**Overall Recommendation:** 4
**Confidence:** 3

**Summary:**

This work proposes Dreaming in Code (DiCode). The pipeline addresses the stagnation often seen in open-ended reinforcement learning by evolving the environment design space from low-dimensional parameters to executable Python code. Utilizing a Foundation Model (FM) as a "curriculum architect," the framework selects "parent" environments based on an agent’s current success and "dreams" up new variations by synthesizing code that modifies world generation, game mechanics, and task objectives. Tested on the Craftax benchmark, DiCode creates a bridge of intermediate tasks, through providing and then removing "scaffolding" (such as initial resources or simplified combat) to lead the agent toward mastering complex, long-horizon sequences. This closed-loop process allows the agent to achieve a 16% improvement in mean return and solve late-game tasks that were previously deemed intractable for standard RL and UED baselines.

**Compliance With Llm Reviewing Policy:**

Affirmed.

**Final Justification:**

I maintain my positive evaluation after reading the paper and the authors’ detailed rebuttal.

**Key Questions For Authors:**

- To what extent is the FM’s "teacher-like" behavior dependent on the specific curriculum design philosophy detailed in the system prompt? Would the system still work if the prompt did not explicitly suggest "thin-slicing" or "scaffolding"? Is there any data supporting the conclusion?
- I assume the adaptive bonus will create a risk of "curriculum drift," where the agent prioritizes the level-specific bonus over learning the robust survival skills needed for the target environment. How does the pipeline solve this problem?
- For the long-term stability, does this risk "forgetting" early-game foundational skills if those levels are eventually purged or deprioritized by the PLR mechanism as the agent evolves?

**Limitations:**

yes

**Strengths And Weaknesses:**

Strengths.
The paper marks a pivotal shift in unsupervised environment design, successfully transitioning from traditional parameter-tuning to a sophisticated program-synthesis paradigm. By operating within a Turing-complete code space, DiCode unlocks an unprecedented level of expressivity, allowing the curriculum to redefine core world logic rather than mere spatial configurations. This conceptual ambition is backed by striking empirical gains in the Craftax benchmark; specifically, the leap from a 0% success rate to non-zero performance on advanced combat tasks provides a compelling "proof of concept" for the necessity of algorithmic stepping-stones in crossing exploration bottlenecks.

Weaknesses.
- The framework’s claim to "open-endedness" is somehow tethered to the hand-crafted granularity of the WorldBuilder API and highly prescriptive system prompts. This raises a fundamental concern regarding the "intelligence" of the curriculum: it remains unclear whether the strategic depth of the generated levels is a true emergence of the model's reasoning or largely a reflection of the human-encoded heuristics embedded in the instructions. This architectural dependency is compounded by a formidable scalability bottleneck, as LLM inference latency drives a nearly five-fold increase in training time, casting doubt on its current efficiency relative to simpler baselines.
- The reliance on an adaptive reward bonus  introduces a manual heuristic that may invite reward hacking or policy instability as the agent's performance fluctuates.

---

> ### Author Rebuttal · Authors · 2026-03-31
>
> We thank Reviewer vMj3 for their detailed and constructive feedback, and for recognizing DiCode's contribution in transitioning environment design from parameter-tuning to program synthesis and its empirical gains on previously intractable tasks.
>
> > Prompt heuristics vs. FM reasoning (W1, Q1)
>
> The API and prompts provide structural scaffolding, but the strategic depth stems directly from the FM's reasoning. To prove this without conflating variables via prompt ablation, we held all prompts and heuristics constant and scaled FM’s reasoning capability from Qwen3-235B-A22B-Thinking-2507 (size 235B) to Qwen3-Next-80B-A3B-Thinking (size 80B) and Qwen3-30B-A3B-Thinking-2507 (size 30B) (models spanning a well-characterized reasoning capability ladder; Yang et al., 2025).
>
> Mean return degrades monotonically as FM capability decreases – 235B achieves 48.6, 80B drops to 45.9, 30B to 44.4 – though all three still outperform the strongest baseline (41.5). If the curriculum's quality were primarily encoded in the prompts and API, models receiving identical instructions should perform comparably. The clear capability gradient demonstrates that the FM's reasoning is an independent and meaningful driver of curriculum quality, and one that scales with FM capability. The API defines the vocabulary through which the FM expresses its curriculum decisions; it does not determine which decisions to make.
>
> We will include the experiments and analysis in the revision.
>
> > Runtime (W1)
>
> We correct a reporting inaccuracy in Appendix A.4: the \~48h figure conflated training time with infrastructure-dependent wait time and did not reflect our asynchronous pipeline. On matched hardware (RTX 6000 Ada), PPO-GTrXL completes in \~8.8h (3 seeds), while DiCode averages \~32h total execution time (2 seeds). DiCode's runtime decomposes into three components: RL training (\~15.9h), JAX compilation (\~14.4h), and LLM wait time (\~1.9h). The RL training overhead (~1.8x baseline) reflects the cost, in our implementation, of batching heterogeneous tasks within a single vmapped JAX training loop: different environment lanes in the batch runs its own task-specific initialization, mechanics, and termination logic within a shared Craftax simulation kernel. JAX compilation (\~14.4h) arises because DiCode continuously introduces new environment logic throughout training, requiring recompilation; baselines avoid this because the environment logic is fixed across all seeds, requiring only a single compilation. Moreover, despite continuous environment generation throughout training, the asynchronous pipeline ensures training is blocked for only \~1.9h on average – a figure that depends on LLM serving infrastructure, not on the method itself.
>
> We will include the runtime analysis in the revision.
>
> > Bonus reward, curriculum drift, and skill retention (W2, Q2, Q3)
>
> Curriculum drift is structurally prevented by the 20% target environment allocation – in every training iteration, the agent trains on unmodified target environment with no bonus, ensuring survival skills are continually reinforced under the real reward structure. Generated-level goals are a subset of target-environment achievements, so the bonus cannot reward skills irrelevant to the target; moreover, it is awarded once upon goal completion and terminates the episode, precluding sustained exploitation. The adaptive scaling (Eq. 5) ties the bonus to the agent's current return, maintaining a consistent relative incentive across training stages. Regarding long-term forgetting, the 20% target allocation also ensures constant exposure to the full game including foundational skills. For R4's specific concern about levels being deprioritized by PLR, the staleness weighting (Eq. 2) ensures levels not recently sampled accumulate increasing replay probability, preventing permanent deprioritization. Empirically, DiCode improves over baselines across all achievement categories including early-game skills (Figure 3), consistent with no drift or forgetting occurring.
>
> We will add clarifications about these concerns in the revision.

---

> > ### Author Rebuttal · Reviewer_vMj3 · 2026-04-02
> >
> > Thank you for your future experiments and results. I tend to keep my score.

---

> > > ### Author Response · Authors · 2026-04-05
> > >
> > > We thank reviewer vMj3 for their review and are glad that the raised concerns have been fully resolved. The paper has been strengthened by the new evidence provided during rebuttal, including the FM scaling experiments, the runtime decomposition, and the detailed analysis of curriculum drift and skill retention mechanisms. We hope the additional evidence provided may warrant reconsideration of the score. We respect the reviewer’s judgment either way.

---

### Official Review · Reviewer_YADN · 2026-03-13

**Soundness:** 3
**Presentation:** 4
**Significance:** 4
**Originality:** 3
**Overall Recommendation:** 5
**Confidence:** 4

**Summary:**

In this paper, the authors introduce a new method for Unsupervised Environment Design (UED) to build a curriculum for a deep RL agent. Inspired by the previous UED literature as well as recent advances in LLMs and the environment evolution methods that has allowed, the authors propose a method for LLM-based evolution of existing levels in the task archive expressed as code. The evolving process is based on previous empirical success rates on a task as well as on the target task. The curriculum is constructed by sampling tasks from the evolved archive based on a learnability score from previous literature.

Using this method, the authors achieve significantly better scores than simple baselines as well as previous methods from the UED literature on the Craftax environment. They in particular show that difficult tasks are achived with some success by their agent compared to the nil score of other methods. They also showcase the stable overall success rate over time, together with continuously improving scores on validation tasks.

**Compliance With Llm Reviewing Policy:**

Affirmed.

**Final Justification:**

I maintain my score after reading the rebuttal, and lower my confidence slightly to leave more weight to interesting discussions with other reviewers.

**Key Questions For Authors:**

* How do you explain that UED baselines perform worse than sampling tasks at random? Are these algorithms used in a domain where they are somehow not applicable?

**Limitations:**

The paper acknowledges some limitations.
I have an additional one the authors might want to discuss: like all sucess-rate-based replay scores, the learnability score used in this method is vulnerable to random tasks, oversampling a hypothetical coin-flip level that can never be solved fully. Can you explain how elements of your proposed approach are effective against random environments, or what could be done to mitigate this in follow-up work?

**Strengths And Weaknesses:**

## Strengths

* This paper is exceptionally well-written, it was a pleasure to follow along. Prior work and background was well-contextualized, the right amount of formalism was given and illustrating examples were given when needed. Figures are clear and integrate with the flow of the paper.
* The paper brigdes prior auto-curriculum methods with environment evolution methods such as OMNI_EPIC; the strength of this paper compared to the latter is sequencing tasks for one single generalist agent.
* The method is general and can be applied to any domain where tasks can be specified as code (continuous control, LLM agents, etc). This indicates high potential significance, as others are likely to build on this method.
* The experimental results are informative and support the validity of the method.

## Weaknesses

* The authors could have included additional evaluation against newer methods such as ACCEL, as well as more detail in the main text on how previous UED methods have been adapted to this domain.
* There are a lot of design choices and hyperparameters in this method; some of them are supported by prior work but the paper would nevertheless be strengthened by an empirical or in-principle justification of these design choices (for instance, the reward structure, the selection strategy that's a bit complex).

---

> ### Author Rebuttal · Authors · 2026-03-31
>
> We thank Reviewer YADN for their thorough and insightful evaluation. We are particularly encouraged by their recognition of the potential significance of our work and their belief that others are likely to build on this method. This aligns closely with our motivation for this line of research.
>
> > UED baselines in Craftax and ACCEL (W1, Q1)
>
> Previous UED baselines were adapted to Craftax in the standard way, by curating over random seeds, which determine the terrain layout of each floor – the spatial arrangement of resources, water, stone, lava, etc. We will expand on this characterization in the revision. As noted in Section 4.1, UED baselines implementation was adapted from Monette et al., (2025) with a shared PPO-GTrXL agent.
>
> Regarding Q1, Craftax was designed with UED evaluation in mind, and seed-level curation does provide value by exposing the agent to varied terrain configurations. However, the seed space controls terrain, not game mechanics or skill dependencies. The dimensions that drive progression – starting inventory, crafting prerequisites, mob difficulty, floor placement – are governed by the technology tree and tightly coupled through the game's internal logic; varying them meaningfully is unlikely through random perturbation alone, and instead benefits from domain-aware reasoning. We hypothesize that by concentrating training on seeds deemed high-regret or high-learnability along a weakly informative axis, curation narrows the training distribution without a compensating improvement in learning signal, explaining why UED baselines perform worse than domain randomization in this domain.
>
> Regarding ACCEL, Matthews et al. (2024) showed that ACCEL underperforms PLR on Craftax. Since DiCode already significantly outperforms PLR, we judged that including ACCEL would not yield additional insight.
>
> > Design choices and hyperparameters (W2)
>
> We provide new ablations that empirically validate the main design choices, alongside in-principle justification where appropriate. The full ablation table is in our response to Reviewer CwzL, in "Ablations" Section. We highlight here the results most relevant to Reviewer YADN’s concern.
>
> We ablated the parent selection mechanism by replacing it with uniform random sampling. Mean return drops modestly (46.9 vs 48.6), but unlike DiCode, the success rate on late-game tasks drops to zero. This indicates that principled parent selection is not critical for early/mid-game gains but is essential for driving progression to late-game tasks.
>
> We will include the ablations in the revision.
>
> As motivated in Section 3.1, each generated level adds a bonus upon goal completion to ensure the agent pursues the level-specific goal. We empirically observed during development that without this alignment, the agent exploits dense early-game rewards and ignores the intended skill – undermining the core benefit of targeted environment shaping, providing the FM with inaccurate learnability signals, and making generated levels no more useful than the target environment.
>
> > Learnability score vulnerability to stochastic levels
>
> This is a genuine inherited limitation of success-rate-based scoring, where a hypothetical coin-flip level would maintain a success rate of \~0.5 indefinitely and be persistently oversampled. In our setting, DiCode's use of discrete goal compositions provides partial insulation, as level outcomes are predominantly determined by agent capability rather than stochastic factors. This is consistent with the success rate trajectories we observe for individual levels, which generally increase over training. However, this does not eliminate the vulnerability in general.
>
> A principled mitigation, well-established in the curriculum learning literature, is to supplement the learnability score with a learning progress signal (Graves et al., 2017; Portelas et al., 2019) – measuring the rate of change in success rate rather than its absolute value. A stochastic level's success rate remains flat at \~0.5 despite repeated training, yielding zero learning progress, and would thus be naturally deprioritized. Integrating such a signal into DiCode's scoring is a natural extension that we will discuss in the paper's limitations section in the revision.

---

> > ### Author Rebuttal · Reviewer_YADN · 2026-04-08
> >
> > No major concerns left, I thank the authors for their response.

---

### Official Review · Reviewer_rZ2M · 2026-03-13

**Soundness:** 3
**Presentation:** 3
**Significance:** 3
**Originality:** 3
**Overall Recommendation:** 4
**Confidence:** 4

**Summary:**

This paper proposes DiCode, a framework for curriculum learning in open-ended RL environments where a foundation model generates executable environment code to scaffold agent training. The key idea is to use an FM as a "teacher" that synthesizes Python programs defining new training levels, conditioned on the agent's current performance and a parent level from an archive. The method is evaluated on Craftax, a procedurally generated open-ended benchmark. DiCode achieves a 16% improvement in mean return over the strongest baseline and, notably, obtains non-zero success on late-game combat tasks where all baselines score 0%.

**Compliance With Llm Reviewing Policy:**

Affirmed.

**Final Justification:**

My final recommendation is to keep my score, the rebuttal has addressed my concerns but by reevaluating the paper, I may give up the opportunity to give it a 5 which explicitly indicate I love this work very much, I think this is good, but not extraordinary to me. Giving a positive but not that strong recommendation would be suitable for me. But still, as I said I would be willing to increase the score in my initial review, I hope the AC to seriously reconsider(maybe upweight) the weight of my rate of 4.

**Key Questions For Authors:**

1. How sensitive is DiCode to the quality of the seed tasks and the hand-crafted API interface? If these are poorly designed, does the curriculum still bootstrap effectively? This matters for assessing generalization effort.

2. What is the compilation success rate of the FM-generated code across training, and how does it evolve over time? Understanding the yield of valid levels is important for judging the practical overhead.

3. The paper fixes 20% of the training budget on the target environment. Was this ratio tuned, and how sensitive is performance to it? A sweep or ablation here would strengthen the claims.

4. Have you considered applying DiCode to a second domain, even a simpler one like a MuJoCo task, to provide evidence that the framework transfers beyond Craftax?

If the author could solve my concern, I would be willing to increase my score.

**Limitations:**

yes

**Strengths And Weaknesses:**

Strengths:
- The closed-loop curriculum design is well motivated, and the ablation against open-loop generation cleanly isolates its contribution.
- Qualitative analysis of the curriculum evolution is convincing: the FM spontaneously removes scaffolding and increases complexity in a sensible progression.
- Using an open-weights FM rather than a proprietary API is a good choice for reproducibility.
- The achievement breakdown analysis is thorough and reveals that DiCode's gains come from structural breakthroughs at specific bottlenecks, not uniform improvement.

Weaknesses:
- Evaluation is limited to a single environment, Craftax, making it hard to assess how general the approach is.
- The 4.5x wall-clock cost over baselines is substantial and not well addressed; the practical tradeoff between FM inference cost and curriculum quality needs more discussion.
- The reliance on a detailed, hand-crafted API interface and domain-specific prompting context is heavy, and the paper does not discuss how much effort this requires for a new domain.

---

> ### Author Rebuttal · Authors · 2026-03-31
>
> We thank Reviewer rZ2M for their constructive feedback and recognition of our closed-loop design.
>
> > Generalizability (W1)
>
> We agree that multi-domain evaluation would strengthen the paper. DiCode was developed to address a specific gap in the UED literature: existing methods handle low-dimensional design spaces well but struggle in complex, open-ended environments with deep progression hierarchies. Craftax offers the compositional skill dependencies and long-horizon progression that stress-test precisely the gap DiCode targets. Moreover, Craftax evaluates diverse RL capabilities within a single environment, e.g. exploration, resource management, crafting, long-horizon planning, and combat. Additionally, we note that single-domain evaluation is standard for methods in this space (MaestroMotif, ICLR 2025 oral; Eurekaverse, CoRL 2024). We share Reviewer YADN’s view that the framework generalizes naturally to domains where tasks can be specified as code, including continuous control.
>
> > Seed sensitivity (Q1)
>
> During development we found that seed levels primarily serve to demonstrate how the FM can communicate with the API to shape the environment in different ways, and not as curriculum blueprints. Regardless, to test robustness we ran DiCode with simplified seed levels that retain only goal specifications, with initial states and dynamics set to Craftax defaults. This means the seed levels no longer demonstrate how to reshape environments, though the FM retains full access to the API to do so. The Goal Only ablation (see our response to Reviewer CwzL Ablations section) confirms that this reshaping capability is a key driver of DiCode's gains. Despite receiving no demonstrations of it, results across 3 seeds show that this simplified variant achieves a close mean return to DiCode (47.3 vs 48.6), far exceeding Goal Only (42.7), confirming that the FM discovers reshaping strategies from API documentation alone. We expect this seed sensitivity to decrease further as FM capabilities improve.
>
> > Domain adaptation (W3, Q4)
>
> We acknowledge that applying DiCode to a new domain requires implementing a domain-specific API and prompt context, which is a real engineering cost. The adaptation involves three components: (1) an API wrapping the engine's state manipulation, (2) prompt context modules describing the domain's mechanics and structure, and (3) a small set of seed levels as few-shot examples. The framework architecture – FM generates code, engine executes it, learnability scores close the loop – is domain-independent; only this interface layer changes. To concretize, the Craftax interface comprises TaskParams (12 parameters governing mechanics such as spawn rates, mob difficulty) and WorldBuilder (15 primitive state-manipulation methods such as setting inventory, placing blocks, and setting starting floor).
>
> For a MuJoCo domain, an analogous API would expose terrain layout (heightfields, obstacles, platforms), initial robot configuration (joint angles, body position), physical parameters (friction, gravity, actuator gains), and goal specification (target positions, desired velocities). MuJoCo introduces its own considerations, such as defining appropriate parameter bounds for continuous spaces. We consider this comparable in scope to standard domain adaptation in this class of methods. Importantly, this is a one-time cost per domain: once open-sourced, it removes this overhead for subsequent researchers.
>
> This domain-specific API requirement is standard across FM-for-environment-design systems: GenSim, and Eurekaverse require domain-specific interfaces.
>
> We leave a second-domain evaluation as future work; the MuJoCo scoping above concretizes the path.
>
> > Practical overhead (W2, Q2)
>
> We detail the runtime concern in our response to R vMj3, Section “Runtime”.
>
> Regarding compilation success, \~60% of FM-generated code compiles successfully; since we intentionally over-request to ensure sufficient yield (see Section 3.2), the effective yield is ~71%, stable across training. Shortfalls are handled by replay padding so the training batch size remains constant. See how the compilation success rate of FM evolve over time here: https://sites.google.com/view/dreaming-in-code/fm-compilation-success-rate .
>
> Within the same 2B step budget, this overhead enables environment designs that maximize the expressivity of the domain's design space – yielding a 16% improvement in mean return and non-zero success on late-game tasks where all baselines score 0%.
>
> We will include both analyses in the revision.
>
> > Target environment ratio (Q3)
>
> The 20% target environment ratio was not tuned as a hyperparameter. It serves two structural roles: (1) collecting perf_target (see Section 3.2), which is fed back to the FM as generation context, and (2) grounding the agent in the target distribution. Without it, the agent would train exclusively on generated environments, risking distributional shift from the target.

---

> > ### Author Rebuttal · Reviewer_rZ2M · 2026-04-02
> >
> > Thanks a lot for your rebuttal, I would like to keep my score and good luck on your submission.

---

> > > ### Author Response · Authors · 2026-04-05
> > >
> > > We thank reviewer rZ2M for their review and are glad that the raised concerns have been fully resolved. The paper has been strengthened by the new evidence provided during rebuttal, including the seed robustness analysis, the compilation yield data, the detailed runtime decomposition, and the concrete MuJoCo domain scoping.
> > >
> > > In the initial review, reviewer rZ2M noted: "If the author could solve my concern, I would be willing to increase my score". Given that the concerns are now fully resolved, we would be grateful if the reviewer might consider whether a score adjustment is warranted. We respect their judgment either way.

---

### Official Review · Reviewer_CwzL · 2026-03-24

**Soundness:** 2
**Presentation:** 3
**Significance:** 2
**Originality:** 2
**Overall Recommendation:** 3
**Confidence:** 4

**Summary:**

This paper proposes a framework called DiCode that uses foundation models (FMs) to synthesize executable environment code that progressively challenges an agent towards greater competence. DiCode uses the FM to "dream" a variations of the world (in Python code) conditioned on the agent's current capabilities; this code is executed in a fixed world engine, ensuring physical validity and avoiding hallucinations. The generation processes uses a metric called learnability from Unsupervised Environment Design (UED) to sample a parent level, conditions the FM with a structured context that includes the agent's current capabilities, generates a new level description/design, and then invokes the FM again to generate the actual executable Python code. The generated code manipulates the environment through an expressive programmatic interface specific to that environment -- the authors apply DiCode to Craftax, a challenging open-ended world with rich mechanics and long-horizon progression. Training is done across the target environment, newly generated levels, and archived levels sampled using Prioritized Level Replay (PLR), a UED method.

The authors show that DiCode is able to achieve 16% higher return on average over baselines including standard RL and prior UED methods, and also achieves non-zero success in late-game tasks that prior methods fail to even reach.

**Compliance With Llm Reviewing Policy:**

Affirmed.

**Final Justification:**

The authors' rebuttal has addressed some of my concerns. I am maintaining my score but am more open-minded given the other reviews and continued discussion.

In response to the latest comments:
- I agree that your newly added ablations explore the design axes you mentioned, albeit at a coarse level. Thank you for including this; I think it addressed several reviewers' concerns.
- You make a good point that Dicode can adapt the chosen stepping stones and is not limited to a maximum difficulty, whereas the approach I proposed (static progression of stepping stones + skill ordering) is. This suggests an experiment where you train two different agent models and show that Dicode chooses different progressions/stepping stones for each.
- Your statement that the design space for the programmatic interface is constrained is not very convincing, given that it must allow the FM to "vary the initial state, transition dynamics, and goals of levels". This sounds quite involved to me.

I remain concerned about the complexity of the design, the hand-crafted API, and the various parameters and system prompts. I imagine trying to apply this approach to a new environment and am overwhelmed by the list of decisions that need to be made. Your suggestion of including a domain adaptation guide in the revision would be helpful. I don't think it's reasonable to assert that Dicode "generalizes naturally" to other domains like continuous control as you note in relation to reviewer YADN’s comment.

**Key Questions For Authors:**

1. Is there a theme or principle that emerges from manually inspecting the (high-scoring) environment sequence generated by the FM? Can you use this to procedurally generate a sequence of environments as a baseline method? I understand this might be too involved, but I wonder if you can repurpose the procedural generation method used to create your test set.
2. Can you summarize and estimate the work involved in applying DiCode to a new environment such as Mujoco?
3. How sensitive is DiCode's performance to the various knobs/parameters involved in constructing the contexts supplied to the FM? If you have done any ablations on this, it would be helpful to share those findings.
4. It would also be helpful if you described (anecdotally) how you iterated and converged to DiCode's current design.

**Limitations:**

The authors provide an adequate discussion of limitations, such as being constrained by a fixed game engine and the costs of using large FMs relative to simpler UED methods. The complexity and number of knobs/parameters in DiCode, and the opacity of the FM's curricular decisions, make it difficult to identify the most important learnings and limitations of this work.

**Strengths And Weaknesses:**

### Strengths:

The idea of using foundation models to generate environment code subject to a fixed world engine is a nice compromise between flexibility and validity. Designing the right intermediate environments to bridge the gap between an agent's current and target capabilities is a difficult and arduous task. I think it makes a lot of sense to delegate this to an FM and try to automate the process.

DiCode is a nice intermediate between fixed environments that support only a few knobs that impose smooth adjustments, and disjoint environments that are too far apart. The idea of generating progressive stepping stones reminds me of this paper: "Progressive Safeguards for Safe and Model-Agnostic Reinforcement Learning" by Omi et al., which you might want to look at.

Several of the results are quite compelling. The example of making iron armor as a defensive prerequisite is a nice example of preparing for future survivability. Fig. 4 illustrates how the FM generates assists for making a task easier, e.g., "start with enhanced inventory for easier iron sword crafting", and then progressively removes these asissts. The average success rate DiCode maintians of ~0.5 is a very nice result, as it shows that it continuously challenges the agent over a long horizon, keeping the agent in a zone of proximal development.

### Weaknesses:

Much of the innovation of DiCode is hidden within the FM and its generations, making it difficult to extract learnings related to curriculum and agent progression. DiCode relies on existing scoring functions (learnability from Rutherford et al.) and sampling methods (PLR form Jiang et al.) to provide structured context to an FM. The actual design of progressive levels and decisions of what skills to acquire are all handled by the FM.

I think it's important to decode the FM's curriculum generation process, not just to extract insights but also to suggest stronger ablations and baselines. If you manually study the sequence of high-scoring, prioritized environments, is there a theme or principle you can identify that captures the FM's decisions? For example, it's possible the FM has simply discovered a better enumeration/ordering of skills in CraftAx. If so, you could create a baseline that procedurally generates environment(s) for each skill and trains the agent on them in sequence. This would also help address the challenge you mentioned of distinguishing useful stepping stones from unhelpful/distracting ones.

You somewhat overstate DiCode's ability to "dream" of future combat scenarios and prepare for them, such as the Defeat Gnome Warrior/Archer tasks. While it is impressive that DiCode achieves nonzero success rate on these tasks, to claim that is due to forward planning or some kind of imagination rollout is a big stretch. I think it's more likely that the FM is using prior knowledge of the task and context you provide to enumerate the necessary skills.

DiCode requires a lot of environment-specific work. The generation code manipulates CraftAx through a rich programmatic interface that is specific to CraftAx, requiring knowledge of the world topology and objects/resources/inventory (for manipulating initial state), interaction rules and progression logic (for manipulating transition dynamics), and achievements/success criteria (for manipulating goals). The structured context provided to the FM includes various components of this information. Applying DiCode to a new game or a new domain (e.g., the Mujoco robotics environment you mentioned) would require implementing all of these components for that environment.

The level generation process in DiCode is quite complex and involves many tunable parameters. In addition to the components mentioned above, there are additional parameters such as bonus rewards and status mappings that need to be set. The generation process conditions on this information to design and draft a new level, and then conducts a second round involving a new structured context to generate the actual Python code. The training process also adds a few parameters, such as the ratio of target environment vs. new levels, how often to introduce new levels, etc. How did you converge to this design and set of parameter choices? Was it a process of trial-and-error, iterative design, or something more/less systematic?

I would like to ask for ablations to identify which aspects of your design are crucial/necessary, but there are simply too many components and parameters to ablate. Perhaps a compromise is to design one or two simpler variants of DiCode and use them as baselines. The current baseline you use is a bit weak an uninformative, since it completely removes the parent level description and agent performance profile.

### Additional comments:

* Define \Lambda before its first use.
* It's not clear how using a fixed environment seed across training episodes for DR versus a fresh seed for PPO-GTrXL affects your ability to compare them.

---

> ### Author Rebuttal · Authors · 2026-03-31
>
> We thank Reviewer CwzL for their thoughtful review and recognition of our environment code-generation design.
>
> > Ablations (W6, Q3)
>
> We provide additional variants that each simplify one aspect of DiCode.
>
> | Method Variant | Seeds | Mean Return |
> | --- | --- | --- |
> | DiCode | 8 | 48.6 |
> | Random Parent Sampling | 4 | 46.8 |
> | Qwen80B | 5 | 45.9 |
> | Qwen30B | 5 | 44.4 |
> | Goal Only | 4 | 42.7 |
> | DiCode-OL | 5 | 41.1 |
>
> **Random Parent Sampling**, which replaces the combined learnability-prioritized and diversity-promoting selection with uniform random selection, shows a modest drop in mean return (46.8) but unlike DiCode, it has almost zero success rate on late-game tasks (e.g. defeat_gnome_warrior: 0.8% or defeat_gnome_archer: 0.2%), indicating that principled parent selection is critical for sustained late-game progression. **Goal Only**, which restricts the FM to goal selection without other environment reshaping, drops to near-baseline (42.7), confirming that the ability to modify initial states and transition dynamics is essential. The LLM scaling variants show progressively lower mean return despite identical prompts and API, confirming that FM capability directly impacts performance.
>
> We will include these ablations in the revision.
>
> The curves for all variants are available at https://sites.google.com/view/dreaming-in-code/ablation-curves .
>
> > Design Choices (W5, Q4)
>
> The overall design was guided by the need for expressive, closed-loop curriculum design through program synthesis in open-ended environments beyond the reach of standard UED. Our starting hypothesis was that having the FM design focused, goal-conditioned scenarios, rather than full environments where complex objectives remain unreachable, would yield clearer capability signals from the agent and enable more targeted curriculum design. An obstacle we encountered during development was reward alignment. The target environment provides dense early-game rewards, and the agent would exploit these instead of pursuing the level's designed goal. This undermined the core benefit of targeted environment shaping: the generated levels became no more useful than the target environment itself, since the agent never engaged with the intended skill. It also broke FM's feedback loop – the FM could not distinguish a level that was genuinely too hard from one the agent was simply ignoring. The adaptive bonus reward (Section 3.1) resolved this by incentivizing goal completion, restoring accurate learnability signals to the FM (note the Goal Only ablation retains the adaptive bonus). To maximize generation quality, we separated level design and code generation into two FM calls. We detail the target environment ratio choice in our response to Reviewer rZ2M, last Section.
>
> > Skill enumeration/Curriculum theme (W2, Q1)
>
> A natural concern is whether the FM has simply discovered a better enumeration of skills. As the Goal Only ablation above shows, giving the FM the same skill-selection capability without further environment reshaping drops performance to near-baseline (42.7 vs 48.6), therefore skill selection is not the primary source of DiCode's gains. Regarding using the procedural method used to create the test set, we note that Craftax's procedural generation varies only terrain layout via random seeds and the existing baselines (DR, PLR) already operate over this space.
>
> Regarding curriculum theme, high-scoring environments reveal a recurring pattern: the FM identifies capability gaps from the agent's performance profiles and designs levels that bridge them by progressively adjusting environmental support relative to the parent level. The FM's reasoning traces on our project website (Section 4.1) support this. The DiCode-OL ablation (41.1 vs 48.6) confirms that the FM's curriculum decisions depend on runtime reasoning over the agent's evolving capabilities and the parent level context, not static domain knowledge alone.
>
> > “Dreaming” clarification (W3)
>
> We clarify that 'dreaming' describes the process of imagining training situations that do not yet exist and materializing them into executable environments, tailored to the agent's current capabilities. The agent does train on these imagined scenarios, but we do not claim the FM performs planning or lookahead.
>
> > Domain adaptation effort (W4, Q2)
>
> We detail the adaptation effort, including a concrete MuJoCo scoping, in our response to Reviewer rZ2M in “Domain adaptation” Section.
>
> > Minor points
>
> We will define $\Lambda$ before its first use in the revision.
>
> Regarding the DR vs PPO-GTrXL seed regime: this is inherent to the methods, not a confound. DR resamples its seed buffer periodically, resetting within the same seeds a few times before resampling, following standard Craftax protocol (Matthews et al., 2024, Monette et al., 2025); PPO-GTrXL samples a fresh seed each episode. We will clarify in the revision.
>
> Thank you for the pointer to Omi et al.; we will cite it in the revision.

---

> > ### Author Rebuttal · Reviewer_CwzL · 2026-04-04
> >
> > Thank you for your response!
> >
> > Thank you for providing additional ablations of DiCode's components. Can you explain how Random Parent Sampling differs from removing the parent level description (as DiCode-OL does)? There are several other ablations I would like to see, but as I noted in my review this is not practical given how many components and parameters there are in DiCode's design.
> >
> > The performance difference between DiCode and Goal Only/DiCode-OL is not that large. I agree that the Goal Only ablation shows that environment reshaping is beneficial, but I think the reason this works is because it makes skill acquisition progressive by providing stepping stones. If you could directly make each skill progressive, e.g., by breaking it into three stages of difficulty, this could be a sufficient solution when combined with the right (possibly adaptive) skill order. Such a solution would demystify what is happening inside the FM blackbox.
> >
> > Thank you for describing your design process. The problems caused by dense early-game rewards are a much clearer and more systematic justification for adding the adaptive bonus reward. Please include this in your paper.
> >
> > The number of movable pieces in DiCode's design and the effort required to adapt it to a new domain remain high. I understand that some domain-specific adaptation is unavoidable, but there are too many different ways to design the programmatic interface for a new domain that I wouldn't now how to approach this systematically or have any faith in my evaluation of the resulting system.

---

> > > ### Author Response · Authors · 2026-04-05
> > >
> > > We thank reviewer CwzL for their continued engagement and thoughtful feedback, which has helped improve the paper.
> > >
> > > > Random Parent Sampling vs. DiCode-OL
> > >
> > > Random Parent Sampling selects the parent level uniformly at random but the FM still receives the full context: parent level description and agent performance profiles. DiCode-OL removes both the parent level description and the agent performance profiles from the FM's input. Together they isolate complementary aspects, where Random Parent Sampling tests whether which parent the FM builds on matters, while DiCode-OL tests whether the FM needs to be grounded in the agent's current capabilities and the parent level at all. Combined with Goal Only (no environment reshaping) and the FM scaling variants (Qwen 30B/80B), the ablations span four design axes: parent selection strategy, environment reshaping, closed-loop grounding, and FM capability.
> > >
> > > > stepping stones for progressive skill acquisition
> > >
> > > We note that the gains from environment reshaping and closed-loop grounding are over ablation variants that already include FM-based generation and the adaptive bonus, not over naive baselines. We agree that making skill acquisition progressive by providing stepping stones accurately describes what environment reshaping provides; indeed, this is how we describe DiCode's generated levels in Section 3. However, the manual version of this, defining difficulty stages and determining ordering per skill, is static by construction and cannot adapt to the agent's evolving trajectory. Even with adaptive ordering, as suggested, the difficulty stages within each skill remain fixed at design time, and adaptive ordering itself requires a mechanism to monitor agent capabilities and decide which skill to prioritize next. In a domain like Craftax, with deep hierarchies and compositional skill dependencies, this is itself a significant design problem. Manual staging also imposes a capability ceiling at the hardest stage the designer anticipated, making it unsuitable for open-ended learning. DiCode automates this stepping stone principle: the FM generates contextually appropriate levels conditioned on the agent's current capabilities continuously throughout training. A human could design any individual level, but cannot replicate this adaptive loop at scale. Moreover, the FM's reasoning traces and level designs are fully inspectable (Figure 4, Appendix C.1, project website: https://sites.google.com/view/dreaming-in-code/run-1), making curricular decisions more transparent than a black-box characterization suggests.
> > >
> > > > design process
> > >
> > > Thank you for this suggestion. The adaptive bonus motivation is briefly noted in Section 3.1, but we will expand it with the detailed justification in the revision.
> > >
> > > > programmatic interface
> > >
> > > We believe the design space for the programmatic interface is more constrained than it may appear. The interface serves a single purpose: to allow the FM to vary the initial state, transition dynamics, and goals of levels within a fixed engine. The main design decision is how much expressivity to grant the FM, balancing its ability to meaningfully reshape the environment against generation noise and compilation reliability. Moreover, this balance becomes easier to strike as FM capability increases. Our scaling ablations (Qwen 30B -> 80B -> 235B) show that stronger FMs achieve higher performance with the same interface, without any changes to the API design. We believe the guiding principle, expose the engine's control axes over initial state, transition dynamics, and goals, provides a clear starting point for any new domain. We will include a general domain adaptation guide in the revision to make this process concrete for practitioners.
> > >
> > > In summary, DiCode extends UED to open-ended environments with deep compositional dependencies that are beyond the reach of existing methods operating over low-dimensional parameter spaces. This is enabled by automating the stepping stone principle, where the FM generates adaptive, progressive curricula without manual stage design and without a predetermined ceiling on agent capability. We believe this direction will generalize to other domains where tasks can be specified as code.

---

### Decision · Program_Chairs · 2026-04-30

**Decision:**

Accept (regular)

**Comment:**

This paper proposes DiCode as a method that leverages LLMs to programmatically generate a curriculum that enables reinforcement learning algorithms to succeed on difficult, open-ended problems.

I recommend accepting this paper. We did not have a clear consensus about the paper, and I even read it myself to reach this decision. The main concern raised was how much we can attribute to DiCode and how much of what we see is simply due to the foundation model and the structured context it receives. Some of those concerns were addressed during the discussion phase, but ultimately, there were many concerns about the sheer complexity of the design, the hand-crafted API, and the various parameters/knobs. Upon reading the paper and reviewing the claimed contributions, I concluded that it has merit at the conceptual level and provides proof of possibility, which is why I recommend its acceptance.